# Double Descent Meets Out-of-Distribution Detection: Theoretical Insights and Empirical Analysis on the Role of Model Complexity

**Mouïn Ben Ammar**[◇,†,∗]**, David Brellmann**[†,∗]**, Arturo Mendoza**[†]**, Antoine Manzanera**[◇]**, Gianni Franchi**[◇]

U2IS Lab ENSTA Paris[◇], Palaiseau, FRANCE
Safran Tech[†], Chateaufort 78117, FRANCE
{first.last}@ensta.fr[◇], safrangroup.com[†]

## Abstract

**Out-of-distribution (OOD) detection** is essential for ensuring the reliability and safety of machine learning systems. In recent years, it has received increasing attention, particularly through post-hoc detection and training-based methods. In this paper, we focus on **post-hoc OOD detection**, which enables identifying OOD samples without altering the model's training procedure or objective. Our primary goal is to investigate the relationship between **model capacity** and its OOD detection performance. Specifically, we aim to answer the following question: *Does the Double Descent phenomenon manifest in post-hoc OOD detection?* This question is crucial, as it can reveal whether overparameterization, which is already known to benefit generalization, can also enhance OOD detection. Despite the growing interest in these topics by the classic supervised machine learning community, this intersection remains unexplored for OOD detection. We empirically demonstrate that the Double Descent effect does indeed appear in post-hoc OOD detection. Furthermore, we provide theoretical insights to explain why this phenomenon emerges in such setting. Finally, we show that the overparameterized regime does not yield superior results consistently, and we propose a method to identify the optimal regime for OOD detection based on our observations. Code available here

## 1 Introduction

Since the breakthrough of AlexNet in 2012 (Krizhevsky et al., 2012), deep learning has seen rapid progress and has become the foundation for solving a wide variety of complex tasks across domains such as vision, language, and robotics (LeCun et al., 2015; Bengio, 2009). While Occam's Razor suggests favoring simpler models with fewer parameters (Blumer et al., 1987), deep neural networks (DNNs) with massive overparameterization have nonetheless demonstrated remarkable generalization ability in practice.

Traditionally, generalization performance has been characterized by a U-shaped test error curve with respect to model complexity (Geman et al., 1992; Hastie, 2009), recommending a "sweet spot" where the model is expressive enough to avoid underfitting but not so complex that it over-fits. However, this classical view has been challenged in 2019 by the emergence of the *double descent* phenomenon (Belkin et al., 2019), which reveals a second descent in test error after the interpolation threshold. This insight has reshaped our understanding of the generalization behavior of overparameterized models.

Generalization typically refers to in-distribution (ID) performance—where test data are assumed to be drawn i.i.d. from the same distribution as training data. However, this closed-world assump-

---

[∗]Equal contribution

39th Conference on Neural Information Processing Systems (NeurIPS 2025).

tion (Scheirer et al., 2012) rarely holds in real-world scenarios. In open-world settings (Bendale & Boult, 2016; Drummond & Shearer, 2006), models frequently encounter *out-of-distribution* (OOD) inputs that differ significantly from training data, either due to *semantic shift* (e.g., new classes) (Hendrycks & Gimpel, 2017) or *covariate shift* (e.g., domain changes) (Ben-David et al., 2010; Wang & Deng, 2018; Li et al., 2017).

In this work, we focus on OOD detection under semantic shifts as defined in Yang et al. (2024), where the goal is to identify inputs from unseen classes. More precisely, we investigate *post-hoc* OOD detection methods (DeVries & Taylor, 2018; Liu et al., 2020; Sun et al., 2021), which operate on top of a pretrained classifier without altering its training process. These methods are widely used due to their practicality and modularity.

Our central research question is: **How does model capacity affect the effectiveness of post-hoc OOD detection?** While double descent has been studied in the context of ID generalization, its role in OOD detection remains unexplored. We empirically demonstrate that post-hoc OOD detection also exhibits a double descent curve with respect to model complexity. This observation is important as it suggests that overparameterization is not universally beneficial for OOD detection.

We conduct extensive experiments across different architectures (CNNs, ResNets, ViTs, Swin) and post-hoc detection methods to confirm this trend. Furthermore, under a simplified Gaussian mixture model, we derive theoretical insights using tools from random matrix theory that mirror the empirical findings. Surprisingly, we also observe cases where the underparameterized regime achieves better OOD detection performance, prompting us to investigate these cases through the lens of Neural Collapse (Papyan et al., 2020), which provides a geometric interpretation of the learned representations near convergence.

**Contributions.** We advance the theoretical and empirical understanding of model complexity in post-hoc OOD detection:

1. We are the first to empirically highlight a double descent phenomenon in OOD detection across both CNN and transformer architectures.

2. We provide theoretical insight using random matrix theory, showing that both ID and OOD risks peak at the interpolation threshold.

3. We show that overparameterization is not always optimal for OOD detection, and propose a Neural Collapse-based criterion to detect when simpler models may perform better.

## 2 Related work

**OOD Detection.** OOD detection research focuses on two primary directions: Training-based and post-hoc methods. We will focus on latter ones. These can be categorized based on the essential feature used for the scoring function. First, logit- or confidence-based methods leverage network logits to derive a confidence measure used as an OOD scoring metric (Hendrycks & Gimpel, 2017; DeVries & Taylor, 2018; Liu et al., 2020; Huang & Li, 2021; Hendrycks et al., 2022). A common

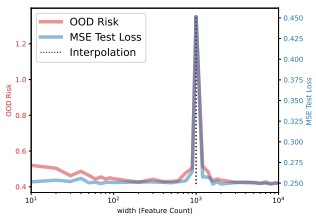
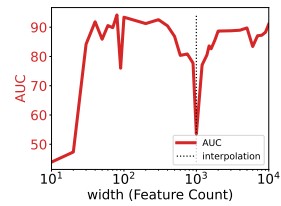
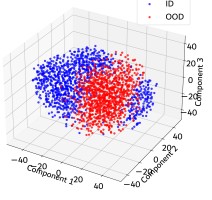

(a) OOD Risk and MSE Test Loss    (b) OOD detec. confidence score    (c) Input space projection

Figure 1: Illustration of the double descent phenomenon in a Random ReLU Feature model as a function of model width (log scale). **(a)** Evolution of the in-distribution (ID) Mean-Squared Error (MSE) and OOD detection risk. **(b)** Confidence-based OOD score defined in equation 14. **(c)** Three-dimensional t-SNE projection of the input space, visualizing the separation between ID and OOD samples. The ID samples ($n = 1,000$) are drawn from a Gaussian Mixture Model (GMM) fitted to a subset of CIFAR-10, while OOD samples are drawn from an independent GMM.

baseline for this methods is the softmax score (Hendrycks & Gimpel, 2017), which simply uses the model softmax prediction as the OOD score. Then, Energy (Liu et al., 2020) elaborates on it by computing the LogSumExp on the logits, thus offering empirical and theoretical advantages over the Softmax confidence score. Second, feature-based and hybrid methods (Lee et al., 2018b; Wang et al., 2022; Sun et al., 2022; Ming et al., 2023; Djurisic et al., 2023; Ammar et al., 2024) exploit the model's final representation to derive the scoring function. Hybrid methods further augments the scoring metrics defined on the features using logits as weighting factors. Mahalanobis (Lee et al., 2018b) estimates density on ID training samples using a mixture of class-conditional Gaussians. NECO (Ammar et al., 2024), on the contrary, leverages the geometric properties of Neural Collapse to construct a scoring function based on the relative norm of a sample within the subspace defined by the ID data. Besides OOD detection methods, Fang et al. (2024) have used risk formulations closely related to the one we use in this work to study PAC bounds for OOD detection.

**Double Descent.** The double descent risk curve was introduced by Belkin et al. (2019) to explain the good performance observed in practice by overparameterized models (Zhang et al., 2021; Belkin et al., 2018; Nakkiran et al., 2021) and to bridge the gap between the classical bias-variance trade-off theory and modern practices. Theoretical investigation into this phenomenon mainly focuses on various linear models in both regression and classification problems through the Random Matrix Theory (Louart et al., 2018; Liao et al., 2020; Jacot et al., 2020; Dereziński et al., 2020; Kini & Thrampoulidis, 2020; Mei & Montanari, 2022; Deng et al., 2022; Bach, 2024; Brellmann et al., 2024), techniques from statistical mechanics (D'Ascoli et al., 2020; Canatar et al., 2021), the VC theory (Lee & Cherkassky, 2022; Cherkassky & Lee, 2024), or novel bias-variance decomposition in deep neural networks (Yang et al., 2020). The double descent phenomenon has also been observed in experiments with popular neural network architectures (Belkin et al., 2019; Nakkiran et al., 2021). In addition to depending on the model complexity, the double descent phenomenon also depends on other dimensions such as the level of regularization (Liao et al., 2020; Mei & Montanari, 2022), the number of epochs (Nakkiran et al., 2021; Stephenson & Lee, 2021; Olmin & Lindsten, 2024), or the data eigen-profile (Liu et al., 2021a). Finally, the theoretical background on double descent and benign overparametrization developed by Bartlett et al. (2020) inspired subsequent works that focused on generalization under dataset shifts (Tripuraneni et al., 2021b; Hao et al., 2024; Kausik et al., 2024; Hao & Zhang, 2024). It is important to note that these dataset shifts concern scenarios where the model can generalize, *e.g.*, the class labels are same as in the training. These studies do not address OOD detection.

## 3 Preliminaries

**Notations.** For a real vector $\boldsymbol{v}$, we denote by $\|\boldsymbol{v}\|_2$ the Euclidean norm of $\boldsymbol{v}$. When the matrix $\boldsymbol{A}$ is full rank, we denote by $\boldsymbol{A}^+$ the Moore-Penrose inverse of $\boldsymbol{A}$. We depict by $[d] := \{1, \ldots, d\}$ the set of the $d$ first natural integers. For a subset $\mathcal{T} \subseteq [d]$, we denote by $\mathcal{T}^c := [d] \setminus \mathcal{T}$ its complement set. For a subset $\mathcal{T} \subseteq [d]$, a $d$-dimensional vector $\boldsymbol{v} \in \mathbb{R}^d$ and a $n \times d$ matrix $\boldsymbol{A} = \left[\boldsymbol{a}^{(1)} | \ldots | \boldsymbol{a}^{(n)}\right]^T \in \mathbb{R}^{n \times d}$, we use $\boldsymbol{v}_{\mathcal{T}} = [\boldsymbol{v}_j : j \in \mathcal{T}]$ to denote its $|\mathcal{T}|$-dimensional subvector of entries from $\mathcal{T}$ and $\boldsymbol{A}_{\mathcal{T}} = \left[\boldsymbol{a}_{\mathcal{T}}^{(1)} | \ldots | \boldsymbol{a}_{\mathcal{T}}^{(n)}\right]^T$ to denote the $n \times |\mathcal{T}|$ design matrix with variables from $\mathcal{T}$. For a subset $\mathcal{T} \subseteq [d]$, we define $R^d(\mathcal{T}) := \left\{\boldsymbol{x} \in \mathbb{R}^d \mid \boldsymbol{x}_{\mathcal{T}^c} = \boldsymbol{0}_{d-|\mathcal{T}|}\right\}$. We use $\lambda_{\min}(\boldsymbol{A})$ and $\lambda_{\max}(\boldsymbol{A})$ to depict the min and max eigenvalues of $\boldsymbol{A}$, respectively. $\mathcal{N}(\boldsymbol{0}_d, \boldsymbol{I}_d)$ denotes the standard multivariate Gaussian distribution of $d$ random variables.

**Supervised Learning Problems.** In supervised learning, we use training dataset $\mathcal{D} := \left\{(\boldsymbol{x}_1, y_1), \cdots, (\boldsymbol{x}_n, y_n)\right\}$ of $n$ independent and identically distributed (i.i.d.) samples drawn from an unknown distribution $P_{\mathcal{X}, \mathcal{Y}}$ over $\mathcal{X} \times \mathcal{Y}$. Using samples from the dataset $\mathcal{D}$, the objective is to find a predictor $\hat{f} : \mathcal{X} \to \mathcal{Y}$ among a class of functions $\mathcal{F}$ to predict the target $y \in \mathcal{Y}$ of a new sample $\boldsymbol{x} \in \mathcal{X}$. In particular, given a loss function $\ell : \mathcal{Y} \times \mathcal{Y} \to \mathbb{R}$, the objective is to minimize the expected risk (or loss) defined, for all $\hat{f} \in \mathcal{F}$, as:

$$R(\hat{f}) = \mathbb{E}_{(\boldsymbol{x}, y) \sim P_{\mathcal{X}, \mathcal{Y}}}\left[\ell\left(\hat{f}(\boldsymbol{x}), y\right)\right]. \tag{1}$$

Typically, we choose the mean-squared loss $\ell(\hat{f}(\boldsymbol{x}), y) = \left(\hat{f}(\boldsymbol{x}) - y\right)^2$ for regression problems or the zero-one loss $\ell(\hat{f}(\boldsymbol{x}), y) = \mathbb{1}_{\hat{f}(\boldsymbol{x}) \neq y}$ for classification problems. We denote the optimal predictor

by $f^* := \arg\min_{f \in \mathcal{F}} R(f)$. Since the distribution $P_{\mathcal{X},\mathcal{Y}}$ is unknown in practice, we instead try to minimize an empirical version of the expected risk based on the dataset $\mathcal{D} := \{(\boldsymbol{x}_i, y_i)\}_{i=1}^{n}$:

$$R_{emp}(\hat{f}) = \tfrac{1}{n}\sum_{i=1}^{n} \ell(\hat{f}(\boldsymbol{x}_i), y_i). \tag{2}$$

**Out-of-Distribution Detection.**    In machine learning problems, we usually assume that the test data distribution is similar to the training data distribution (the closed-world assumption). As this is not the case in real-world applications, the Out-of-Distribution (OOD) detection (Yang et al., 2024) aims to flag inputs that significantly deviate from the training data to prevent unreliable predictions. In the following, we denote by $P_{\mathcal{X},\mathcal{Y}}^{\text{OOD}}$ a distribution over $\mathcal{X} \times \mathcal{Y}$ that differs from the training distribution $P_{\mathcal{X},\mathcal{Y}}$. As proposed in  (Yang et al., 2024) we consider OOD that involves a semantic shift and represent concepts or labels not seen during training. A popular class of OOD detection techniques relies on the definition of a scoring function $s(\cdot\,;\hat{f}) = \hat{f}(\cdot)$, which uses the probability predictions of the classifier $\hat{f}(\cdot)$ as scores to flag an instance $\boldsymbol{x}$ as OOD when the score probability prediction is below a certain threshold $\lambda$. For example, a common approach is to use the Maximum Softmax Probability (MSP) (Hendrycks & Gimpel, 2017) that returns the higher softmax probabilities of the predictor $\hat{f}(\cdot)$ as a scoring function to measure the prediction confidence.

# 4    Insights on the Double Descent for the Binary Classification in Gaussian Covariate Model

In this section, we introduce the expected OOD risk metric and we present our main theoretical results on binary least-squares classifiers applied to Gaussian data. We assume that $\mathcal{X} \subseteq \mathbb{R}^d$ and $\mathcal{Y} := [0, 1]$. Let $\phi : \mathbb{R} \to \mathcal{Y}$ be a mapping, we denote by $\mathcal{F}_d := \{f : \mathcal{X} \to \mathcal{Y}, \boldsymbol{x} \mapsto \phi(\boldsymbol{x}^T \boldsymbol{w}) \mid \boldsymbol{w} \in \mathbb{R}^d\}$ the class of functions considered in this study.

## 4.1    System Model

**Gaussian Covariate Model & Binary Classification.**    Let $\boldsymbol{w}^* \in \mathbb{R}^d$ and let $f^* : \boldsymbol{x} \mapsto \phi(\boldsymbol{x}^T \boldsymbol{w}^*)$ be an optimal binary classifier. We assume we have a training dataset $\mathcal{D} := \{(\boldsymbol{x}_i, y_i)\}_{j=1}^{n}$ of $n$ i.i.d samples drawn from a Gaussian covariate model, *i.e.*, from a distribution $P_{\mathcal{X},\mathcal{Y}}$ over $\mathcal{X} \times \mathcal{Y}$; where $\boldsymbol{x}_i \sim \mathcal{N}(\boldsymbol{0}_d, \boldsymbol{I}_d)$ and $y_i = \phi(z_i) = \phi(\boldsymbol{x}_i^T \boldsymbol{w}^* + \epsilon_i)$ is the noisy response of $\boldsymbol{x}_i$ with respect to $f^*(\cdot)$ in which $\epsilon_i \sim \mathcal{N}(0, \sigma^2)$ is a noise capturing approximation errors on logits $z_i$ with $\sigma > 0$. In binary classification problems, the objective is to find a classifier $\hat{f}(\cdot) \in \mathcal{F}_d$ that fits $f^*(\cdot)$ using the training dataset $\mathcal{D}$. Although simple, the Gaussian covariate model is also considered in the double descent literature to provide theoretical insights (Belkin et al., 2020; Mei & Montanari, 2022).

**Least-Squares Binary Classifiers.**    In this section, we consider least-squares binary classifiers to approximate logits of the optimal binary classifier $f^*(\cdot)$. Specifically, we define the least-squares binary classifier $\hat{f} : \boldsymbol{x} \mapsto \phi(\boldsymbol{x}^T \hat{\boldsymbol{w}})$ in which $\hat{\boldsymbol{w}}$ is obtained by analytically solving:

$$\hat{\boldsymbol{w}} = \arg\min_{\boldsymbol{w}} R_{emp}(\boldsymbol{w}) = \tfrac{1}{n}\sum_{i=1}^{n}(\boldsymbol{x}_i^T \boldsymbol{w} - z_i)^2 = \tfrac{1}{n}\|\boldsymbol{X}\boldsymbol{w} - \boldsymbol{z}\|_2^2, \tag{3}$$

were $\boldsymbol{X} = [\boldsymbol{x}_1, \ldots, \boldsymbol{x}_n]^T \in \mathbb{R}^{n \times d}$ is the data matrix containing the $n$ samples $\boldsymbol{x}_i \in \mathbb{R}^d$ and $\boldsymbol{z} = [\boldsymbol{z}_1, \ldots, \boldsymbol{z}_n]^T \in \mathbb{R}^n$ is the target vector of noisy logits. We consider a particular subset of least-squares binary classifier $\hat{f}_{\mathcal{T}} \in \mathcal{F}_d$ that uses a subset $\mathcal{T} \subseteq [d]$ of $p$ features that fits coefficients $\hat{\boldsymbol{w}} \in R^d(\mathcal{T})$ as

$$\hat{\boldsymbol{w}}_{\mathcal{T}} = \boldsymbol{X}_{\mathcal{T}}^{+}\boldsymbol{z} \quad \text{and} \quad \hat{\boldsymbol{w}}_{\mathcal{T}^c} = \boldsymbol{0}_{d-p}. \tag{4}$$

**Out-of-Distribution Risk.**    To measure the ability of binary classifiers $\hat{f}(\cdot) \in \mathcal{F}_d$ to provide prediction confidence on samples drawn from both the training distribution $P_{\mathcal{X},\mathcal{Y}}$ and the OOD distribution $P_{\mathcal{X},\mathcal{Y}}^{\text{OOD}}$, we introduce an expected OOD risk function similar to the expected risk defined in equation 3. Let $f^{\text{OOD}} : \boldsymbol{x} \mapsto \phi(\boldsymbol{x}^T \boldsymbol{w}^{\text{OOD}}) \in \mathcal{F}_d$ be an optimal classifier such that $f^{\text{OOD}}(\boldsymbol{x})$ is

close to 0.5 when the sample $\boldsymbol{x}$ is more likely drawn from the $P_{\mathcal{X},\mathcal{Y}}^{\text{OOD}}$ and close to $f^*(\boldsymbol{x})$ when $\boldsymbol{x}$ is more likely drawn from $P_{\mathcal{X},\mathcal{Y}}$. We define the expected OOD risk $R_{\text{OOD}} : \mathcal{F}_d \rightarrow \mathbb{R}$ as:

$$R_{\text{OOD}}(\hat{f}) \;=\; \mathbb{E}_{(\boldsymbol{x},\cdot)\sim P_{\mathcal{X},\mathcal{Y}}}\big[\big(\hat{f}(\boldsymbol{x}) - f^{\text{OOD}}(\boldsymbol{x})\big)^2\big] \;+\; \mathbb{E}_{(\boldsymbol{x},\cdot)\sim P_{\mathcal{X},\mathcal{Y}}^{\text{OOD}}}\big[\big(\hat{f}(\boldsymbol{x}) - f^{\text{OOD}}(\boldsymbol{x})\big)^2\big], \quad (5)$$

which depicts the expected risk of the binary classifier $\hat{f}(\cdot)$ on the loss function $\ell : (\hat{y}, y) \mapsto (\hat{y} - y)^2$ and distributions $P_{\mathcal{X},\mathcal{Y}}$ and $P_{\mathcal{X},\mathcal{Y}}^{\text{OOD}}$.

**Remark 4.1.** A low value for $R_{\text{OOD}}(\hat{f})$ indicates two aspects: $(i)$ the classifier $\hat{f}(\cdot)$ is confident on predictions over the distribution $P_{\mathcal{X},\mathcal{Y}}$, which corresponds to a low $\mathbb{E}_{(\boldsymbol{x},\cdot)\sim P_{\mathcal{X},\mathcal{Y}}}\big[\big(\hat{f}(\boldsymbol{x}) - f^{\text{OOD}}(\boldsymbol{x})\big)^2\big]$; and/or $(ii)$ $\hat{f}(\cdot)$ is not confident on predictions over the distribution $P_{\mathcal{X},\mathcal{Y}}^{\text{OOD}}$, which is reflected by a low $\mathbb{E}_{(\boldsymbol{x},\cdot)\sim P_{\mathcal{X},\mathcal{Y}}^{\text{OOD}}}\big[\big(\hat{f}(\boldsymbol{x}) - f^{\text{OOD}}(\boldsymbol{x})\big)^2\big]$. In particular, $R_{\text{OOD}}(\hat{f})$ is minimized when logits are maximally confident on ID samples (only one logit is non-zero) and uniformly distributed on OOD samples.

**Remark 4.2.** Note that the expected OOD risk defined in equation 5 can be extended to multi-class classifiers $\hat{f}(\cdot)$ using the softmax function with

$$R_{\text{OOD}}(\hat{f}) = \mathbb{E}_{(\boldsymbol{x},\cdot)\sim P_{\mathcal{X},\mathcal{Y}}}\big[\big(\|\hat{f}(\boldsymbol{x})\|_{\infty} - \|f^{\text{OOD}}(\boldsymbol{x})\|_{\infty}\big)^2\big] + \mathbb{E}_{(\boldsymbol{x},\cdot)\sim P_{\mathcal{X},\mathcal{Y}}^{\text{OOD}}}\big[\big(\|\hat{f}(\boldsymbol{x})\|_{\infty} - \|f^{\text{OOD}}(\boldsymbol{x})\|_{\infty}\big)^2\big],$$

where $f^{\text{OOD}}(\boldsymbol{x})$ is close to $1/C$ when the sample $\boldsymbol{x}$ is more likely drawn from the $P_{\mathcal{X},\mathcal{Y}}^{\text{OOD}}$, $C$ depicts the number of classes, and $\|\cdot\|_{\infty}$ denotes the infinity norm.

In order to use the Random Matrix Theory, we make the following assumptions.

**Assumption 4.1.** The activation function $\phi(\cdot)$ is strictly monotonically increasing. Furthermore, its derivative $\phi'(\cdot)$ is strictly positive and bounded.

**Remark 4.3.** This assumption holds for many of the activation functions traditionally considered in neural networks, such as sigmoid functions.

**Assumption 4.2.** Let $\boldsymbol{\Sigma}, \boldsymbol{\Sigma}^{\text{OOD}} \in \mathbb{R}^{d\times d}$ defined as

$$\boldsymbol{\Sigma} = \mathbb{E}_{(\boldsymbol{x},\cdot)\sim P_{\mathcal{X},\mathcal{Y}}}\left[\left(\frac{\phi(\boldsymbol{x}^T\hat{\boldsymbol{w}}) - \phi(\boldsymbol{x}^T\boldsymbol{w}^{\text{OOD}})}{\boldsymbol{x}^T\hat{\boldsymbol{w}} - \boldsymbol{x}^T\boldsymbol{w}^{\text{OOD}}}\right)^2 \boldsymbol{x}\boldsymbol{x}^T\right] \text{ and } \boldsymbol{\Sigma}^{\text{OOD}} = \mathbb{E}_{(\boldsymbol{x},\cdot)\sim P_{\mathcal{X},\mathcal{Y}}^{\text{OOD}}}\left[\left(\frac{\phi(\boldsymbol{x}^T\hat{\boldsymbol{w}}) - \phi(\boldsymbol{x}^T\boldsymbol{w}^{\text{OOD}})}{\boldsymbol{x}^T\hat{\boldsymbol{w}} - \boldsymbol{x}^T\boldsymbol{w}^{\text{OOD}}}\right)^2 \boldsymbol{x}\boldsymbol{x}^T\right].$$

We assume that

$$\arg\min_{\boldsymbol{a}\in\mathbb{R}^d} \frac{\boldsymbol{a}^T\boldsymbol{\Sigma}\boldsymbol{a}}{\boldsymbol{a}^T\boldsymbol{a}} \notin R^d(\mathcal{T}) \quad \text{and} \quad \arg\min_{\boldsymbol{a}\in\mathbb{R}^d} \frac{\boldsymbol{a}^T\boldsymbol{\Sigma}^{\text{OOD}}\boldsymbol{a}}{\boldsymbol{a}^T\boldsymbol{a}} \notin R^d(\mathcal{T}).$$

## 4.2 Out-Of-Distribution Risk

Leveraging the Random Matrix Theory and extending the Theorem 1 in Belkin et al. (2020), we can derive bounds for the expected risk (Appendix A.1) and for the expected OOD risk (proof in Appendix A.2) of the subset of classifiers defined with equation 4:

**Theorem 1.** *Let* $(p, q) \in [d]^2$ *such that* $p + q = d$, $\mathcal{T} \subseteq [d]$ *with* $|\mathcal{T}| = p$ *an arbitrary subset of the $d$ first natural integers, and* $\mathcal{T}^c := [d] \setminus \mathcal{T}$ *its complement set. Let* $\hat{\boldsymbol{w}} \in R^d(\mathcal{T})$ *such that* $\hat{\boldsymbol{w}}_{\mathcal{T}} = \boldsymbol{X}_{\mathcal{T}}^{+}\boldsymbol{z} \in \mathbb{R}^p$ *and* $\hat{\boldsymbol{w}}_{\mathcal{T}^c} = \boldsymbol{0}_q \in \mathbb{R}^q$. *Under Assumptions 4.1-4.2, the expected OOD risk on the predictor* $\hat{f}_{\mathcal{T}} : \boldsymbol{x} \mapsto \phi(\boldsymbol{x}^T\hat{\boldsymbol{w}})$ *satisfies*

$$c\,c(n,p) \leq \mathbb{E}_{\boldsymbol{X}}\big[R_{OOD}(\hat{f})\big] \leq C\,c(n,p),$$

*where* $c, C > 0$ *and*

$$c(n,p) = \begin{cases} \frac{p}{n-p-1}\left(\|\boldsymbol{w}_{\mathcal{T}^c}^{OOD}\|_2^2 + \sigma^2\right) + \|\boldsymbol{w}_{\mathcal{T}^c}^{OOD}\|_2^2 & \text{if } p \leq n-2, \\ +\infty & \text{if } n-1 \leq p \leq n+1, \\ \left(1 - \frac{n}{p}\right)\|\boldsymbol{w}_{\mathcal{T}}^{OOD}\|_2^2 + \frac{n}{p-n-1}\left(\|\boldsymbol{w}_{\mathcal{T}^c}^{OOD}\|_2^2 + \sigma^2\right) + \|\boldsymbol{w}_{\mathcal{T}^c}^{OOD}\|_2^2 & \text{if } p \geq n+2. \end{cases}$$

$$(6)$$

**Remark 4.4.** While our theoretical insights are inspired by Theorem 1 in Belkin et al. (2020), we extend the framework to classification and OOD data. This extension is non-trivial, as the original result solely focuses on classical supervised regression. Notably, Theorem 1 derives inequalities for the expected OOD risk unlike the equalities found in Belkin et al. (2019) original formulation.

**Remark 4.5.** We find $\mathbb{E}_{\boldsymbol{X}}\big[R_{\text{OOD}}(\hat{f})\big] = \infty$ around $p = n$, which is characteristic of a double descent phenomenon. This result suggests that OOD scoring functions based on the prediction confidence of binary classifiers $\hat{f}(\cdot)$ exhibit a double descent phenomenon similar to what has been reported for the expected risk (see Appendix A.1). Like the expected risk in the double descent literature (Mei & Montanari, 2022; Louart et al., 2018; Liao et al., 2020; Bach, 2024), this result identifies the ratio $p/n$ as the complexity of a linear model to describe an under- ($p/n < 1$) and an over- ($p/n > 1$) parameterized regimes for the expected OOD risk with a phase transition around $p/n = 1$ characterized by a peak.

## 5 Empirical Evidence: Double Descent in Practice

In this section, we provide an empirical evaluation of different OOD detection methods with respect to the model width across multiple neural network architectures.

### 5.1 Toy Example : Gaussian Covariate Model & OOD Detection

We investigate the double descent phenomenon in both generalization and OOD detection using the Gaussian covariate model introduced in Section 4.1. The data is generated from a binary Gaussian Mixture Model (GMM), as illustrated in Figure 1. Our model employs Random ReLU Features (RRF): inputs are mapped to a high-dimensional space via random projections followed by a ReLU activation, and then classified using a linear binary classifier. The OOD samples are drawn from a distinct Gaussian distribution, as described in Section 4.1. By varying the model width, we observe that both the in-distribution Mean-Squared Error (MSE) and the expected OOD risk exhibit double descent behavior, peaking near the interpolation threshold (Figure 1). These results show a concrete empirical manifestation of the theoretical Gaussian setup. Full details are in Appendix D.1.

### 5.2 Setup of the real case

**General Setup.** We perform experiments on multiple DNN architectures: ResNet-18 (He et al., 2016), ResNet-34 (Appendix D.4), a 4-block convolutional neural network (CNN), Vision Transformers (ViTs) (Dosovitskiy et al., 2021) and Swin Transformers (Liu et al., 2021b).

**Model Setup.** To replicate double descent, we follow the experimental setup from Nakkiran et al. (2021), which uses ResNet-18 as the baseline architecture. We apply a similar setup to the 4-block CNN model, ViTs and Swin. We vary the model capacity by altering the hidden dimension (denoted as $k$) per layer, with values ranging from $k = 1$ to $k = 128$. ResNet-18, which uses a hidden dimension of 64 channels, operates within the overparameterized regime. The depth of the models is kept constant to isolate the effects of width (effective model complexity). The convolutional models are trained using the cross-entropy loss function, with a learning rate of $10^{-4}$ and the Adam optimizer for 4 000 epochs. This extended training regime ensures that models converge for all explored model widths. Moreover, each experiment is conducted five times (with different random seeds). Further details on the experimental setup for the Transformers are given in the Appendix B.2.

**Label Noise.** To observe the double descent effect, we introduce label noise into the training set by randomly swapping 20% of the labels. This setup simulates real-world scenarios, in which noisy data is common. The models are trained on this noisy dataset but evaluated on a clean test set. Random data augmentations, including random cropping and horizontal flipping, are applied during training. Experiments on the noiseless case are presented in D.6 and OOD risk curves are presented in D.5.

### 5.3 Evaluation Metrics

We evaluate both generalization and OOD detection using multiple metrics:

- **Generalization**: We report the test accuracy for in-distribution (ID) classification tasks.

- **OOD Detection**: We measure OOD detection performance using the area under the receiver operating characteristic curve (`AUC`), which is threshold-free and widely adopted in OOD detection research. A higher `AUC` indicates better performance.
- **Neural Collapse**: We report NC metrics, which, as noted in Ammar et al. (2024); Haas et al. (2023); Zhao & Cao (2023), are associated with certain aspects of OOD detection.

## 5.4 OOD Datasets

For OOD detection, we evaluate each model using six well-established OOD benchmark datasets: **Textures** (Cimpoi et al., 2014), **Places365** (Zhou et al., 2017), **iNaturalist** (Van Horn et al., 2018), a 10 000 image subset from (Huang & Li, 2021), **ImageNet-O** (Hendrycks et al., 2021) and **SUN** (Xiao et al., 2010). For experiments where CIFAR-10 (or CIFAR-100) is the in-distribution dataset, we also include CIFAR-100 (or CIFAR-10) as an additional OOD benchmark. CIFAR-10/100 contains 50 000 training images and 10 000 test images.

## 5.5 OOD Detection Methods

In order to have a discussion that generalizes across different OOD detection methods, we evaluate several state-of-the-art methods, categorized by the information they rely on:

- **Logit-based methods**: Maximum Softmax Probability (MSP) (Hendrycks & Gimpel, 2017), Energy scores (Liu et al., 2020), React (Sun et al., 2021), MaxLogit and KL-Matching (Hendrycks et al., 2022),
- **Feature-based methods**: Mahalanobis (Lee et al., 2018b) and Residual (Wang et al., 2022).
- **Hybrid methods**: ViM (Wang et al., 2022), ASH (Djurisic et al., 2023) and NECO (Ammar et al., 2024).

Although the double descent effect is observed in all of our experiments, results from only a few representative methods are presented in the main paper. Additional results can be found in Appendix D.

## 5.6 OOD Detection and Double Descent

**Double Descent & OOD Detection.** The primary question addressed in this section is whether the double descent phenomenon extends to OOD detection, as suggested by our theoretical insights. We conduct experiments on CIFAR-10 and CIFAR-100 as ID datasets, and assess OOD detection as a function of model widths. In particular, Figure 2 depicts the evolution of generalization error and OOD detection performance (`AUC`) for a challenging covariate shift scenario between CIFAR-10 and CIFAR-100. Refer to Appendix D for more results on multiple OOD datasets. Figure 2 shows a double descent phenomenon in all models, with logit-based and hybrid OOD detection methods exhibiting a similar behavior. Experiments evaluating the OOD risk derived from Theorem 1 are provided in Appendix D.5. As shown in the appendix, the OOD risk function also exhibits a double descent phenomenon. This alignment between our theoretical insights and experimental results further reinforces the validity of our analysis.

**Feature-Based Techniques & Interpolation Threshold.** In some cases, no double descent curve is observed for feature-based techniques. This result suggests that the double descent depends either on the used architecture or the data, as discussed in Appendix E.3. Furthermore, we observe that the interpolation threshold is not always perfectly consistent across OOD datasets or techniques. Those observations are consistent with the Nakkiran et al. (2021)'s results on the CIFAR-10 and CIFAR-100 datasets. Those results suggest that the effective model complexity (EMC) framework (Nakkiran et al., 2021) defined for the generalization error can be extended to OOD detection.

**Smaller Models for OOD Detection.** Interestingly, in many cases, smaller models are very good OOD detectors. This is particularly relevant for applications with limited computational resources. While techniques such as pruning (Frankle & Carbin, 2019) and quantization (Gholami et al., 2022) can also be used to reduce model size, our study focuses specifically on the impact of model complexity. We believe this behavior may be due to smaller models utilizing their parameters more efficiently. This raises the question of when a lower-complexity model might be more advantageous

than a deep model for OOD detection. The conditions under which this choice becomes optimal will be discussed in Section 5.7.

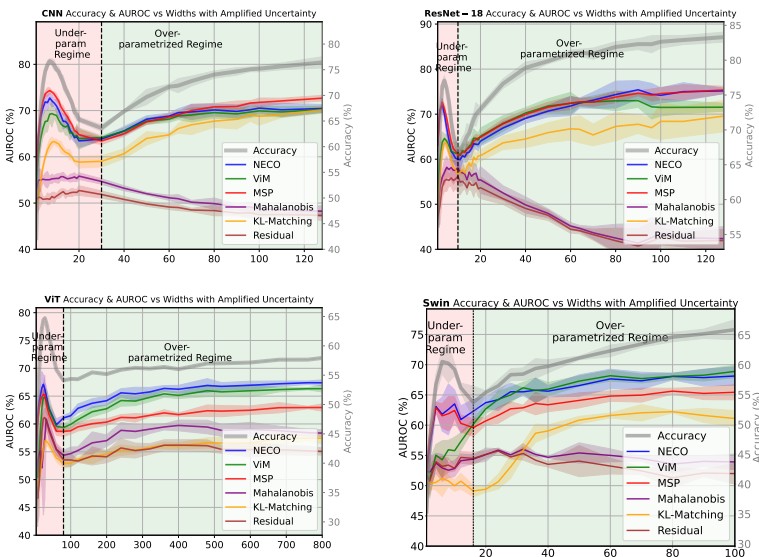

Figure 2: Accuracy and AUC OOD detection metric versus model width. Experiments performed on CNN, ResNet-18, ViT and Swin architectures with CIFAR10 and CIFAR100 as ID and OOD datasets.

**Discussion on OOD Methods.** OOD detection depends on two key factors: (i) the quality of learned representations, and (ii) the reliability of the confidence score. Logit-based methods rely mainly on model output logits, which are sensitive to model size and complexity (Hendrycks & Gimpel, 2017; Liang et al., 2018), making them closely aligned with the double descent behavior. These methods often show smoother double descent trends. In contrast, feature-based methods depend on how well the model separates ID and OOD data in latent space, which may not always be affected by double descent in the same way.

**Discussion on Architectures.** Architectural biases significantly affect representation quality and OOD performance (Hein et al., 2019; Tripuraneni et al., 2021a). While all models show double descent, their performance differs: ResNet-18 and Swin improve in the overparameterized regime, CNN performs similarly across regimes, and ViT suffers from poor generalization when overparameterized. To better understand this, we analyze the latent representations using the Neural Collapse framework (Papyan et al., 2020).

### 5.7 Overparameterization vs. Underparameterization through the Lens of Neural Collapse

**From Double Descent to Neural Collapse.** We analyze how representation geometry affects the double descent phenomenon in OOD detection. While increased model complexity often improves performance beyond the interpolation threshold, this trend is not universal (Fig. 2). Some models perform better in the underparameterized regime. To explain this, we use the Neural Collapse (NC) framework (Papyan et al., 2020; Ming et al., 2023; Haas et al., 2023; Ammar et al., 2024), which describes how deep models' final-layer features converge to a structured configuration where samples collapse around their respective class centroids.

**NC1 Metric for Analyzing Overparameterization.** We use the NC1 metric to measure the clustering quality of representations: $\text{NC1} = \text{Tr}\big[\frac{\mathbf{\Sigma}_W \mathbf{\Sigma}_B^+}{C}\big]$, where $\mathbf{\Sigma}_W$ and $\mathbf{\Sigma}_B$ are the intra- and inter-class covariances, and $C$ is the number of classes. Lower NC1 values indicate better class separation. To quantify the effect of overparameterization, we compute the ratio:

$$NC1_{u/o} = \frac{NC1_u}{NC1_o}, \tag{7}$$

where $NC1_u$ and $NC1_o$ are the NC1 values in the under- and overparameterized regimes optimums, respectively. Higher $NC1_{u/o}$ implies better separation with increased capacity.

**Empirical Insights.**  As shown in Table 1, $NC1_{u/o}$ aligns with OOD performance trends. Overparameterized models with improved $NC1_{u/o} > 1$ typically detect OOD better on the overparamatrized regime. Except the CNN architecture, most architectures seem to show improved performance in the overparamatrized regime. These findings suggest that representation collapse, as captured by NC1, might indicate OOD detection quality in complex models. Further studies on broader datasets are needed to generalize this insight.

Table 1: Models performance in terms of AUC in the underparametrized local minima ($\text{AUC}_u$) and the overparametrized maximum width ($\text{AUC}_o$), *w.r.t* $NC1_{u/o}$ value. Best is highlighted in green when $\text{AUC}_o$ is higher, red when $\text{AUC}_u$ is higher and blue if both AUC are within standard deviation range. The highest AUC value per-dataset and per-architecture is highlighted in **bold**.

| Model | $NC1_{u/o}$ | Method | SUN | | Places365 | | CIFAR-100 | |
|---|---|---|---|---|---|---|---|---|
| | | | $\text{AUC}_u$ ↑ | $\text{AUC}_o$ ↑ | $\text{AUC}_u$ ↑ | $\text{AUC}_o$ ↑ | $\text{AUC}_u$ ↑ | $\text{AUC}_o$ ↑ |
| CNN | 0.88 | Softmax score | **76.09**±0.96 | 75.08±0.75 | **75.95**±0.76 | 74.59±0.45 | **74.33**±0.32 | 72.68±0.31 |
| | | MaxLogit | 72.98±2.22 | 60.13±0.35 | 73.33±2.17 | 61.25±0.42 | 73.38±0.39 | 70.37±0.34 |
| | | Energy | 68.08±3.66 | 59.73±0.36 | 69.00±3.56 | 60.90±0.43 | 70.78±0.73 | 70.24±0.35 |
| | | Energy+ReAct | 59.12±4.73 | 47.49±0.58 | 60.79±4.53 | 49.45±0.52 | 66.00±1.58 | 63.57±0.51 |
| | | NECO | 70.43±2.53 | 64.22±1.36 | 71.20±2.59 | 63.59±0.98 | 72.40±0.95 | 70.17±0.76 |
| | | ViM | 59.94±2.01 | 59.77±0.66 | 61.88±1.89 | 60.93±0.40 | 69.23±0.78 | 70.25±0.35 |
| | | ASH-P | 68.60±3.59 | 60.36±0.37 | 69.35±3.50 | 61.48±0.43 | 71.11±0.53 | 70.45±0.41 |
| ResNet | 1.96 | Softmax score | 71.18±0.93 | 75.82±0.89 | 71.22±0.93 | **75.52**±0.88 | 71.21±0.48 | **75.37**±0.42 |
| | | MaxLogit | 70.64±1.53 | 72.51±1.03 | 70.69±1.29 | 72.64±0.94 | 69.76±0.39 | 73.65±0.38 |
| | | Energy | 69.11±2.49 | 72.46±1.03 | 69.19±2.08 | 72.59±0.94 | 67.58±0.46 | 73.61±0.39 |
| | | Energy+ReAct | 69.57±2.35 | 71.83±0.88 | 69.63±1.93 | 71.97±0.78 | 67.25±0.91 | 72.63±0.45 |
| | | NECO | 70.39±2.30 | **75.60**±1.56 | 70.46±1.85 | 75.20±1.42 | 69.92±0.36 | 75.28±0.50 |
| | | ViM | 66.54±1.99 | 74.44±0.65 | 65.38±1.99 | 73.42±0.65 | 64.61±0.47 | 71.54±0.44 |
| | | ASH-P | 69.11±2.49 | 71.73 ±1.09 | 69.19±2.08 | 71.85±0.98 | 67.58±0.46 | 72.89±0.35 |
| Swin | 1.70 | Softmax score | 58.82±2.98 | 67.91±0.69 | 59.01±2.90 | 67.66±0.59 | 61.89±1.42 | 65.44±0.62 |
| | | MaxLogit | 59.91±3.11 | 70.75±0.45 | 59.84±3.40 | 70.46±0.46 | 61.79±1.73 | 66.95±0.53 |
| | | Energy | 60.12±3.68 | 70.79±0.42 | 58.77±3.98 | 70.50±0.44 | 55.52±2.10 | 66.92±0.51 |
| | | Energy+ReAct | 60.16±3.79 | 71.15±0.44 | 58.85±4.03 | 70.83±0.48 | 55.53±2.10 | 67.27±0.52 |
| | | NECO | 64.26±3.20 | **73.29**±0.75 | 63.88±3.37 | **72.38**±0.66 | 62.62±2.10 | 68.13±0.76 |
| | | ViM | 60.68±2.61 | 71.69±0.19 | 58.34±2.60 | 71.39±0.15 | 55.95±0.77 | 68.89±0.51 |
| | | ASH-P | 59.49±4.21 | 70.74±0.44 | 58.15±4.42 | 70.42±0.40 | 55.10±2.28 | **66.89**±0.55 |
| ViT | 2.32 | Softmax score | 66.28±0.19 | 64.87±0.27 | 66.26±0.36 | 64.61±0.26 | 65.18±0.38 | 62.96±0.33 |
| | | MaxLogit | 66.09±1.48 | 70.30±0.46 | 66.13±1.50 | 69.79±0.26 | 64.60±0.35 | 66.69±0.39 |
| | | Energy | 64.79±2.81 | 70.50±0.48 | 64.86±2.65 | 69.98±0.26 | 63.08±0.44 | 66.79±038 |
| | | Energy+ReAct | 64.51±2.93 | 70.51±0.49 | 64.65±2.75 | 69.97±0.26 | 62.86±0.58 | 66.78±039 |
| | | NECO | 67.61±1.61 | **75.89**±0.47 | 67.47±1.68 | **74.29**±0.29 | 66.28±0.54 | **67.40**±0.27 |
| | | ViM | 63.14±3.54 | 72.25±0.37 | 63.30±3.36 | 71.41±0.15 | 64.81±0.65 | 66.34±0.30 |
| | | ASH-P | 64.79±2.81 | 70.27±0.50 | 64.86±2.65 | 69.79±0.25 | 63.08±0.44 | 66.61±0.36 |

**Empirical Evidence of Depth-Wise Double Descent.**  We also investigate the influence of model depth and assess OOD detection and accuracy as functions of model depth with fixed width in Appendix F. Our results highlight similar double descent patterns in both ID generalization and post-hoc OOD detection. To the best of our knowledge, these are the first results illustrating a depth-wise double descent.

# 6   Conclusion

In this paper, we presented empirical and theoretical insight that the double descent phenomenon also impacts OOD detection across architectures. Using Random Matrix Theory we showed that both prediction and OOD risks peak at the interpolation threshold for least-squares classifiers on Gaussian data. We have studied the double descent phenomenon across various architectures and OOD detection algorithms. Additionally, we provided insights into why the overparameterized regime can perform better or worse than the underparameterized one. These observations emphasize the critical role of latent representations in OOD detection under different levels of model complexity. While our theoretical analysis relies on simplified settings (e.g., linear models and Gaussian inputs), future works may investigate more complex systems and include additional factors such as training dynamics, or regularization.

**Limitations:** Finally, our study is limited to post-hoc out-of-distribution (OOD) detection; exploring alternative OOD detection paradigms could be a valuable direction for future research.

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

# Double Descent Meets Out-of-Distribution Detection: Theoretical Insights and Empirical Analysis on the Role of Model Complexity

## Table of Contents - Appendix

# A Proof of Theorems

While our theoretical insights are inspired by Theorem 1 in Belkin et al. (2020), we extend the framework to classification and OOD data.

## A.1 Proof for the Expected Risk

Leveraging the Random Matrix Theory and extending the Theorem 1 in Belkin et al. (2020), we derive bounds in Theorem 2 for the expected risk of the subset of classifiers defined in $\hat{\mathcal{F}}_d$ with equation 4. This section is dedicated to the proof of Theorem 2 and follows the same line of arguments of Theorem 1 in Belkin et al. (2020).

**Theorem 2.** *Let $(p, q) \in [d]^2$ such that $p + q = d$, $\mathcal{T} \subseteq [d]$ with $|\mathcal{T}| = p$ an arbitrary subset of the $d$ first natural integers, and $\mathcal{T}^c := [d] \setminus \mathcal{T}$ its complement set. Let $\hat{w} \in R^d(\mathcal{T})$ such that $\hat{w}_{\mathcal{T}} = X_{\mathcal{T}}^+ z \in \mathbb{R}^p$ and $\hat{w}_{\mathcal{T}^c} = \mathbf{0}_q \in \mathbb{R}^q$. Under Assumptions 4.1-4.2, the expected risk with respect to the loss function $\ell : (\hat{y}, y) \mapsto (\hat{y} - y)^2$ of the predictor $\hat{f}_{\mathcal{T}} : x \mapsto \phi(x^T \hat{w})$ satisfies*

$$c \, c(n, p) \leq \mathbb{E}_X \left[ R(\hat{f}_{\mathcal{T}}) \right] \leq C \, c(n, p),$$

*where $c, C > 0$ and*

$$c(n, p) = \begin{cases} \frac{p}{n-p-1}\left(\|w_{\mathcal{T}^c}^*\|_2^2 + \sigma^2\right) + \|w_{\mathcal{T}^c}^*\|_2^2 & \text{if } p \leq n - 2, \\ +\infty & \text{if } n - 1 \leq p \leq n + 1, \quad (8) \\ \left(1 - \frac{n}{p}\right)\|w_{\mathcal{T}}^*\|_2^2 + \frac{n}{p-n-1}\left(\|w_{\mathcal{T}^c}^*\|_2^2 + \sigma^2\right) + \|w_{\mathcal{T}^c}^*\|_2^2 & \text{if } p \geq n + 2. \end{cases}$$

**Remark A.1.** Theorem 1 in Belkin et al. (2020) constitutes a special case of Theorem 2 for $\phi : x \mapsto x$, which corresponds to the case where $\Sigma = I_d$.

**Remark A.2.** From Theorem 2, we have $\mathbb{E}_X \left[ R(\hat{f}_{\mathcal{T}}) \right] = \infty$ around $p = n$. The expected risk decreases again as $p$ increases beyond $n$ and highlights a double descent phenomenon. This result is consistent with the literature of double descent (Mei & Montanari, 2022; Louart et al., 2018; Liao et al., 2020; Bach, 2024), which identifies the ratio $p/n$ as the complexity of a linear model to describe an under- ($p/n < 1$) and an over- ($p/n > 1$) parameterized regimes for the expected risk with a phase transition around $p/n = 1$ characterized by a peak.

*Proof.* Let $x \in \mathcal{X}$ and

$$\Delta_w = \hat{w} - w^*.$$

We have

$$\hat{f}_{\mathcal{T}}(x) = \phi(x^T \hat{w}) = \phi(x^T w^*) + x^T \Delta_w \phi'\left(c(x, \hat{w}, w^*)\right).$$

From the mean-value theorem, there exists

$$c(x, \hat{w}, w^*) \in \left(\min(x^T \hat{w}, x^T w^*), \max(x^T \hat{w}, x^T w^*)\right),$$

such that

$$\phi\left(x^T(w^* + \Delta_w)\right) = \phi(x^T w^*) + x^T \Delta_w \phi'\left(c(x, \hat{w}, w^*)\right).$$

We have thus

$$\begin{aligned}
\mathbb{E}_X \left[ R(\hat{f}_{\mathcal{T}}) \right] = \mathbb{E}_{x, X} \left[ \left(\hat{f}(x) - f^*(x)\right)^2 \right] &= \mathbb{E}_{x, X} \left[ \left(\phi(x^T \hat{w}) - \phi(x^T w^*)\right)^2 \right] \\
&= \mathbb{E}_{x, X} \left[ \left(x^T \Delta_w \phi'\left(c(x, \hat{w}, w^*)\right)\right)^2 \right] \\
&= \mathbb{E}_{x, X} \left[ \Delta_w^T x x^T \Delta_w \phi'\left(c(x, \hat{w}, w^*)\right)^2 \right] \\
&= \text{Tr}\left( \mathbb{E}_{x, X} \left[ \phi'\left(c(x, \hat{w}, w^*)\right)^2 x x^T \Delta_w \Delta_w^T \right] \right) \\
&= \text{Tr}\left( \mathbb{E}_X \left[ \mathbb{E}_x \left[ \phi'\left(c(x, \hat{w}, w^*)\right)^2 x x^T \right] \Delta_w \Delta_w^T \right] \right) \\
&= \text{Tr}\left( \mathbb{E}_X \left[ \Sigma \Delta_w \Delta_w^T \right] \right) \\
&= \mathbb{E}_X \left[ \Delta_w^T \Sigma \Delta_w \right],
\end{aligned}$$

where

$$\boldsymbol{\Sigma} = \mathbb{E}_{\boldsymbol{x}}\big[\phi'\big(c(\boldsymbol{x}, \hat{\boldsymbol{w}}, \boldsymbol{w}^*)\big)^2 \boldsymbol{x}\boldsymbol{x}^T\big]$$
$$= \mathbb{E}_{\boldsymbol{x}}\left[\left(\tfrac{\phi(\boldsymbol{x}^T\hat{\boldsymbol{w}}) - \phi(\boldsymbol{x}^T\boldsymbol{w}^*)}{\boldsymbol{x}^T\hat{\boldsymbol{w}} - \boldsymbol{x}^T\boldsymbol{w}^*}\right)^2 \boldsymbol{x}\boldsymbol{x}^T\right].$$

From the min-max theorem, we have

$$\mathbb{E}_{\boldsymbol{X}}\big[\lambda_{\min}(\boldsymbol{\Sigma})\boldsymbol{\Delta}_{\boldsymbol{w}}^T\boldsymbol{\Delta}_{\boldsymbol{w}}\big] \le \mathbb{E}_{\boldsymbol{X}}\big[\boldsymbol{\Delta}_{\boldsymbol{w}}^T\boldsymbol{\Sigma}\boldsymbol{\Delta}_{\boldsymbol{w}}\big] \le \mathbb{E}_{\boldsymbol{X}}\big[\lambda_{\max}(\boldsymbol{\Sigma})\boldsymbol{\Delta}_{\boldsymbol{w}}^T\boldsymbol{\Delta}_{\boldsymbol{w}}\big].$$

From Lemma 1.1, there exists $c, C > 0$ independent of $\hat{\boldsymbol{w}}$ (and thus $\boldsymbol{X}$) such that

$$c\,\mathbb{E}_{\boldsymbol{X}}\big[\boldsymbol{\Delta}_{\boldsymbol{w}}^T\boldsymbol{\Delta}_{\boldsymbol{w}}\big] \le \mathbb{E}_{\boldsymbol{X}}\big[\boldsymbol{\Delta}_{\boldsymbol{w}}^T\boldsymbol{\Sigma}\boldsymbol{\Delta}_{\boldsymbol{w}}\big] \le C\,\mathbb{E}_{\boldsymbol{X}}\big[\boldsymbol{\Delta}_{\boldsymbol{w}}^T\boldsymbol{\Delta}_{\boldsymbol{w}}\big].$$

For $\mathbb{E}_{\boldsymbol{X}}\big[\boldsymbol{\Delta}_{\boldsymbol{w}}^T\boldsymbol{\Delta}_{\boldsymbol{w}}\big]$, we have

$$\mathbb{E}_{\boldsymbol{X}}\big[\boldsymbol{\Delta}_{\boldsymbol{w}}^T\boldsymbol{\Delta}_{\boldsymbol{w}}\big] = \mathbb{E}_{\boldsymbol{X}}\Big[\|\hat{\boldsymbol{w}} - \boldsymbol{w}^*\|_2^2\Big]$$
$$= \mathbb{E}_{\boldsymbol{X}}\Big[\|\hat{\boldsymbol{w}}_{\mathcal{T}} - \boldsymbol{w}^*_{\mathcal{T}}\|_2^2\Big] + \mathbb{E}_{\boldsymbol{X}}\Big[\|\hat{\boldsymbol{w}}_{\mathcal{T}^c} - \boldsymbol{w}^*_{\mathcal{T}^c}\|_2^2\Big]$$
$$= \mathbb{E}_{\boldsymbol{X}}\Big[\|\hat{\boldsymbol{w}}_{\mathcal{T}} - \boldsymbol{w}^*_{\mathcal{T}}\|_2^2\Big] + \|\boldsymbol{w}^*_{\mathcal{T}^c}\|_2^2.$$

From Lemma 1.2, we have

$$\mathbb{E}_{\boldsymbol{X}}\big[\|\hat{\boldsymbol{w}}_{\mathcal{T}} - \boldsymbol{w}^*_{\mathcal{T}}\|_2^2\big] = \begin{cases} \frac{p}{n-p-1}\Big(\|\boldsymbol{w}^*_{\mathcal{T}^c}\|_2^2 + \sigma^2\Big) & \text{if } p \le n - 2, \\ +\infty & \text{if } n - 1 \le p \le n + 1, \\ \big(1 - \frac{n}{p}\big)\|\boldsymbol{w}^*_{\mathcal{T}}\|_2^2 + \frac{n}{p-n-1}\Big(\|\boldsymbol{w}^*_{\mathcal{T}^c}\|_2^2 + \sigma^2\Big) & \text{if } p \ge n + 2, \end{cases}$$

which concludes the proof. $\qquad\qquad\square$

## A.2  Proof for the Expected OOD Risk (Theorem 1 )

This section is dedicated to the proof of Theorem 1.

**Theorem 1.** *Let $(p, q) \in [d]^2$ such that $p + q = d$, $\mathcal{T} \subseteq [d]$ with $|\mathcal{T}| = p$ an arbitrary subset of the $d$ first natural integers, and $\mathcal{T}^c := [d] \setminus \mathcal{T}$ its complement set. Let $\hat{\boldsymbol{w}} \in R^d(\mathcal{T})$ such that $\hat{\boldsymbol{w}}_{\mathcal{T}} = \boldsymbol{X}_{\mathcal{T}}^+\boldsymbol{z} \in \mathbb{R}^p$ and $\hat{\boldsymbol{w}}_{\mathcal{T}^c} = \boldsymbol{0}_q \in \mathbb{R}^q$. Under Assumptions 4.1-4.2, the expected OOD risk on the predictor $\hat{f}_{\mathcal{T}} : \boldsymbol{x} \mapsto \phi(\boldsymbol{x}^T\hat{\boldsymbol{w}})$ satisfies*

$$c\,c(n, p) \le \mathbb{E}_{\boldsymbol{X}}\big[R_{OOD}(\hat{f})\big] \le C\,c(n, p),$$

*where $c, C > 0$ and*

$$c(n, p) = \begin{cases} \frac{p}{n-p-1}\Big(\|\boldsymbol{w}_{\mathcal{T}^c}^{OOD}\|_2^2 + \sigma^2\Big) + \|\boldsymbol{w}_{\mathcal{T}^c}^{OOD}\|_2^2 & \text{if } p \le n - 2, \\ +\infty & \text{if } n - 1 \le p \le n + 1, \\ \big(1 - \frac{n}{p}\big)\|\boldsymbol{w}_{\mathcal{T}}^{OOD}\|_2^2 + \frac{n}{p-n-1}\Big(\|\boldsymbol{w}_{\mathcal{T}^c}^{OOD}\|_2^2 + \sigma^2\Big) + \|\boldsymbol{w}_{\mathcal{T}^c}^{OOD}\|_2^2 & \text{if } p \ge n + 2. \end{cases}$$

*Proof.* The layout of the proof is similar to the proof of Theorem 2. Let $\boldsymbol{x} \in \mathcal{X}$ and

$$\boldsymbol{\Delta}_{\boldsymbol{w}} = \hat{\boldsymbol{w}} - \boldsymbol{w}^{\text{OOD}}.$$

From the mean-value theorem, there exists

$$c(\boldsymbol{x}, \hat{\boldsymbol{w}}, \boldsymbol{w}^{\text{OOD}}) \in \big(\min(\boldsymbol{x}^T\hat{\boldsymbol{w}}, \boldsymbol{x}^T\boldsymbol{w}^{\text{OOD}}), \max(\boldsymbol{x}^T\hat{\boldsymbol{w}}, \boldsymbol{x}^T\boldsymbol{w}^{\text{OOD}})\big),$$

such that

$$\hat{f}_{\mathcal{T}}(\boldsymbol{x}) = \phi(\boldsymbol{x}^T\hat{\boldsymbol{w}}) = \phi\big(\boldsymbol{x}^T(\boldsymbol{w}^{\text{OOD}} + \boldsymbol{\Delta}_{\boldsymbol{w}})\big)$$
$$= \phi(\boldsymbol{x}^T\boldsymbol{w}^{\text{OOD}}) + \boldsymbol{x}^T\boldsymbol{\Delta}_{\boldsymbol{w}}\phi'\big(c(\boldsymbol{x}, \hat{\boldsymbol{w}}, \boldsymbol{w}^{\text{OOD}})\big).$$

We have

$$\mathbb{E}_{\boldsymbol{X}}\big[R_{\text{OOD}}(\hat{f})\big] = \mathbb{E}_{(\boldsymbol{x},\cdot)\sim P_{\mathcal{X},\mathcal{Y}},\boldsymbol{X}}\Big[\big(\hat{f}(\boldsymbol{x}) - f^{\text{OOD}}(\boldsymbol{x})\big)^2\Big] + \mathbb{E}_{(\boldsymbol{x},\cdot)\sim P_{\mathcal{X},\mathcal{Y}}^{\text{OOD}},\boldsymbol{X}}\Big[\big(\hat{f}(\boldsymbol{x}) - f^{\text{OOD}}(\boldsymbol{x})\big)^2\Big]$$

$$= \mathbb{E}_{\boldsymbol{x}\sim P_{\mathcal{X},\mathcal{Y}},\boldsymbol{X}}\Big[\big(\phi(\boldsymbol{x}^T\hat{\boldsymbol{w}}) - \phi(\boldsymbol{x}^T\boldsymbol{w}^{\text{OOD}})\big)^2\Big] + \mathbb{E}_{\boldsymbol{x}\sim P_{\mathcal{X},\mathcal{Y}}^{\text{OOD}},\boldsymbol{X}}\Big[\big(\phi(\boldsymbol{x}^T\hat{\boldsymbol{w}}) - \phi(\boldsymbol{x}^T\boldsymbol{w}^{\text{OOD}})\big)^2\Big]$$

$$= \mathbb{E}_{\boldsymbol{x}\sim P_{\mathcal{X},\mathcal{Y}},\boldsymbol{X}}\Big[\big(\boldsymbol{x}^T\boldsymbol{\Delta}_{\boldsymbol{w}}\phi'\big(c(\boldsymbol{x},\hat{\boldsymbol{w}},\boldsymbol{w}^{\text{OOD}})\big)\big)^2\Big] + \mathbb{E}_{\boldsymbol{x}\sim P_{\mathcal{X},\mathcal{Y}}^{\text{OOD}},\boldsymbol{X}}\Big[\big(\boldsymbol{x}^T\boldsymbol{\Delta}_{\boldsymbol{w}}\phi'\big(c(\boldsymbol{x},\hat{\boldsymbol{w}},\boldsymbol{w}^{\text{OOD}})\big)\big)^2\Big]$$

$$= \mathbb{E}_{\boldsymbol{x}\sim P_{\mathcal{X},\mathcal{Y}},\boldsymbol{X}}\Big[\boldsymbol{\Delta}_{\boldsymbol{w}}^T\boldsymbol{x}\boldsymbol{x}^T\boldsymbol{\Delta}_{\boldsymbol{w}}\phi'\big(c(\boldsymbol{x},\hat{\boldsymbol{w}},\boldsymbol{w}^{\text{OOD}})\big)^2\Big] + \mathbb{E}_{\boldsymbol{x}\sim P_{\mathcal{X},\mathcal{Y}}^{\text{OOD}},\boldsymbol{X}}\Big[\boldsymbol{\Delta}_{\boldsymbol{w}}^T\boldsymbol{x}\boldsymbol{x}^T\boldsymbol{\Delta}_{\boldsymbol{w}}\phi'\big(c(\boldsymbol{x},\hat{\boldsymbol{w}},\boldsymbol{w}^{\text{OOD}})\big)^2\Big]$$

$$= \text{Tr}\Big(\mathbb{E}_{\boldsymbol{X}}\Big[\big[\mathbb{E}_{\boldsymbol{x}\sim P_{\mathcal{X},\mathcal{Y}}}\big[\phi'\big(c(\boldsymbol{x},\hat{\boldsymbol{w}},\boldsymbol{w}^{\text{OOD}})\big)^2\boldsymbol{x}\boldsymbol{x}^T\big] + \mathbb{E}_{\boldsymbol{x}\sim P_{\mathcal{X},\mathcal{Y}}^{\text{OOD}}}\big[\phi'\big(c(\boldsymbol{x},\hat{\boldsymbol{w}},\boldsymbol{w}^{\text{OOD}})\big)^2\boldsymbol{x}\boldsymbol{x}^T\big]\big]\boldsymbol{\Delta}_{\boldsymbol{w}}\boldsymbol{\Delta}_{\boldsymbol{w}}^T\Big]\Big)$$

$$= \text{Tr}\Big(\mathbb{E}_{\boldsymbol{X}}\Big[\big[\boldsymbol{\Sigma} + \boldsymbol{\Sigma}^{\text{OOD}}\big]\boldsymbol{\Delta}_{\boldsymbol{w}}\boldsymbol{\Delta}_{\boldsymbol{w}}^T\Big]\Big)$$

$$= \mathbb{E}_{\boldsymbol{X}}\big[\boldsymbol{\Delta}_{\boldsymbol{w}}^T\boldsymbol{\Sigma}\boldsymbol{\Delta}_{\boldsymbol{w}}\big] + \mathbb{E}_{\boldsymbol{X}}\big[\boldsymbol{\Delta}_{\boldsymbol{w}}^T\boldsymbol{\Sigma}^{\text{OOD}}\boldsymbol{\Delta}_{\boldsymbol{w}}\big],$$

where

$$\boldsymbol{\Sigma} = \mathbb{E}_{\boldsymbol{x}\sim P_{\mathcal{X},\mathcal{Y}}}\big[\phi'\big(c(\boldsymbol{x},\hat{\boldsymbol{w}},\boldsymbol{w}^{\text{OOD}})\big)^2\boldsymbol{x}\boldsymbol{x}^T\big]$$

$$= \mathbb{E}_{\boldsymbol{x}\sim P_{\mathcal{X},\mathcal{Y}}}\left[\left(\frac{\phi(\boldsymbol{x}^T\hat{\boldsymbol{w}})-\phi(\boldsymbol{x}^T\boldsymbol{w}^{\text{OOD}})}{\boldsymbol{x}^T\hat{\boldsymbol{w}}-\boldsymbol{x}^T\boldsymbol{w}^{\text{OOD}}}\right)^2\boldsymbol{x}\boldsymbol{x}^T\right]$$

and

$$\boldsymbol{\Sigma}^{\text{OOD}} = \mathbb{E}_{\boldsymbol{x}\sim P_{\mathcal{X},\mathcal{Y}}^{\text{OOD}}}\big[\phi'\big(c(\boldsymbol{x},\hat{\boldsymbol{w}},\boldsymbol{w}^{\text{OOD}})\big)^2\boldsymbol{x}\boldsymbol{x}^T\big]$$

$$= \mathbb{E}_{\boldsymbol{x}\sim P_{\mathcal{X},\mathcal{Y}}^{\text{OOD}}}\left[\left(\frac{\phi(\boldsymbol{x}^T\hat{\boldsymbol{w}})-\phi(\boldsymbol{x}^T\boldsymbol{w}^{\text{OOD}})}{\boldsymbol{x}^T\hat{\boldsymbol{w}}-\boldsymbol{x}^T\boldsymbol{w}^{\text{OOD}}}\right)^2\boldsymbol{x}\boldsymbol{x}^T\right].$$

From the min-max theorem, we have

$$\mathbb{E}_{\boldsymbol{X}}\big[\boldsymbol{\Delta}_{\boldsymbol{w}}^T\boldsymbol{\Sigma}\boldsymbol{\Delta}_{\boldsymbol{w}}\big] + \mathbb{E}_{\boldsymbol{X}}\big[\boldsymbol{\Delta}_{\boldsymbol{w}}^T\boldsymbol{\Sigma}^{\text{OOD}}\boldsymbol{\Delta}_{\boldsymbol{w}}\big] \geq \mathbb{E}_{\boldsymbol{X}}\Big[\big(\lambda_{\min}(\boldsymbol{\Sigma}) + \lambda_{\min}(\boldsymbol{\Sigma}^{\text{OOD}})\big)\boldsymbol{\Delta}_{\boldsymbol{w}}^T\boldsymbol{\Delta}_{\boldsymbol{w}}\Big]$$

and

$$\mathbb{E}_{\boldsymbol{X}}\big[\boldsymbol{\Delta}_{\boldsymbol{w}}^T\boldsymbol{\Sigma}\boldsymbol{\Delta}_{\boldsymbol{w}}\big] + \mathbb{E}_{\boldsymbol{X}}\big[\boldsymbol{\Delta}_{\boldsymbol{w}}^T\boldsymbol{\Sigma}^{\text{OOD}}\boldsymbol{\Delta}_{\boldsymbol{w}}\big] \leq \mathbb{E}_{\boldsymbol{X}}\Big[\big(\lambda_{\max}(\boldsymbol{\Sigma}) + \lambda_{\max}(\boldsymbol{\Sigma}^{\text{OOD}})\big)\boldsymbol{\Delta}_{\boldsymbol{w}}^T\boldsymbol{\Delta}_{\boldsymbol{w}}\Big].$$

From Lemma 1.1, there exists $c, C > 0$ independent of $\hat{\boldsymbol{w}}$ (and thus $\boldsymbol{X}$) such that

$$c\,\mathbb{E}_{\boldsymbol{X}}\big[\boldsymbol{\Delta}_{\boldsymbol{w}}^T\boldsymbol{\Delta}_{\boldsymbol{w}}\big] \leq \mathbb{E}_{\boldsymbol{X}}\big[\boldsymbol{\Delta}_{\boldsymbol{w}}^T\boldsymbol{\Sigma}\boldsymbol{\Delta}_{\boldsymbol{w}}\big] + \mathbb{E}_{\boldsymbol{X}}\big[\boldsymbol{\Delta}_{\boldsymbol{w}}^T\boldsymbol{\Sigma}^{\text{OOD}}\boldsymbol{\Delta}_{\boldsymbol{w}}\big] \leq C\,\mathbb{E}_{\boldsymbol{X}}\big[\boldsymbol{\Delta}_{\boldsymbol{w}}^T\boldsymbol{\Delta}_{\boldsymbol{w}}\big].$$

For $\mathbb{E}_{\boldsymbol{X}}\big[\boldsymbol{\Delta}_{\boldsymbol{w}}^T\boldsymbol{\Delta}_{\boldsymbol{w}}\big]$, we have

$$\mathbb{E}_{\boldsymbol{X}}\big[\boldsymbol{\Delta}_{\boldsymbol{w}}^T\boldsymbol{\Delta}_{\boldsymbol{w}}\big] = \mathbb{E}_{\boldsymbol{X}}\Big[\|\hat{\boldsymbol{w}} - \boldsymbol{w}^{\text{OOD}}\|_2^2\Big]$$

$$= \mathbb{E}_{\boldsymbol{X}}\Big[\|\hat{\boldsymbol{w}}_{\mathcal{T}} - \boldsymbol{w}_{\mathcal{T}}^{\text{OOD}}\|_2^2\Big] + \mathbb{E}_{\boldsymbol{X}}\Big[\|\hat{\boldsymbol{w}}_{\mathcal{T}^c} - \boldsymbol{w}_{\mathcal{T}^c}^{\text{OOD}}\|_2^2\Big]$$

$$= \mathbb{E}_{\boldsymbol{X}}\Big[\|\hat{\boldsymbol{w}}_{\mathcal{T}} - \boldsymbol{w}_{\mathcal{T}}^{\text{OOD}}\|_2^2\Big] + \|\boldsymbol{w}_{\mathcal{T}^c}^{\text{OOD}}\|_2^2.$$

From Lemma 1.2, we have

$$\mathbb{E}_{\boldsymbol{X}}\big[\|\hat{\boldsymbol{w}}_{\mathcal{T}} - \boldsymbol{w}_{\mathcal{T}}^{\text{OOD}}\|_2^2\big] = \begin{cases} \frac{p}{n-p-1}\big(\|\boldsymbol{w}_{\mathcal{T}^c}^{\text{OOD}}\|_2^2 + \sigma^2\big) & \text{if } p \leq n-2, \\ +\infty & \text{if } n-1 \leq p \leq n+1, \\ \big(1 - \frac{n}{p}\big)\|\boldsymbol{w}_{\mathcal{T}}^{\text{OOD}}\|_2^2 + \frac{n}{p-n-1}\big(\|\boldsymbol{w}_{\mathcal{T}^c}^{\text{OOD}}\|_2^2 + \sigma^2\big) & \text{if } p \geq n+2, \end{cases}$$

which concludes the proof. $\qquad\square$

**Lemma 1.1.** Let $\boldsymbol{\Sigma}, \boldsymbol{\Sigma}^{\text{OOD}} \in \mathbb{R}^{d\times d}$ defined as

$$\boldsymbol{\Sigma} = \mathbb{E}_{\boldsymbol{x}\sim P_{\mathcal{X},\mathcal{Y}}}\left[\left(\frac{\phi(\boldsymbol{x}^T\hat{\boldsymbol{w}})-\phi(\boldsymbol{x}^T\boldsymbol{w}^{\text{OOD}})}{\boldsymbol{x}^T\hat{\boldsymbol{w}}-\boldsymbol{x}^T\boldsymbol{w}^{\text{OOD}}}\right)^2\boldsymbol{x}\boldsymbol{x}^T\right]$$

and

$$\boldsymbol{\Sigma}^{\mathrm{OOD}} = \mathbb{E}_{\boldsymbol{x} \sim P_{\mathcal{X},\mathcal{Y}}^{\mathrm{OOD}}}\left[\left(\frac{\phi(\boldsymbol{x}^T\hat{\boldsymbol{w}})-\phi(\boldsymbol{x}^T\boldsymbol{w}^{\mathrm{OOD}})}{\boldsymbol{x}^T\hat{\boldsymbol{w}}-\boldsymbol{x}^T\boldsymbol{w}^{\mathrm{OOD}}}\right)^2 \boldsymbol{x}\boldsymbol{x}^T\right].$$

Under Assumption 4.1-4.2, the matrices $\boldsymbol{\Sigma}$ and $\boldsymbol{\Sigma}^{\mathrm{OOD}}$ are positive-definite. Furthermore, there exists $c_1, C_1 > 0$ and $c_2, C_2 > 0$ independent of $\hat{\boldsymbol{w}}$ such that

$$\lambda_{\min}(\boldsymbol{\Sigma}) \geq c_1 \quad \text{and} \quad \lambda_{\max}(\boldsymbol{\Sigma}) \leq C_1$$

and

$$\lambda_{\min}(\boldsymbol{\Sigma}^{\mathrm{OOD}}) \geq c_2 \quad \text{and} \quad \lambda_{\max}(\boldsymbol{\Sigma}^{\mathrm{OOD}}) \leq C_2.$$

*Proof.* From the mean-value theorem, there exists

$$c(\boldsymbol{x}, \hat{\boldsymbol{w}}, \boldsymbol{w}^{\mathrm{OOD}}) \in \left(\min(\boldsymbol{x}^T\hat{\boldsymbol{w}}, \boldsymbol{x}^T\boldsymbol{w}^{\mathrm{OOD}}), \max(\boldsymbol{x}^T\hat{\boldsymbol{w}}, \boldsymbol{x}^T\boldsymbol{w}^{\mathrm{OOD}})\right),$$

such that

$$\phi'\left(c(\boldsymbol{x}, \hat{\boldsymbol{w}}, \boldsymbol{w}^{\mathrm{OOD}})\right) = \frac{\phi(\boldsymbol{x}^T\hat{\boldsymbol{w}})-\phi(\boldsymbol{x}^T\boldsymbol{w}^{\mathrm{OOD}})}{\boldsymbol{x}^T\hat{\boldsymbol{w}}-\boldsymbol{x}^T\boldsymbol{w}^{\mathrm{OOD}}}.$$

Using $c(\boldsymbol{x}, \hat{\boldsymbol{w}}, \boldsymbol{w}^{\mathrm{OOD}})$, we rewrite the matrix $\boldsymbol{\Sigma}$ as

$$\boldsymbol{\Sigma} = \mathbb{E}_{(\boldsymbol{x},\cdot)\sim P_{\mathcal{X},\mathcal{Y}}}\left[\phi'\left(c(\boldsymbol{x}, \hat{\boldsymbol{w}}, \boldsymbol{w}^{\mathrm{OOD}})\right)^2 \boldsymbol{x}\boldsymbol{x}^T\right].$$

The matrix $\boldsymbol{\Sigma}$ is semi-positive-definite. Let $\mu(\cdot)$ denotes the probability density function of $P_{\mathcal{X},\mathcal{Y}}$. Let $\boldsymbol{u} = \arg\min_{\boldsymbol{a}\in\mathbb{R}^d\setminus\{\boldsymbol{0}_d\}} \frac{\boldsymbol{a}^T\boldsymbol{\Sigma}\boldsymbol{a}}{\boldsymbol{a}^T\boldsymbol{a}}$. From the min-max theorem, we have

$$\lambda_{\min}(\boldsymbol{\Sigma}) = \frac{\boldsymbol{u}^T\boldsymbol{\Sigma}\boldsymbol{u}}{\boldsymbol{u}^T\boldsymbol{u}}.$$

Since the derivative of $\phi(\cdot)$ is strictly positive, we have

$$\lambda_{\min}(\boldsymbol{\Sigma}) = \frac{\boldsymbol{u}^T\boldsymbol{\Sigma}\boldsymbol{u}}{\boldsymbol{u}^T\boldsymbol{u}} = \frac{1}{\boldsymbol{u}^T\boldsymbol{u}}\boldsymbol{u}^T\mathbb{E}_{(\boldsymbol{x},\cdot)\sim P_{\mathcal{X},\mathcal{Y}}}\left[\phi'\left(c(\boldsymbol{x}, \hat{\boldsymbol{w}}, \boldsymbol{w}^{\mathrm{OOD}})\right)^2 \boldsymbol{x}\boldsymbol{x}^T\right]\boldsymbol{u}$$

$$= \frac{1}{\boldsymbol{u}^T\boldsymbol{u}}\int_{\mathbb{R}^d}\phi'\left(c(\boldsymbol{x}, \hat{\boldsymbol{w}}, \boldsymbol{w}^{\mathrm{OOD}})\right)^2 (\boldsymbol{u}^T\boldsymbol{x})^2\mu(\boldsymbol{x})d\boldsymbol{x}$$

$$\geq \underbrace{\frac{1}{\boldsymbol{u}^T\boldsymbol{u}}\int_{R^d(\mathcal{T}^c)}\phi'\left(c(\boldsymbol{x}, \hat{\boldsymbol{w}}, \boldsymbol{w}^{\mathrm{OOD}})\right)^2 (\boldsymbol{u}^T\boldsymbol{x})^2\mu(\boldsymbol{x})d\boldsymbol{x}}_{=c}.$$

Note that for all $\boldsymbol{x} \in R^d(\mathcal{T}^c)$, we have

$$\hat{\boldsymbol{w}}^T\boldsymbol{x} = \hat{\boldsymbol{w}}_{\mathcal{T}}^T\boldsymbol{x}_{\mathcal{T}} + \hat{\boldsymbol{w}}_{\mathcal{T}^c}^T\boldsymbol{x}_{\mathcal{T}^c} = \hat{\boldsymbol{w}}_{\mathcal{T}}^T\boldsymbol{0}_p + \boldsymbol{0}_{d-p}^T\boldsymbol{x}_{\mathcal{T}^c} = 0.$$

We deduce that $c$ is independent of $\hat{\boldsymbol{w}}$ since

$$c(\boldsymbol{x}, \hat{\boldsymbol{w}}, \boldsymbol{w}^{\mathrm{OOD}}) \in \left(\min(0, \boldsymbol{x}^T\boldsymbol{w}^{\mathrm{OOD}}), \max(0, \boldsymbol{x}^T\boldsymbol{w}^{\mathrm{OOD}})\right), \quad \forall \boldsymbol{x} \in R^d(\mathcal{T}^c).$$

Furthermore, $c > 0$ since $\boldsymbol{x} \mapsto \phi'\left(c(\boldsymbol{x}, \hat{\boldsymbol{w}}, \boldsymbol{w}^{\mathrm{OOD}})\right)^2 (\boldsymbol{u}^T\boldsymbol{x})^2$ is nonnegative and is not the zero map on $R^d(\mathcal{T})$ as $\boldsymbol{u} \notin R^d(\mathcal{T})$ (Assumption 4.2). We conclude there exists $c > 0$ independant of $\hat{\boldsymbol{w}}$ such that $\lambda_{\min}(\boldsymbol{\Sigma}) \geq c$.

Next, we are going to show that there exists $C > 0$ independant of $\hat{\boldsymbol{w}}$ such that $\lambda_{\max}(\boldsymbol{\Sigma}) \leq C$. From Assumption 4.1, we know there exists $K' > 0$ such that for all $x \in \mathbb{R}$ we have

$$\phi'(x) \leq K'.$$

Let $K > K'$ and $\boldsymbol{M} = K^2\mathbb{E}_{(\boldsymbol{x},\cdot)\sim P_{\mathcal{X},\mathcal{Y}}}\left[\boldsymbol{x}\boldsymbol{x}^T\right] - \boldsymbol{\Sigma}$. $\boldsymbol{M}$ is a symmetric matrix. From the min-max theorem $\boldsymbol{M}$ is semi positive-definite if for all $\boldsymbol{a} \in \mathbb{R}^d\setminus\{\boldsymbol{0}_d\}$, we have

$$\boldsymbol{a}^T\boldsymbol{M}\boldsymbol{a} \geq 0.$$

Let $\boldsymbol{a} \in \mathbb{R}^d \backslash \{\mathbf{0}_d\}$, we have

$$
\begin{aligned}
\boldsymbol{a}^T \boldsymbol{M} \boldsymbol{a} &= \boldsymbol{a}^T \big[ K^2 \mathbb{E}_{(\boldsymbol{x},\cdot) \sim P_{\mathcal{X},\mathcal{Y}}} [\boldsymbol{x}\boldsymbol{x}^T] - \boldsymbol{\Sigma} \big] \boldsymbol{a} \\
&= \boldsymbol{a}^T \Big[ \mathbb{E}_{(\boldsymbol{x},\cdot) \sim P_{\mathcal{X},\mathcal{Y}}} [K^2 \boldsymbol{x}\boldsymbol{x}^T] - \mathbb{E}_{(\boldsymbol{x},\cdot) \sim P_{\mathcal{X},\mathcal{Y}}} \big[ \phi'\big(c(\boldsymbol{x}, \hat{\boldsymbol{w}}, \boldsymbol{w}^{\mathrm{OOD}})\big)^2 \boldsymbol{x}\boldsymbol{x}^T \big] \Big] \boldsymbol{a} \\
&= \mathbb{E}_{(\boldsymbol{x},\cdot) \sim P_{\mathcal{X},\mathcal{Y}}} \Big[ \underbrace{\big( K^2 - \phi'\big(c(\boldsymbol{x}, \hat{\boldsymbol{w}}, \boldsymbol{w}^{\mathrm{OOD}})\big)^2 \big)}_{>0} \underbrace{(\boldsymbol{a}^T \boldsymbol{x})^2}_{\geq 0} \Big] \\
&\geq 0.
\end{aligned}
$$

We deduce that $\boldsymbol{M} \succeq 0$. Therefore, $\boldsymbol{\Sigma} \preceq K^2 \mathbb{E}_{(\boldsymbol{x},\cdot) \sim P_{\mathcal{X},\mathcal{Y}}} [\boldsymbol{x}\boldsymbol{x}^T]$ and we have

$$
\lambda_{\max}(\boldsymbol{\Sigma}) \leq \underbrace{K^2 \lambda_{\max}\big( \mathbb{E}_{(\boldsymbol{x},\cdot) \sim P_{\mathcal{X},\mathcal{Y}}} [\boldsymbol{x}\boldsymbol{x}^T] \big)}_{=C}.
$$

C is independent of $\hat{\boldsymbol{w}}$. With a similar proof, we can derive a result for $\boldsymbol{\Sigma}^{\mathrm{OOD}}$. $\qquad \square$

**Lemma 1.2.** Let $(p, q) \in [d]^2$ such that $p + q = d$, $\mathcal{T} \subseteq [d]$ an arbitrary subset of the $d$ first natural integers, and $\mathcal{T}^c := [d] \backslash \mathcal{T}$ its complement set. Let $\hat{\boldsymbol{w}} \in \mathbb{R}^d$ such that $\hat{\boldsymbol{w}}_{\mathcal{T}} = \boldsymbol{X}_{\mathcal{T}}^+ \boldsymbol{z} \in \mathbb{R}^p$ and $\hat{\boldsymbol{w}}_{\mathcal{T}^c} = \mathbf{0}_q \in \mathbb{R}^q$. We have

$$
\mathbb{E}_{\boldsymbol{X}} \big[ \|\hat{\boldsymbol{w}}_{\mathcal{T}} - \boldsymbol{w}_{\mathcal{T}}^{\mathrm{OOD}}\|_2^2 \big] = \begin{cases} \frac{p}{n-p-1} \big( \|\boldsymbol{w}_{\mathcal{T}^c}^{\mathrm{OOD}}\|_2^2 + \sigma^2 \big) & \text{if } p \leq n - 2, \\ +\infty & \text{if } n - 1 \leq p \leq n + 1, \\ \big( 1 - \frac{n}{p} \big) \|\boldsymbol{w}_{\mathcal{T}}^{\mathrm{OOD}}\|_2^2 + \frac{n}{p-n-1} \big( \|\boldsymbol{w}_{\mathcal{T}^c}^{\mathrm{OOD}}\|_2^2 + \sigma^2 \big) & \text{if } p \geq n + 2. \end{cases}
$$

*Proof.* Let $\boldsymbol{\eta} = \boldsymbol{z} - \boldsymbol{X}_{\mathcal{T}} \boldsymbol{w}_{\mathcal{T}}^{\mathrm{OOD}}$. Since $\hat{\boldsymbol{w}}_{\mathcal{T}} = \boldsymbol{X}_{\mathcal{T}}^+ \boldsymbol{z}$, we have $\hat{\boldsymbol{w}}_{\mathcal{T}} = \boldsymbol{X}_{\mathcal{T}}^+ (\boldsymbol{\eta} + \boldsymbol{X}_{\mathcal{T}} \boldsymbol{w}_{\mathcal{T}}^{\mathrm{OOD}})$. Therefore,

$$
\begin{aligned}
\mathbb{E}_{\boldsymbol{X}} \big[ \|\boldsymbol{w}_{\mathcal{T}}^{\mathrm{OOD}} - \hat{\boldsymbol{w}}_{\mathcal{T}}\|_2^2 \big] &= \mathbb{E}_{\boldsymbol{X}} \Big[ \big\| (\boldsymbol{I}_p - \boldsymbol{X}_{\mathcal{T}}^+ \boldsymbol{X}_{\mathcal{T}}) \boldsymbol{w}_{\mathcal{T}}^{\mathrm{OOD}} - \boldsymbol{X}_{\mathcal{T}}^+ \boldsymbol{\eta} \big\|_2^2 \Big] \\
&= \mathbb{E}_{\boldsymbol{X}} \Big[ \big\| (\boldsymbol{I}_p - \boldsymbol{X}_{\mathcal{T}}^+ \boldsymbol{X}_{\mathcal{T}}) \boldsymbol{w}_{\mathcal{T}}^{\mathrm{OOD}} \big\|_2^2 + \big\| \boldsymbol{X}_{\mathcal{T}}^+ \boldsymbol{\eta} \big\|_2^2 - 2 \big\langle \big( \boldsymbol{I}_p - \boldsymbol{X}_{\mathcal{T}}^+ \boldsymbol{X}_{\mathcal{T}} \big) \boldsymbol{w}_{\mathcal{T}}^{\mathrm{OOD}}, \boldsymbol{X}_{\mathcal{T}}^+ \boldsymbol{\eta} \big\rangle \Big].
\end{aligned}
$$

Since $(\boldsymbol{X}_{\mathcal{T}}^+ \boldsymbol{X}_{\mathcal{T}})^T = \boldsymbol{X}_{\mathcal{T}}^+ \boldsymbol{X}_{\mathcal{T}}$ and $(\boldsymbol{X}_{\mathcal{T}}^+ \boldsymbol{X}_{\mathcal{T}})^T \boldsymbol{X}_{\mathcal{T}}^+ = \boldsymbol{X}_{\mathcal{T}}^+$, we have

$$
\begin{aligned}
\big\langle (\boldsymbol{I}_p - \boldsymbol{X}_{\mathcal{T}}^+ \boldsymbol{X}_{\mathcal{T}}) \boldsymbol{w}_{\mathcal{T}}^{\mathrm{OOD}}, \boldsymbol{X}_{\mathcal{T}}^+ \boldsymbol{\eta} \big\rangle_2 &= \big( (\boldsymbol{I}_p - \boldsymbol{X}_{\mathcal{T}}^+ \boldsymbol{X}_{\mathcal{T}}) \boldsymbol{w}_{\mathcal{T}}^{\mathrm{OOD}} \big)^T \boldsymbol{X}_{\mathcal{T}}^+ \boldsymbol{\eta} \\
&= [\boldsymbol{w}_{\mathcal{T}}^{\mathrm{OOD}}]^T (\boldsymbol{I}_p - \boldsymbol{X}_{\mathcal{T}}^+ \boldsymbol{X}_{\mathcal{T}})^T \boldsymbol{X}_{\mathcal{T}}^+ \boldsymbol{\eta} \\
&= [\boldsymbol{w}_{\mathcal{T}}^{\mathrm{OOD}}]^T \boldsymbol{X}_{\mathcal{T}}^+ \boldsymbol{\eta} - [\boldsymbol{w}_{\mathcal{T}}^{\mathrm{OOD}}]^T (\boldsymbol{X}_{\mathcal{T}}^+ \boldsymbol{X}_{\mathcal{T}}) \boldsymbol{X}_{\mathcal{T}}^+ \boldsymbol{\eta} \\
&= [\boldsymbol{w}_{\mathcal{T}}^{\mathrm{OOD}}]^T \boldsymbol{X}_{\mathcal{T}}^+ \boldsymbol{\eta} - [\boldsymbol{w}_{\mathcal{T}}^{\mathrm{OOD}}]^T \boldsymbol{X}_{\mathcal{T}}^+ \boldsymbol{\eta} \\
&= 0.
\end{aligned}
$$

$(\boldsymbol{I}_p - \boldsymbol{X}_{\mathcal{T}}^+ \boldsymbol{X}_{\mathcal{T}}) \boldsymbol{w}_{\mathcal{T}}^{\mathrm{OOD}}$ and $\boldsymbol{X}_{\mathcal{T}}^+ \boldsymbol{\eta}$ are thus orthogonal. We deduce that

$$
\begin{aligned}
\mathbb{E}_{\boldsymbol{X}} \Big[ \big\| \hat{\boldsymbol{w}}_{\mathcal{T}} - \boldsymbol{w}_{\mathcal{T}}^{\mathrm{OOD}} \big\|_2^2 \Big] &= \mathbb{E}_{\boldsymbol{X}} \Big[ \big\| (\boldsymbol{I}_p - \boldsymbol{X}_{\mathcal{T}}^+ \boldsymbol{X}_{\mathcal{T}}) \boldsymbol{w}_{\mathcal{T}}^{\mathrm{OOD}} \big\|_2^2 + \big\| \boldsymbol{X}_{\mathcal{T}}^+ \boldsymbol{\eta} \big\|_2^2 \Big] \\
&= \mathbb{E}_{\boldsymbol{X}} \Big[ \big\| (\boldsymbol{I}_p - \boldsymbol{X}_{\mathcal{T}}^+ \boldsymbol{X}_{\mathcal{T}}) \boldsymbol{w}_{\mathcal{T}}^{\mathrm{OOD}} \big\|_2^2 \Big] + \mathbb{E}_{\boldsymbol{X}} \Big[ \big\| \boldsymbol{X}_{\mathcal{T}}^+ \boldsymbol{\eta} \big\|_2^2 \Big].
\end{aligned} \tag{9}
$$

Leveraging the same arguments used by Belkin et al. (2020) to prove the existence of the double descent phenomenon in the regression problem, we distinguish two cases depending on $n$ and $p$ to derive equation 9.

**Classical Regime** ($p < n$). Breiman & Freedman (1983) studied this regime for the regression problem and found:

$$
\mathbb{E}_{\boldsymbol{X}} \Big[ \big\| \hat{\boldsymbol{w}}_{\mathcal{T}} - \boldsymbol{w}_{\mathcal{T}}^{\mathrm{OOD}} \big\|_2^2 \Big] = \frac{p}{n-p-1} \big( \|\boldsymbol{w}_{\mathcal{T}^c}^{\mathrm{OOD}}\|_2^2 + \sigma^2 \big).
$$

**Interpolating Regime ($p \geq n$).** **The interpolating regime** has been considered in Belkin et al. (2020) for the regression problem. We can observe that:

$$\boldsymbol{w}_{\mathcal{T}}^{\mathrm{OOD}} = (\boldsymbol{I}_p - \boldsymbol{X}_{\mathcal{T}}^+ \boldsymbol{X}_{\mathcal{T}}) \boldsymbol{w}_{\mathcal{T}}^{\mathrm{OOD}} + \boldsymbol{X}_{\mathcal{T}}^+ \boldsymbol{X}_{\mathcal{T}} \boldsymbol{w}_{\mathcal{T}}^{\mathrm{OOD}}$$

and

$$\begin{aligned}
\langle (\boldsymbol{I}_p - \boldsymbol{X}_{\mathcal{T}}^+ \boldsymbol{X}_{\mathcal{T}}) \boldsymbol{w}_{\mathcal{T}}^{\mathrm{OOD}}, \boldsymbol{X}_{\mathcal{T}}^+ \boldsymbol{X}_{\mathcal{T}} \boldsymbol{w}_{\mathcal{T}}^{\mathrm{OOD}} \rangle &= (\boldsymbol{I}_p - \boldsymbol{X}_{\mathcal{T}}^+ \boldsymbol{X}_{\mathcal{T}}) \boldsymbol{w}_{\mathcal{T}}^{\mathrm{OOD}})^T (\boldsymbol{X}_{\mathcal{T}}^+ \boldsymbol{X}_{\mathcal{T}} \boldsymbol{w}_{\mathcal{T}}^{\mathrm{OOD}}) \\
&= [\boldsymbol{w}_{\mathcal{T}}^{\mathrm{OOD}}]^T (\boldsymbol{X}_{\mathcal{T}}^+ \boldsymbol{X}_{\mathcal{T}} - (\boldsymbol{X}_{\mathcal{T}}^+ \boldsymbol{X}_{\mathcal{T}})(\boldsymbol{X}_{\mathcal{T}}^+ \boldsymbol{X}_{\mathcal{T}})) \boldsymbol{w}_{\mathcal{T}}^{\mathrm{OOD}} \\
&= 0.
\end{aligned}$$

Since $(\boldsymbol{I}_p - \boldsymbol{X}_{\mathcal{T}}^+ \boldsymbol{X}_{\mathcal{T}}) \boldsymbol{w}_{\mathcal{T}}^{\mathrm{OOD}}$ and $\boldsymbol{X}_{\mathcal{T}}^+ \boldsymbol{X}_{\mathcal{T}} \boldsymbol{w}_{\mathcal{T}}^{\mathrm{OOD}}$ are orthogonal, we have

$$\left\| \boldsymbol{w}_{\mathcal{T}}^{\mathrm{OOD}} \right\|_2^2 = \left\| (\boldsymbol{I}_p - \boldsymbol{X}_{\mathcal{T}}^+ \boldsymbol{X}_{\mathcal{T}}) \boldsymbol{w}_{\mathcal{T}}^{\mathrm{OOD}} \right\|_2^2 + \left\| \boldsymbol{X}_{\mathcal{T}}^+ \boldsymbol{X}_{\mathcal{T}} \boldsymbol{w}_{\mathcal{T}}^{\mathrm{OOD}} \right\|_2^2 \quad \text{(Pythagorean theorem)}$$

and thus

$$\left\| (\boldsymbol{I}_p - \boldsymbol{X}_{\mathcal{T}}^+ \boldsymbol{X}_{\mathcal{T}}) \boldsymbol{w}_{\mathcal{T}}^{\mathrm{OOD}} \right\|_2^2 = \left\| \boldsymbol{w}_{\mathcal{T}}^{\mathrm{OOD}} \right\|_2^2 - \left\| \boldsymbol{X}_{\mathcal{T}}^+ \boldsymbol{X}_{\mathcal{T}} \boldsymbol{w}_{\mathcal{T}}^{\mathrm{OOD}} \right\|_2^2.$$

Putting the equation above into equation 9, we obtain

$$\mathbb{E}_{\boldsymbol{X}} \left[ \left\| \hat{\boldsymbol{w}}_{\mathcal{T}} - \boldsymbol{w}_{\mathcal{T}}^{\mathrm{OOD}} \right\|_2^2 \right] = \mathbb{E}_{\boldsymbol{X}} \left[ \left\| \boldsymbol{w}_{\mathcal{T}}^{\mathrm{OOD}} \right\|_2^2 \right] - \mathbb{E}_{\boldsymbol{X}} \left[ \left\| \boldsymbol{X}_{\mathcal{T}}^+ \boldsymbol{X}_{\mathcal{T}} \boldsymbol{w}_{\mathcal{T}}^{\mathrm{OOD}} \right\|_2^2 \right] + \mathbb{E}_{\boldsymbol{X}} \left[ \left\| \boldsymbol{X}_{\mathcal{T}}^+ \boldsymbol{\eta} \right\|_2^2 \right]. \quad (10)$$

Note that $\boldsymbol{\Pi}_{\mathcal{T}} = \boldsymbol{X}_{\mathcal{T}}^+ \boldsymbol{X}_{\mathcal{T}} = \boldsymbol{X}_{\mathcal{T}}^T (\boldsymbol{X}_{\mathcal{T}} \boldsymbol{X}_{\mathcal{T}}^T)^{-1} \boldsymbol{X}_{\mathcal{T}}$ is the orthogonal projection matrix for the row space of $\boldsymbol{X}_{\mathcal{T}}$. We can thus write $\boldsymbol{X}_{\mathcal{T}}^+ \boldsymbol{X}_{\mathcal{T}} \boldsymbol{w}_{\mathcal{T}} = \boldsymbol{\Pi}_{\mathcal{T}} \boldsymbol{w}_{\mathcal{T}}^{\mathrm{OOD}}$ as a linear combination of rows of $\boldsymbol{X}_{\mathcal{T}}$. Then, using the fact that the $\boldsymbol{x}_i$ in $\boldsymbol{X}$ are i.i.d. and drawn from a standard normal distribution and by rotational symmetry of the standard normal distribution, it follows:

$$\mathbb{E}_{\boldsymbol{X}} \left[ \left\| \boldsymbol{X}_{\mathcal{T}}^+ \boldsymbol{X}_{\mathcal{T}} \boldsymbol{w}_{\mathcal{T}}^{\mathrm{OOD}} \right\|_2^2 \right] = \frac{n}{p} \left\| \boldsymbol{w}_{\mathcal{T}}^{\mathrm{OOD}} \right\|_2^2.$$

Putting equation above in equation 10 into, we obtain

$$\mathbb{E}_{\boldsymbol{X}} \left[ \left\| \hat{\boldsymbol{w}}_{\mathcal{T}} - \boldsymbol{w}_{\mathcal{T}}^{\mathrm{OOD}} \right\|_2^2 \right] = \left\| \boldsymbol{w}_{\mathcal{T}}^{\mathrm{OOD}} \right\|_2^2 \left( 1 - \frac{n}{p} \right) + \mathbb{E}_{\boldsymbol{X}} \left[ \left\| \boldsymbol{X}_{\mathcal{T}}^+ \boldsymbol{\eta} \right\|_2^2 \right].$$

We are going to evaluate $\mathbb{E}_{\boldsymbol{X}} \left[ \left\| \boldsymbol{X}_{\mathcal{T}}^+ \boldsymbol{\eta} \right\|_2^2 \right]$. First, we observe that

$$\begin{aligned}
\boldsymbol{\eta} &= \boldsymbol{z} - \boldsymbol{X}_{\mathcal{T}} \boldsymbol{w}_{\mathcal{T}}^{\mathrm{OOD}} \\
&= \boldsymbol{X}_{\mathcal{T}} \boldsymbol{w}_{\mathcal{T}}^{\mathrm{OOD}} + \boldsymbol{X}_{\mathcal{T}^c} \boldsymbol{w}_{\mathcal{T}^c}^{\mathrm{OOD}} + \boldsymbol{\epsilon} - \boldsymbol{X}_{\mathcal{T}} \boldsymbol{w}_{\mathcal{T}}^{\mathrm{OOD}} \qquad (11) \\
&= \boldsymbol{X}_{\mathcal{T}^c} \boldsymbol{w}_{\mathcal{T}^c}^{\mathrm{OOD}} + \boldsymbol{\epsilon},
\end{aligned}$$

where $\boldsymbol{\epsilon} = [\epsilon_1, \ldots, \epsilon_n]^T$. Because $\boldsymbol{X}_{\mathcal{T}} \boldsymbol{w}_{\mathcal{T}}^{\mathrm{OOD}}$ and $\boldsymbol{X}_{\mathcal{T}^c} \boldsymbol{w}_{\mathcal{T}^c}^{\mathrm{OOD}} + \boldsymbol{\epsilon}$ are both uncorrelated, we have

$$\mathbb{E}_{\boldsymbol{X}} \left[ \left\| \boldsymbol{X}_{\mathcal{T}}^+ \boldsymbol{\eta} \right\|_2^2 \right] = \mathrm{Tr} \left( \mathbb{E}_{\boldsymbol{X}} \left[ \boldsymbol{X}_{\mathcal{T}}^{+T} \boldsymbol{X}_{\mathcal{T}}^+ \right] \mathbb{E}_{\boldsymbol{X}} \left[ \boldsymbol{\eta} \boldsymbol{\eta}^T \right] \right).$$

As $p > n$, we have $\boldsymbol{X}_{\mathcal{T}}^+ = \boldsymbol{X}_{\mathcal{T}}^T (\boldsymbol{X}_{\mathcal{T}} \boldsymbol{X}_{\mathcal{T}}^T)^{-1}$ and thus

$$\mathbb{E}_{\boldsymbol{X}} \left[ \boldsymbol{X}_{\mathcal{T}}^{+T} \boldsymbol{X}_{\mathcal{T}}^+ \right] = \mathbb{E}_{\boldsymbol{X}} \left[ (\boldsymbol{X}_{\mathcal{T}} \boldsymbol{X}_{\mathcal{T}}^T)^{-1} \right]$$

$\boldsymbol{X}_{\mathcal{T}} \boldsymbol{X}_{\mathcal{T}}^T$ follows a Wishart distribution: $\boldsymbol{X}_{\mathcal{T}} \boldsymbol{X}_{\mathcal{T}}^T \sim \mathcal{W}_n(p, I_n)$, and $(\boldsymbol{X}_{\mathcal{T}} \boldsymbol{X}_{\mathcal{T}}^T)^{-1}$ follows an inverse Wishart distribution: $(\boldsymbol{X}_{\mathcal{T}} \boldsymbol{X}_{\mathcal{T}}^T)^{-1} \sim \mathcal{W}_n^{-1}(p, \boldsymbol{I}_n)$. Its expectation is given by:

$$\mathbb{E}[(\boldsymbol{X}_{\mathcal{T}} \boldsymbol{X}_{\mathcal{T}}^T)^{-1}] = \frac{\boldsymbol{I}_n}{p-n-1}.$$

We have

$$\mathbb{E}_{\boldsymbol{X}} \left[ \left\| \boldsymbol{X}_{\mathcal{T}}^+ \boldsymbol{\eta} \right\|_2^2 \right] = \frac{1}{p-n-1} \mathbb{E}_{\boldsymbol{X}} \left[ \boldsymbol{\eta}^T \boldsymbol{\eta} \right].$$

From equation 11, we have

$$\begin{aligned}
\mathbb{E}_{\boldsymbol{X}} \left[ \boldsymbol{\eta}^T \boldsymbol{\eta} \right] &= \mathbb{E}_{\boldsymbol{X}} \left[ (\boldsymbol{z} - \boldsymbol{X}_{\mathcal{T}} \boldsymbol{w}_{\mathcal{T}}^{\mathrm{OOD}})^T (\boldsymbol{z} - \boldsymbol{X}_{\mathcal{T}} \boldsymbol{w}_{\mathcal{T}}^{\mathrm{OOD}}) \right] \\
&= \mathbb{E}_{\boldsymbol{X}} \left[ (\boldsymbol{X}_{\mathcal{T}^c} \boldsymbol{w}_{\mathcal{T}^c}^{\mathrm{OOD}} + \boldsymbol{\epsilon})^T (\boldsymbol{X}_{\mathcal{T}^c} \boldsymbol{w}_{\mathcal{T}^c}^{\mathrm{OOD}} + \boldsymbol{\epsilon}) \right] \\
&= [\boldsymbol{w}_{\mathcal{T}^c}^{\mathrm{OOD}}]^T \mathbb{E}_{\boldsymbol{X}} \left[ \boldsymbol{X}_{\mathcal{T}^c}^T \boldsymbol{X}_{\mathcal{T}^c} \right] \boldsymbol{w}_{\mathcal{T}^c}^{\mathrm{OOD}} + \underbrace{\mathbb{E}_{\boldsymbol{X}} \left[ \boldsymbol{\epsilon}^T \right]}_{=0} \underbrace{\mathbb{E}_{\boldsymbol{X}} \left[ \boldsymbol{X}_{\mathcal{T}^c} \right]}_{=0} \boldsymbol{w}_{\mathcal{T}^c}^{\mathrm{OOD}} + [\boldsymbol{w}_{\mathcal{T}^c}^{\mathrm{OOD}}]^T \underbrace{\mathbb{E}_{\boldsymbol{X}} \left[ \boldsymbol{X}_{\mathcal{T}^c}^T \right]}_{=0} \underbrace{\mathbb{E}_{\boldsymbol{X}} \left[ \boldsymbol{\epsilon} \right]}_{=0} + \mathbb{E}_{\boldsymbol{X}} \left[ \boldsymbol{\epsilon}^T \boldsymbol{\epsilon} \right].
\end{aligned}$$

$\boldsymbol{X}_{\mathcal{T}^c}^T \boldsymbol{X}_{\mathcal{T}^c}$ follows a Wishart distribution, *i.e.*, $\boldsymbol{X}_{\mathcal{T}^c}^T \boldsymbol{X}_{\mathcal{T}^c} \sim \mathcal{W}_q(n, \boldsymbol{I}_q)$. We obtain thus

$$
\begin{aligned}
\mathbb{E}_{\boldsymbol{X}}\left[\boldsymbol{\eta}^T \boldsymbol{\eta}\right] &= n[\boldsymbol{w}_{\mathcal{T}^c}^{\text{OOD}}]^T \boldsymbol{w}_{\mathcal{T}^c}^{\text{OOD}} + n\sigma^2 \\
&= n\|\boldsymbol{w}_{\mathcal{T}^c}^{\text{OOD}}\|_2^2 + n\sigma^2.
\end{aligned} \tag{12}
$$

From equation 12, we deduce that

$$
\mathbb{E}_{\boldsymbol{X}}\left[\left\|\boldsymbol{X}_{\mathcal{T}}^+ \boldsymbol{\eta}\right\|_2^2\right] = \tfrac{n}{p-n-1}\left(\|\boldsymbol{w}_{\mathcal{T}^c}^{\text{OOD}}\|_2^2 + \sigma^2\right).
$$

and

$$
\mathbb{E}_{\boldsymbol{X}}\left[\left\|\hat{\boldsymbol{w}}_{\mathcal{T}} - \boldsymbol{w}_{\mathcal{T}}^{\text{OOD}}\right\|_2^2\right] = \|\boldsymbol{w}_{\mathcal{T}}^{\text{OOD}}\|_2^2\left(1 - \tfrac{n}{p}\right) + \tfrac{n}{p-n-1}\left(\|\boldsymbol{w}_{\mathcal{T}^c}^{\text{OOD}}\|_2^2 + \sigma^2\right).
$$

$\square$

# B  Details about Baselines and Architecture Implementations

## B.1  Baseline OOD Methods

In this section, we present an overview of the baseline methods used in our experiments. We describe the principles behind these baselines, and the chosen hyperparameters. It is worth noting that extensive hyperparameter search for each method were not performed to maintain stability. Hence, once the final model is selected, hybrid methods like ViM, ASH and NECO performance may increase if such task is performed.

**Softmax Score.**  This score uses the maximum softmax probability (MSP) of the model as an OOD scoring function (Hendrycks & Gimpel, 2017).

**Energy.**  Liu et al. (2020) proposes using the energy score for OOD detection, where the energy function maps the logit outputs to a scalar. To maintain the convention that lower scores correspond to in-distribution (ID) data, (Liu et al., 2020) uses the negative energy as the OOD score.

**ReAct.**  Sun et al. (2021) propose clipping extreme-valued activations. The original paper found that clipping activations at the $90th$ percentile of ID data was optimal. Moreover, as the authors propose, we employ the ReAct+Energy configuration.

**KL-Matching & MaxLogit.**  KL-Matching computes the class-wise average probability using the entire training dataset. Consistent with the approach outlined in (Hendrycks et al., 2022), this calculation is based on the predicted class rather than the ground-truth labels. MaxLogit employs the maximum logit value of the model as an OOD scoring function.

**Mahalanobis.**  This score leverages the feature vector from the layer preceding the final classification layer (Lee et al., 2018a). To estimate the precision matrix and the class-wise mean vector, we used the entire training dataset. It's important to note that we incorporated ground-truth labels during this computation process.

**ViM & Residual.**  Wang et al. (2022) decomposes the latent space into a principal space $P$ and a null space $P^\perp$. The ViM score is calculated by projecting the features onto the null space to create a virtual logit, which is then combined with the logits using the norm of this projection. To enhance performance, they calibrate this norm with a constant which is determined by dividing the sum of the maximum logits by the sum of the norms of the null space projections, both measured on the training set. The Residual score is derived by computing the norm of the latent vector's projection onto the null space. We followed the author's suggestions for the null space, by setting it to half the size of the full feature vector, adapted to each model width.

**ASH.**  Djurisic et al. (2023) employs activation pruning at the penultimate layer, just before the application of the DNN classifier. This pruning threshold is determined on a per-sample basis, eliminating the need for pre-computation of ID data statistics. The original paper presents three different post-hoc scoring functions, with the only distinction among them being the imputation

method applied after pruning. We employ ASH-P in our experiments as it performed the best, in which the clipped values are replaced with zeros. As specified in the original paper, we fix the pruning threshold value to $90\%$.

**NECO.** Ammar et al. (2024) leverages the geometric properties of Neural Collapse, measuring the relative norm of a sample within the subspace defined by the ETF to identify OOD samples. NC typically involves a collapse in the variability of class representations, leading to a more structured and simplified feature space. It is hypothesized that this collapse also impacts OOD detection, particularly through the emerging orthogonality between ID and OOD data. NECO utilizes this orthogonality to effectively distinguish between ID and OOD data by measuring the relative norm of each data point within the approximated ETF space scaled by the maximum logit value as the OOD score. We use a dimension $d = c$ to approximate the ETF sub-space for all architectures, with $c$ being the number of classes.

### B.2   Experiments Setup

**ViT Experimental Setup.**   For all experiments, we trained a set of ViT models with hidden dimensions [4, 8, 12, 16, 20, 24, 28, 32, 36, 40, 60, 80, 100, 120, 160, 200, 240, 280, 320, 360, 400, 480, 520, 600, 680, 760, 800]. This hidden dimension serve as the dimension of the output layer after the linear transformation (the class-token size). The dimension of the FeedForward layer is the width multiplied by 4. The input size is set to 32 and the patch size to 8, no dropout is used and we use 4 heads with 4 Transformer blocks. The ViT models are first randomly initialized and then trained on CIFAR-10 using stochastic gradient descent with CE loss. The weights are fine-tuned for 60 000 steps, with no warm-up steps, 1 024 batch size, 0.9 momentum, and a learning rate of 0.03.

**Swin Experimental Setup.**   We used a standard 4 block Swin architecture, with a downscaling factor of (2,2,2,1) for each block respectively. The width ranges from 1 to 100, with a window size of 4, an input size of 32, and a filter size of 4. The model is randomly initialized and then optimized using an Adam optimizer with CE loss for a 1 000 epoch using a batch size of 1 024. The initial learning rate is 0.0001.

**CNN Experimental Setup.**   Similar to Nakkiran et al. (2021), we define a standard family CNN models formed by 4 convolutional stages of controlled base width $[k, 2k, 4k, 8k]$, for $k$ in the range of [1, 128], with a fully connected layer as classifier. The MaxPool is set to [2, 2, 2, 4] for the four blocks respectively. For all the convolution layers, the kernel size is set to 3, stride and padding to 1.

## C   Details about Neural Collapse

For overparametrised model trained through the terminal phase of training (TPT), Neural Collapse (NC) phenomenon emerges, particularly in the penultimate layer and in the linear classifier of DNNs (Papyan et al., 2020; Ammar et al., 2024). NC is characterized by five main properties:

1. **Variability Collapse (NC1):** the within-class variation in activations becomes negligible as each activation collapses toward its respective class mean.

2. **Convergence to Simplex ETF (NC2):** the class-mean vectors converge to having equal lengths, as well as having equal-sized angles between any pair of class means. This configuration corresponds to a Simplex Equiangular Tight Frame (ETF).

3. **Convergence to Self-Duality (NC3):** in the limit of an ideal classifier, the class means and linear classifiers of a neural network converge to each other up to rescaling, implying that the decision regions become geometrically similar and that the class means lie at the centers of their respective regions.

4. **Simplification to Nearest Class-Center (NC4):** The network classifier progressively tends to select the class with the nearest class mean for a given activation, typically based on standard Euclidean distance.

5. **ID/OOD Orthogonality (NC5):** As the training procedure advances, OOD and ID data tend to become increasingly more orthogonal to each other. In other words, the clusters of

OOD data become more perpendicular to the configuration adopted by ID data (*i.e.*, the Simplex ETF).

These NC properties provide valuable insights into DNNs learned representation structure and properties, which allows for a considerable simplification. Additionally, the convergence of NC can be linked to OOD detection Ammar et al. (2024); Haas et al. (2023); Zhao & Cao (2023); Zhang et al. (2024). For further details refer to (Papyan et al., 2020; Ammar et al., 2024)

## D  Complementary Results on Double Descent and OOD Detection

### D.1  ReLU Random Feature Model Experimental Setting

ReLU Random Feature (RRF) models (Rahimi & Recht, 2007) offer a class of simple yet expressive models. These models approximate two-layer neural networks by fixing the first-layer weights randomly and learning linear output weights. Formally, the model is defined as:

$$RRF(x) = \sum_{i=1}^{m} a_i \, \phi(w_i^\top x + b_i), \quad \text{where} \quad \phi(z) = \max(0, z), \tag{13}$$

with $w_i \sim \mathcal{D}_w$, $b_i \sim \mathcal{D}_b$ drawn i.i.d. from fixed distributions, and the output weights $a_i \in \mathbb{R}$ learned via linear regression on the transformed data. At their core, these models perform a randomized nonlinear projection followed by a linear mapping, offering a computationally tractable means of studying nonlinear learning behaviors.

We stack a sigmoid activation function on the model outputs, which aligns with the theoretical setting underlying Theorem 1 and Theorem 2:

$$f(x) = \phi'(RRF(x)), \quad \text{where} \quad \phi'(z) = \frac{1}{1 + e^{-z}},$$

This allows us to closely mirror the binary classification setting.

We vary the model width (i.e., the number of random features) from 2 to 10 000 and train each model for 600 epochs. For training and evaluation, we generate two datasets of 1 000 samples each by selecting two CIFAR-10 classes (class 0 and class 1) and fitting Gaussian Mixture Models (GMMs) to this data. These GMMs then serve as generative sources for the training and test samples. Out-of-distribution (OOD) data is generated by sampling 1 000 inputs from a distinct Gaussian distribution with the same shape as CIFAR-10 images.

To evaluate OOD detection, we employ the following score, which is tailored to the binary classification context:

$$g(x) = \frac{|\hat{y} - 0.5|}{0.5} \tag{14}$$

where $\hat{y}$ denotes the model's prediction. This score reflects our OOD detection risk proxy defined in equation 5, capturing the model's confidence by penalizing under-confident predictions (a sigmoid value close to 0.5). Interestingly, the AUC drops to around 50% at the interpolation threshold, indicating behavior no better than random guessing. To better understand how the OOD score evolves across the double descent curve, Figure 3 shows the distribution of $g(x)$ from equation 14 across different model complexities—specifically in underparameterized, overparameterized, and interpolation-threshold regimes. At the interpolation threshold, the distribution is notably heavy-tailed, revealing that the model is producing extreme and unjustified confident outputs, often close to 0 or 1 for all inputs. This overconfidence, applied indiscriminately to both ID and OOD samples, explains the poor AUC for the defined OOD score in equation 14. In contrast, models outside the interpolation region exhibit more informative score distributions: OOD samples tend to receive low scores, while ID scores are more uniformly spread, enabling better separation. However, as model complexity approaches the interpolation threshold from either side (width of value 500 and 5 000), score separability start diminishing, primarily because OOD samples begin receiving increasingly higher confidence scores.

### D.2  CIFAR-10 Additional Results

To further show the consistency of double descent for OOD detection, Figures 4, 5 6, 7 and 8 show the OOD detection metrics performance on six more semantic-shift OOD datasets. To illustrate the

performance of other OOD-methods while maintaining visibility, we show two different methods at each dataset alongside the better-performing and most stable three: MSP, NECO, and ASH.

## D.3 CIFAR-100 Results

In this section, we present results for the CIFAR-100 dataset as ID for a ResNet-18 model. Figure 9 illustrates the OOD detection metrics performance, Figure 10 shows the accuracy and eigenvalues distribution (see section E.1 for discussion about eigenvalues). We can observe similar behaviors between the ResNet-18 trained on CIFAR-10, and this current configuration on a harder dataset (CIFAR-100).

Table 2 illustrates the evolution of *AUC* between the underparametrized and overparametrized regime and its correlation with the $NC1_{u/o}$ for the remaining OOD datasets. As the CNN's $NC1_{u/o} < 1$, its performance stagnates or deteriorates with overparametrization, while the other models improve. Additionally, we can see how the hybrid-based methods improve considerably and become competitive with logit-methods when $NC1_{u/o} > 1$.

Table 2: Models performance in terms of `AUC` in the underparametrized local minima (`AUC`$_u$) and the overparametrized maximum width (`AUC`$_o$), *w.r.t* $NC1_{u/o}$ value. Best is highlighted in green when `AUC`$_u$ is higher, red when `AUC`$_u$ is higher and blue if both `AUC` are within standard deviation range. The highest `AUC` value per-dataset and per-architecture is highlighted in **bold**.

| Model | $NC1_{u/o}$ | Method | ImageNet-O AUC$_u$ ↑ | ImageNet-O AUC$_o$ ↑ | Textures AUC$_u$ ↑ | Textures AUC$_o$ ↑ | iNaturalist AUC$_u$ ↑ | iNaturalist AUC$_o$ ↑ |
|---|---|---|---|---|---|---|---|---|
| CNN | 0.88 | Softmax score | 71.60+0.37 | **71.78±0.57** | 70.76±0.99 | **74.80±0.78** | 66.19±1.27 | **72.07±1.56** |
| | | MaxLogit | 68.91±0.96 | 65.94±0.51 | 62.01±3.75 | 45.56±1.06 | 63.38±1.89 | 61.58±2.07 |
| | | Energy | 65.06±1.57 | 65.76±0.51 | 53.62±5.87 | 45.07±1.04 | 60.11±2.32 | 61.32±2.05 |
| | | Energy+ReAct | 59.06±2.60 | 58.08±0.80 | 42.80±7.54 | 32.67±0.86 | 51.17±2.82 | 49.41±2.47 |
| | | NECO | 67.45±1.14 | 68.89±0.35 | 56.54±3.83 | 55.75±3.63 | 59.79±1.96 | 68.52±3.60 |
| | | ViM | 67.13±1.61 | 65.76±0.51 | 49.84±3.59 | 45.09±0.79 | 51.58±2.82 | 61.37±1.96 |
| | | ASH-P | 66.20±1.48 | 66.26±0.57 | 56.07±5.94 | 46.23±1.07 | 60.73±2.83 | 61.52±2.11 |
| ResNet | 1.96 | Softmax score | 70.90±0.52 | 75.91±0.49 | 68.03±0.75 | 72.78±1.14 | 65.79±2.16 | 74.85±1.39 |
| | | MaxLogit | 69.45±0.88 | 72.50±0.88 | 65.06±1.80 | 64.43±2.04 | 63.88±3.47 | 72.82±2.32 |
| | | Energy | 67.36±1.26 | 72.44±0.89 | 62.28±2.55 | 64.35±1.84 | 61.73±4.97 | 72.77±2.33 |
| | | Energy+ReAct | 68.02±1.41 | 72.00±0.81 | 64.15±2.68 | 66.13±1.20 | 61.68±6.83 | 71.85±2.60 |
| | | NECO | 70.42±0.83 | **76.11±1.42** | 67.56±1.84 | 73.18±3.20 | 64.50±3.51 | **75.12±2.21** |
| | | ViM | 70.94±1.54 | 75.03±0.62 | 77.88±1.66 | **81.02±1.42** | 62.56±4.91 | 67.12±2.44 |
| | | ASH-P | 67.36±1.26 | 71.57±0.94 | 62.28±2.55 | 62.96±2.11 | 61.73±4.97 | 71.72±2.28 |
| Swin | 1.70 | Softmax score | 56.62±3.15 | 64.78±1.48 | 49.71±3.38 | 63.51±0.33 | 49.10±6.81 | 60.26±2.08 |
| | | MaxLogit | 55.86±3.95 | 64.70±2.07 | 49.47±4.18 | 60.29±0.38 | 49.22±5.07 | 58.91±1.81 |
| | | Energy | 49.58±5.48 | 64.56±2.05 | 49.49±6.73 | 59.96±0.46 | 51.61±7.96 | 58.73±1.75 |
| | | Energy+ReAct | 49.81±5.51 | 65.43±2.09 | 50.66±6.07 | 62.38±0.24 | 51.87±7.51 | 59.39±1.83 |
| | | NECO | 57.63±4.14 | 68.19±2.71 | 56.50±3.64 | 68.06±0.66 | 49.09±5.14 | 62.58±2.21 |
| | | ViM | 65.18±1.83 | **73.45±2.25** | **84.47±1.46** | 78.67±1.65 | **67.86±2.44** | 63.83±2.63 |
| | | ASH-P | 49.46±5.66 | 64.48±2.11 | 48.96±7.85 | 59.90±0.48 | 51.33±10.79 | 58.69±1.99 |
| ViT | 2.32 | Softmax score | 64.17±0.64 | 63.64±0.80 | 67.73±1.82 | 70.27±0.36 | 52.79±0.77 | 58.23±0.51 |
| | | MaxLogit | 63.15±0.88 | 68.87±0.52 | 67.22±2.25 | 79.25±0.38 | 51.90±1.95 | 61.28±0.94 |
| | | Energy | 61.24±1.08 | 69.10±0.50 | 65.59±2.78 | 79.68±0.38 | 51.11±3.04 | 61.40±0.97 |
| | | Energy+ReAct | 61.30±1.21 | 69.09±0.49 | 65.64±2.86 | **79.68±0.38** | 51.63±4.61 | 61.39±0.98 |
| | | NECO | 65.83±1.35 | **69.95±0.37** | 69.41±2.16 | 77.32±0.49 | 52.50±2.05 | **62.96±0.89** |
| | | ViM | 68.76±1.52 | 67.82±0.55 | 65.58±2.69 | 74.13±0.38 | 56.41±3.81 | 60.42±0.90 |
| | | ASH-P | 61.24±1.08 | 68.96±0.48 | 65.59±2.78 | 79.39±0.42 | 51.11±3.04 | 61.28±0.99 |

## D.4 ResNet-34 Results

In this section, we present the results for the ResNet-34 architecture, a deeper version of the ResNet family. Results include both CIFAR-10 and CIFAR-100 datasets as ID. Figure 12 shows the accuracy curve and Figure 11 depicts the OOD detection metrics performance. We observe a similar curve to that of ResNet-18 for both datasets and for all OOD methods, with slightly higher performances. We highlight that the interpolation threshold occurs at a smaller width for ResNet-34 (k=8), compared to

ResNet-18 (k=10) for the CIFAR-10 case. This higher complexity also contributes to its lower NC1 values in Figure 22.

### D.5 Out-of-Distribution Risk Experiments

In Section 4.1, we introduced $R_{\text{OOD}}(\hat{f})$ as a metric for OOD detection. Figures 13, 14, 15, 16, and 17 show the expected OOD risk as a function of the model width and highlight a similar double descent phenomenon across all OOD datasets.

### D.6 Experiments Without Label-Noise Results

Like experiments depicted in Nakkiran et al. (2021), our experimental results in Section 5 consider noisy labels. In this section, we consider a noiseless setting with Figure 18 that depicts the model accuracy as a function of the model width in a noiseless setup. Instead of a double descent phenomenon, a plateau or stagnation appears near the interpolation threshold. Similarly, Figure 19 features the OOD detection performance with respect to the model width under this setup. The curves exhibit a similar behavior than for the model accuracy. Moreover, we observe that removing label noise makes the learning task easier with better performance, e.g., ResNet-18 max accuracy rising from $83.40\%$ to $94.48\%$ with a similar rise observed for `AUC`.

## E Model representations and Neural Collapse Analysis

### E.1 Structure of the Model Representation

Convergence towards Neural Collapse can be an indicator of improved model representation, as defined by the ETF structure. As such, the model eigenvalues distribution can be seen as describing how much the model representation aligns with this structured manifold. The properties of the ETF implies that the top $c$ eigenvalues will be equally prominent, as they represent the top $c$ eigenvectors that constructs the ETF subspace (Papyan et al., 2020; Ammar et al., 2024). The remaining eigenvalues should be less influential, as the ID the data lies essentially in the $c$-dimensional ETF subspace. Figure 20 shows the distribution of each model eigenvalues at the overparametrized regime. It is worth noting that it seems that for all architectures, all overparametrized model widths show similar curves. From Figure 20 we can see that ResNet-18 and Swin models follow the expected NC pattern, with a steep drop in importance at the $c^{th}$ eigenvalue indicating the limit of the ETF. However, Figure 20 also shows that ViT and CNN exhibit a slowly decaying curve, which indicates a lack of clear separation in the model representation between highly important ID information and noisy features. This lack of a global structure in their representation results in both models failing to reliably outperform their underparametrized minima.

This highlights the importance of the ETF structure and NC convergence in enhancing representation stability, which can be useful for improving ID classification and OOD detection tasks.

### E.2 Evolution of NC1

In this section, we further analyze the evolution of NC1 and the model's learned representation with overparametrization, to further show their correlation. Our analysis will focus primarily on the ResNet and CNN models, due to their similarities. To compare the NC convergence across architectures, we include the overparametrized and underparametrized NC1 value of each architecture in Table 3. We will not address the transformer-based models whose performance, especially for generalization, were lower than those of ResNet or CNN. This is because transformers typically require extensive pre-training, particularly for small datasets, and this was not the case for our experiments.

In order to visualize the variability collapse predicted by NC1, Figure 21 shows the last-layer activations for both models at their optimal underparameterized and overparameterized widths. In ResNet, transitioning to overparameterized models leads to significant improvements in the compactness and separation of ID clusters, as well as enhanced orthogonality with OOD points. In contrast, the CNN model does not show clear improvements in ID compactness or OOD separation, making it difficult to determine which representation is better.

Figure 22 shows NC1 metric against the model widths for both ResNets and CNN. While both models exhibit a double descent pattern, CNN barely matches its underparameterized metric value, whereas ResNet continuously improves with added complexity. This discrepancy in NC convergence with overparametrization explains why only ResNet benefits from increased complexity, suggesting that without improvement in NC, increasing model complexity provides no benefits for the learned representations.

Table 3: Comparison of NC1 values for different architectures in under- and over-parameterized regimes.

| Architecture | NC1 (Under-parameterized) | NC1 (Over-parameterized) |
|---|---|---|
| ResNet-18 | $1.35 \pm 0.04$ | $0.687 \pm 0.03$ |
| ResNet-34 | $1.29 \pm 0.02$ | $0.64 \pm 0.02$ |
| CNN | $2.37 \pm 0.08$ | $2.71 \pm 0.03$ |
| Swin | $35.08 \pm 3.53$ | $20.65 \pm 1.88$ |
| ViT | $29.52 \pm 2.17$ | $12.71 \pm 0.74$ |
| ResNet (Depth) | $1.53 \pm 0.02$ | $1.57 \pm 0.09$ |

### E.3 Mahalanobis and Residual Joint Performance

We noticed that for all architectures, and on all OOD datasets, Mahalanobis and Residual follow the same evolution curve, with usually slightly higher *AUC* in favor of Mahalanobis. This behavior is intriguing, due to the fact that each method relies on different types of information. While Mahalanobis models the ID distribution, *i.e.*, the principal space, Residual relies on computing the null space norm, which is orthogonal to the principal space.

We associate this behavior with the noise isolation in each architecture, which is specific to the double descent training paradigm. Indeed, in order for models to be able to perfectly interpolate all the training data and achieve (almost) zero training error, noisy samples must be represented closer to their assigned (noisy) label, rather than to their true label. This will cause the train class clusters (using the true labels) to be less compact and separable, making their high-likelihood region to span almost the entire principal space, in which the ID data is represented. Hence, to separate ID from OOD, learning the Mahalanobis GMM (fitted on the train data) becomes equivalent to separating the principal and null space, which is the same reasoning behind the Residual score.

This overfitting occurs at the interpolation threshold, which causes the learned distributions by Mahalanobis to be sparse and not robust to OOD data, impeding its improvement as we transition towards overparametrization. It is important to note also that both of these methods are usually below, or struggle to surpass the random choice threshold of $0.5$ *AUC* in the overparametrized regime (with the exception of texture dataset on ResNet-18 case).

Interestingly, both of these methods suffer much less from this behavior under the Transformer based architecture, and even exhibit a double descent curve on most datasets. This can be explained by the fact that even the most overparametrized Transformer variant have a training error higher than $4\%$. In comparison, convolutional models consistently achieve a training error lower than $0.01\%$. Hence, Transformers suffer less from this effect because they have not interpolated the noise in the training data perfectly. It is worth noting that interpolating the noise is desirable, as it is necessary for Generalization in this setup (Bartlett et al., 2020). Transformer-based architectures require extensive pre-training to generalize well, especially for small scale dataset, which was not performed in our experiments. This inability of transformers to perfectly interpolate the training data contributes to their lower performance in terms of Generalization in the overparametrized regime, especially in the ViT case.

## F Depth-Wise Double Descent

In this section, we investigate how the model depth affects ID generalization and post-hoc OOD detection performance. Most works on the double descent phenomenon focus on varying the network width rather than depth (Belkin et al., 2020; Couillet & Liao, 2022; Bach, 2024; Nakkiran et al., 2021). A primary reason stems from challenges in modifying network depth within both empirical and theoretical frameworks. From an empirical perspective, increasing depth often introduces

training instabilities and requires hyperparameter adjustments (e.g., learning rates) that complicate comparative analyses. From a theoretical perspective, studies typically rely on frameworks like Random Matrix Theory (RMT) or statistical physics, which often model networks as linear systems with random features. Increasing the depth also increases nonlinear interactions and hierarchical dependencies, which may limit the applicability of these mathematical tools.

## F.1 Experimental Setup

We conduct experiments on ResNet-like architectures composed of four residual blocks, with CIFAR-10 as the ID dataset. The width of the network is fixed (at $k = 10$), while the model complexity is controlled by modifying the model depth. In particular, the model depth is defined as the total number of residual layers across all four blocks. Starting with shallow configurations, we incrementally increase the number of layers per block, focusing on deepening of the blocks with the fewest layers first. For instance, the sequence of architectures progresses as follows:

$$1, 1, 1, 1 \rightarrow 2, 1, 1, 1 \rightarrow 2, 2, 1, 1 \rightarrow 2, 2, 2, 2 \rightarrow 3, 2, 2, 2 \rightarrow \cdots \rightarrow 10, 10, 10, 10,$$

where each tuple denotes the number of residual layers in each block, respectively. In Figure 23, the $x$-axis depicts the total depth, i.e. the sum of the four block values. To obtain architectures with depth less than $4$, we replace residual blocks with MaxPooling layers to maintain a constant overall width across all models. The training protocol follows the same setup as the width-wise experiments (see Section 5.2). Furthermore, the same set of metrics is considered: test accuracy, OOD detection AUC, Neural Collapse, and the OOD risk defined in equation 5.

## F.2 Results

Figure 23 depicts a similar double descent phenomenon for model depth as observed for model width. Furthermore, Table 4 shows a correlation between the $NC1_{u/o}$ metric and OOD detection performance, which may suggest an alignement between the $NC1_{u/o}$ and OOD performance.

Table 4: Models performance in terms of AUC in the underparametrized local minima ($\text{AUC}_u$) and the overparametrized maximum width ($\text{AUC}_o$), *w.r.t* $NC1_{u/o}$ value. Best is highlighted in green when $\text{AUC}_o$ is higher, red when $\text{AUC}_u$ is higher and blue if both AUC are within standard deviation range. The highest AUC value per-dataset and per-architecture is highlighted in **bold**.

| Model | $NC1_{u/o}$ | Method | SUN | | Places365 | | CIFAR-100 | |
| | | | $\text{AUC}_u \uparrow$ | $\text{AUC}_o \uparrow$ | $\text{AUC}_u \uparrow$ | $\text{AUC}_o \uparrow$ | $\text{AUC}_u \uparrow$ | $\text{AUC}_o \uparrow$ |
|---|---|---|---|---|---|---|---|---|
| Depth | 0.97 | Softmax | 72.45±0.99 | 69.23±2.66 | 72.47±0.86 | 69.01±0.97 | **71.62±0.65** | 68.84±1.39 |
| | | MaxLogit | 72.57±0.72 | 71.00±2.86 | 72.57±1.15 | 70.76±1.93 | 70.66±0.74 | 70.82±2.16 |
| | | Energy | 71.81±0.70 | 70.98±2.87 | 71.83±1.24 | 70.74±1.92 | 69.43±0.80 | 70.81±2.17 |
| | | ReAct | 71.83±1.58 | 71.56±2.99 | 71.89±1.83 | 71.30±2.04 | 70.13±1.32 | 70.71±2.28 |
| | | NECO | **72.83±0.89** | 71.24± 2.12 | **72.74±1.03** | 70.77±1.99 | 70.85±0.76 | 69.54±1.07 |
| | | ViM | 71.86±2.11 | 70.32±4.91 | 71.07±1.70 | 69.46±3.92 | 71.14±1.31 | 67.29±1.70 |
| | | ASH-P | 71.78±0.66 | 70.50±3.29 | 71.79±1.27 | 70.19±2.40 | 69.40±0.87 | 70.34±2.45 |

## F.3 Connection to Width-Wise Experiments.

Although the depth-wise experiments also exhibit a double descent phenomenon, it differs in important ways from the width-wise case observed in ResNet-18 and ResNet-34. Notably, the best models in both the underparameterized and overparameterized regimes achieve comparable performance. This distinct behavior is reflected in the ratio $NC1_{u/o} = 0.97 < 1$, indicating that feature collapse did not improve with increased depth.

These findings emphasize that out-of-distribution (OOD) performance is influenced not only by architectural biases but also by the specific training dynamics and representational geometry induced by scaling along different dimensions. It is important to note that at very large depths, additional complexities such as optimization instabilities and slower convergence emerge.

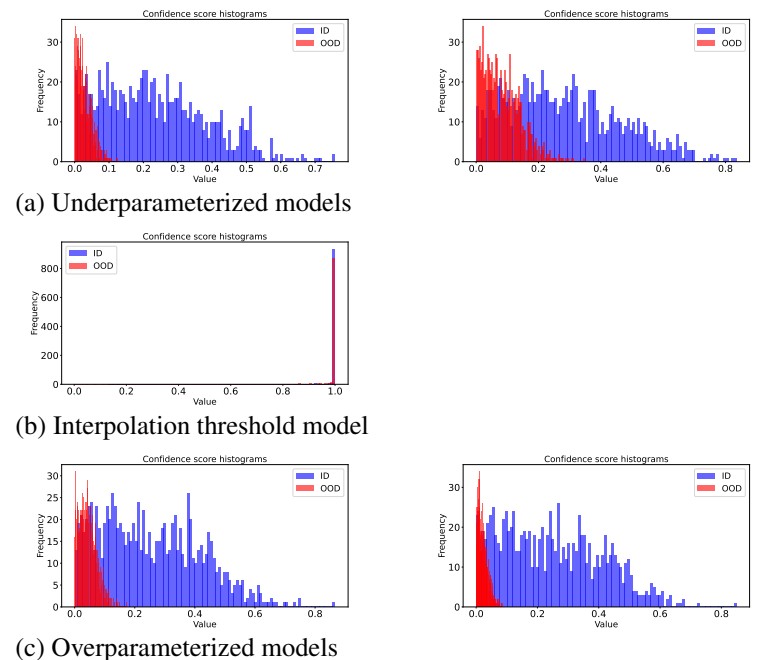

(a) Underparameterized models

(b) Interpolation threshold model

(c) Overparameterized models

Figure 3: Distribution of the values of the OOD scoring function g(x) defined in equation 14, evaluated on Random ReLU Feature (RRF) models with varying widths. (a) Score distributions for underparameterized models with widths, from left to right, of 100 and 500 respectively. (b) Score distribution at the interpolation threshold (width = 1 000), where performance degrades sharply. (c) Score distributions for overparameterized models with widths, from left to right, of 5 000 and 10 000 respectively.

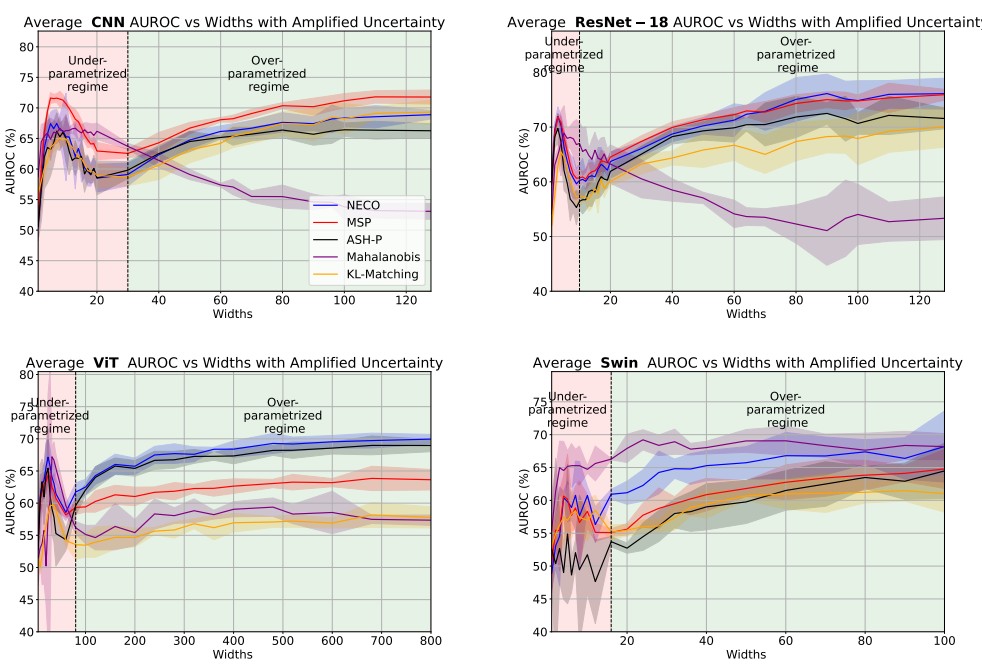

Figure 4: OOD detection (AUC) metric versus model width. Experiments performed on CNN, ResNet-18, ViT, and Swin with CIFAR10 as ID dataset and ImageNet-O as OOD dataset.

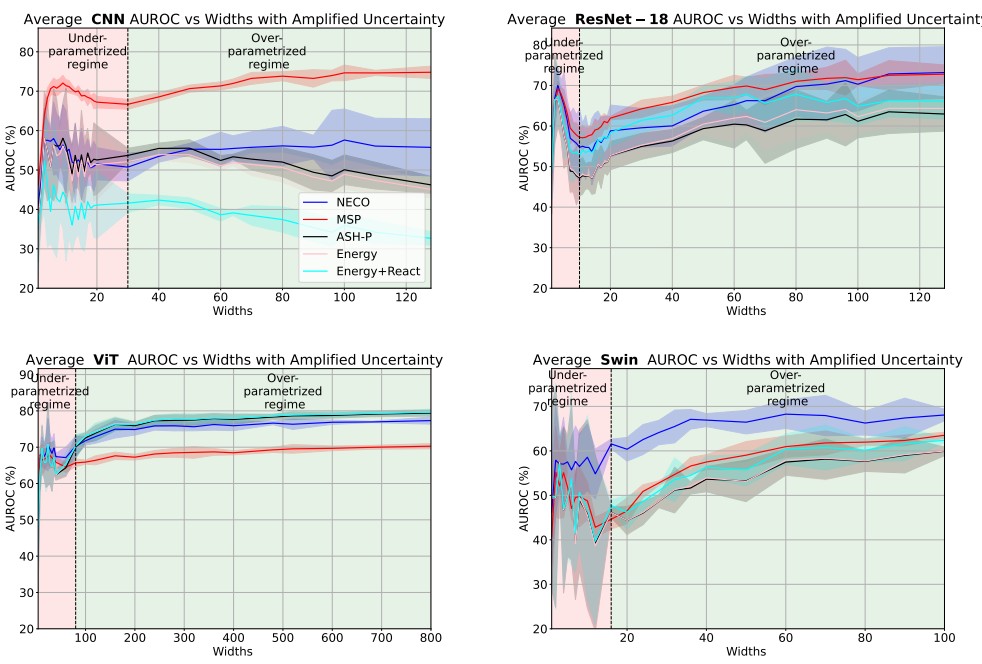

Figure 5: OOD detection (`AUC`) metric versus model width. Experiments performed on CNN, ResNet-18, ViT and Swin with CIFAR10 as ID dataset and Texture as OOD dataset.

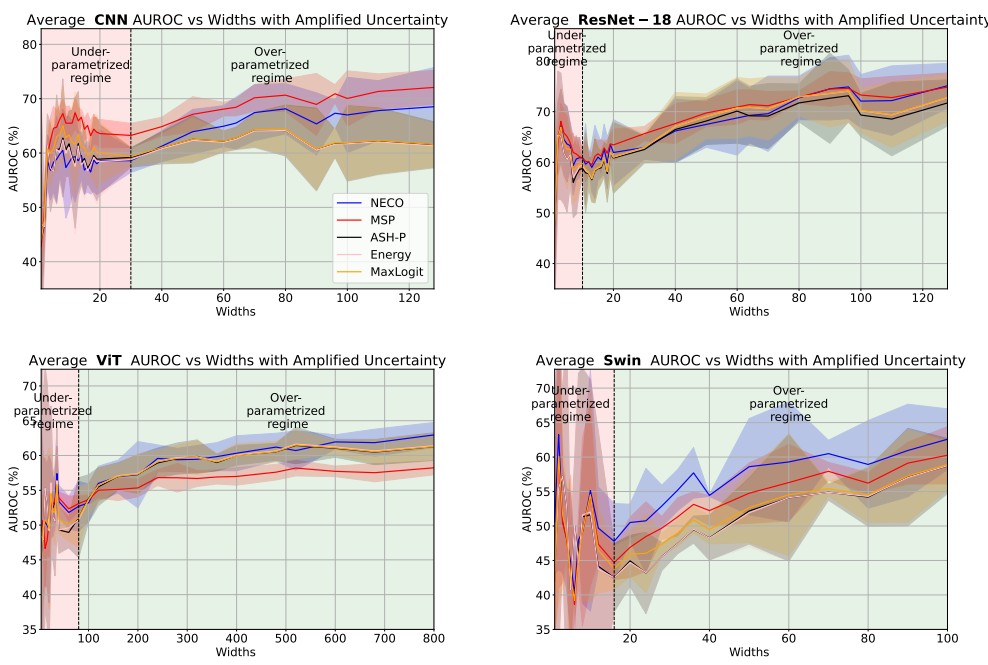

Figure 6: OOD detection (`AUC`) metric versus model width. Experiments performed on CNN, ResNet-18, ViT, and Swin with CIFAR10 as ID dataset and iNaturalist as OOD dataset.

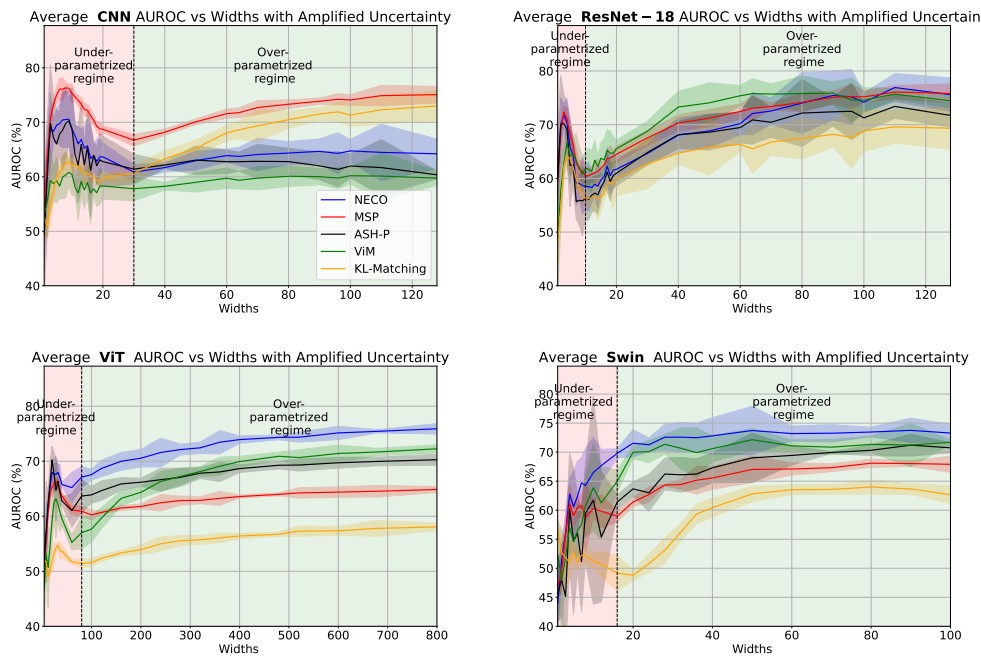

Figure 7: OOD detection (`AUC`) metric versus model width. Experiments performed on CNN, ResNet-18, ViT, and Swin with CIFAR10 as ID dataset and SUN as OOD dataset.

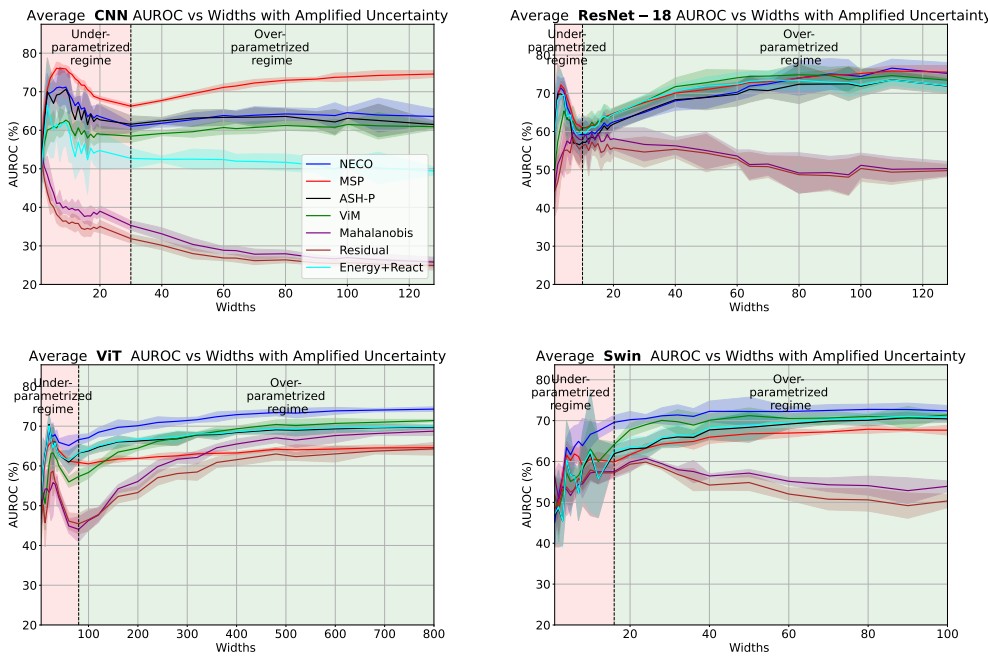

Figure 8: OOD detection (`AUC`) metric versus model width. Experiments performed on CNN, ResNet-18, ViT, and Swin with CIFAR10 as ID dataset and places365 as OOD dataset.

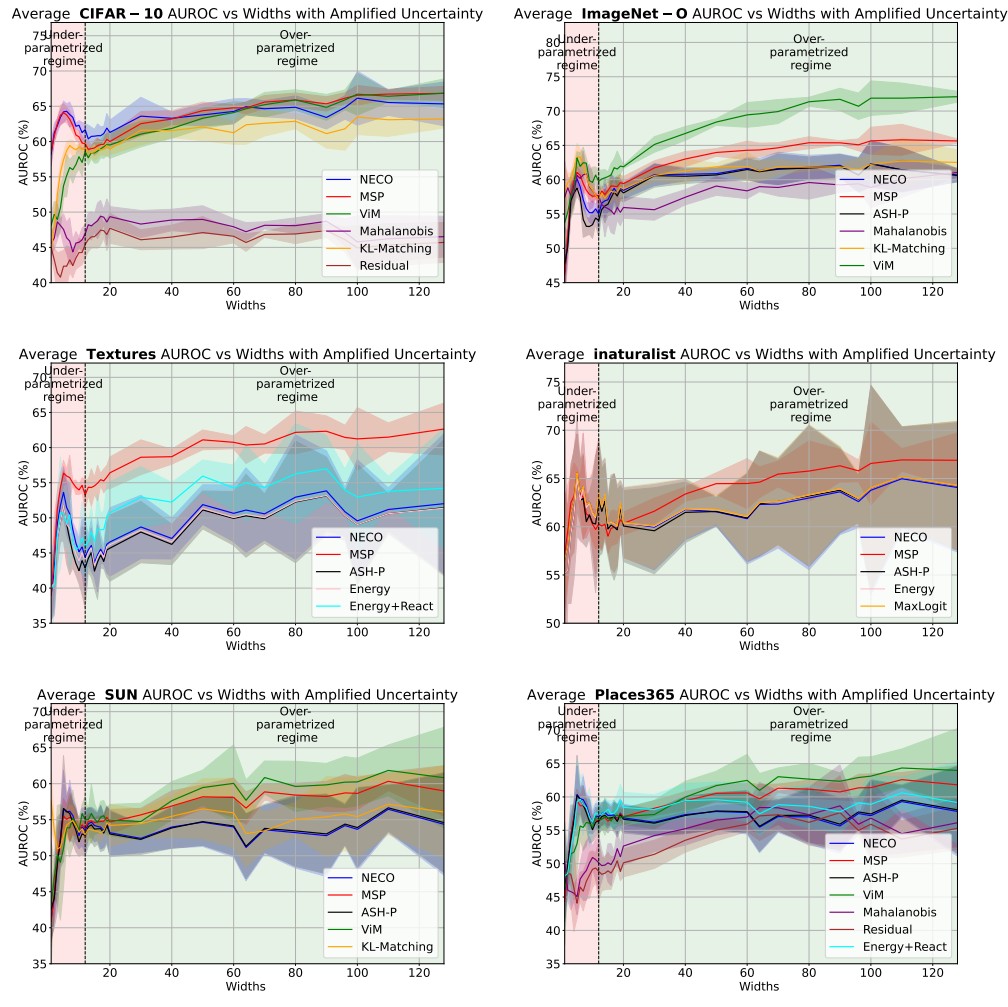

Figure 9: OOD detection (`AUC`) metric versus model width. Experiments performed on ResNet-18 with CIFAR100 as ID dataset and CIFAR-10, ImageNet-O, Texture, iNaturalist, SUN and places365 as OOD dataset.

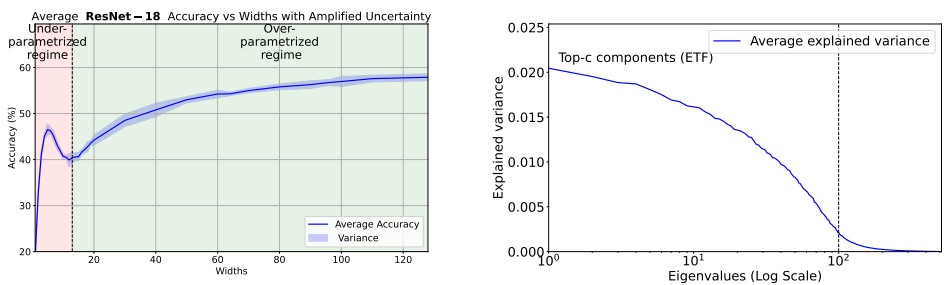

Figure 10: Accuracy versus model width (left) of ResNet-18 model on CIFAR-100. Explained variance versus eigenvalues (right) in the overparametrized regime for ResNet-18 (width 64) on CIFAR-100. The black line depicts the $100^{th}$ eigenvalue.

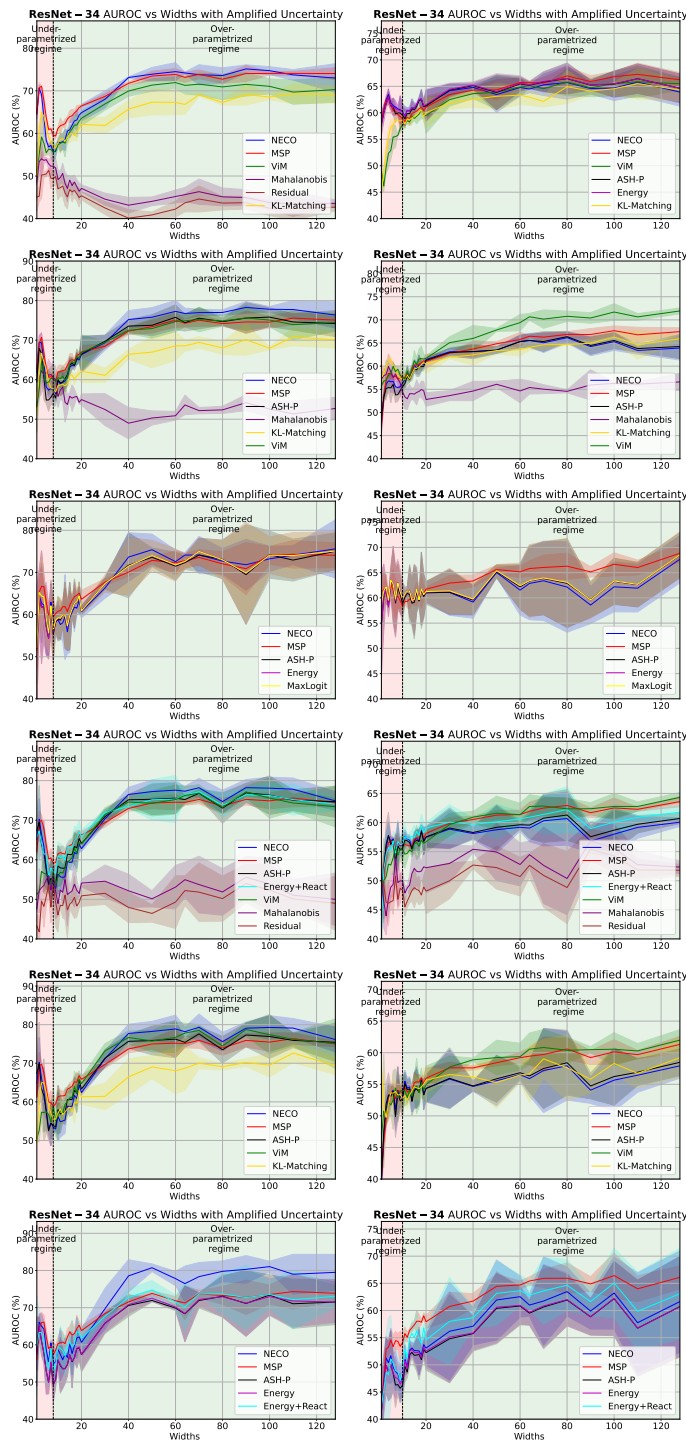

Figure 11: OOD detection (AUC) metric versus model width. Experiments performed on ResNet-34 with CIFAR10 (left) and CIFAR-100 (right) as ID dataset and CIFAR-100 (right), CIFAR-10 (left), ImageNet-O, iNaturalist, places365, SUN and Textures as OOD dataset (from top to bottom).

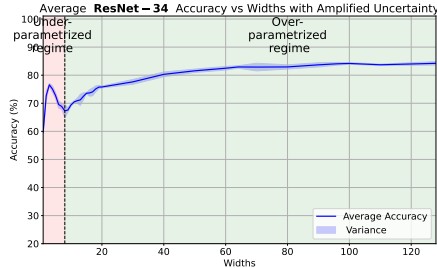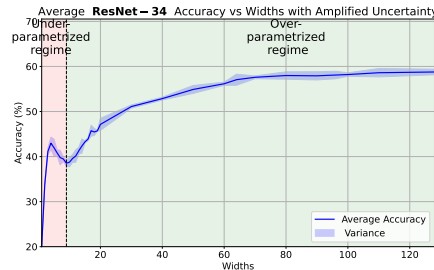

Figure 12: Accuracy versus model width. Experiments performed on CNN, ResNet-34 with CIFAR10 (left) and CIFAR100 (right) as ID datasets.

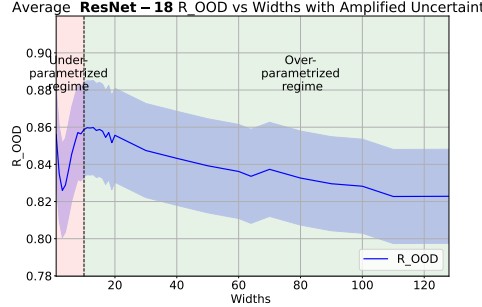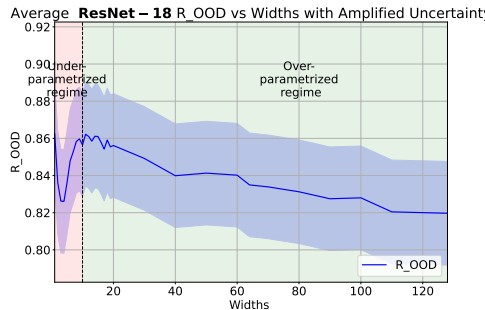

Figure 13: OOD risk metric versus model width. Experiments performed on ResNet-18 with CIFAR10 as ID dataset and CIFAR100 (left) and ImageNet-O (right) as OOD datasets.

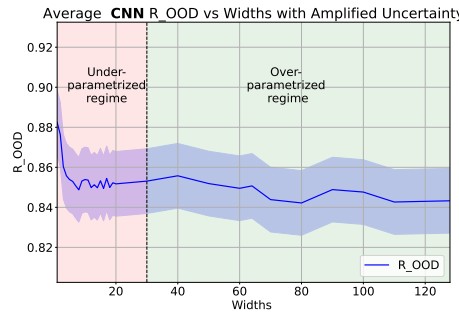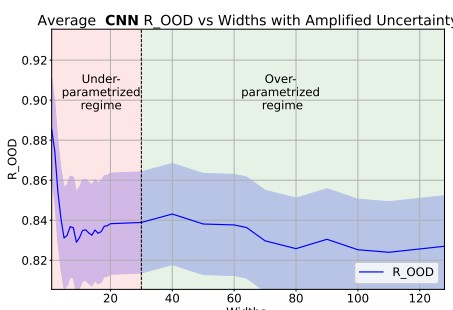

Figure 14: OOD risk metric versus model width. Experiments performed on CNN with CIFAR10 as ID dataset and iNaturalist (left) and Textures (right) as OOD datasets.

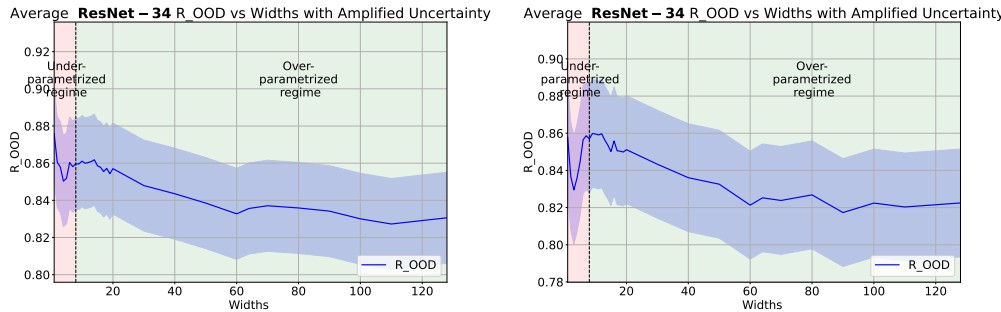

Figure 15: OOD risk metric versus model width. Experiments performed on ResNet-34 with CIFAR10 as ID dataset and iNaturalist (left) and ImageNet-O (right) as OOD datasets.

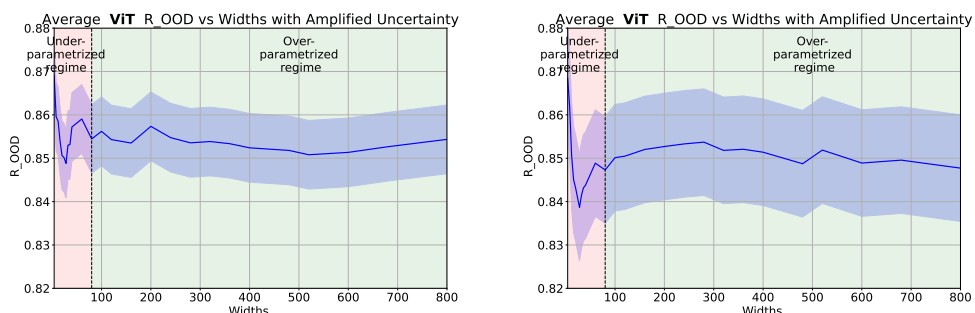

Figure 16: OOD risk metric versus model width. Experiments performed on ViT with CIFAR10 as ID dataset and Places365 (left) and ImageNet-O (right) as OOD datasets.

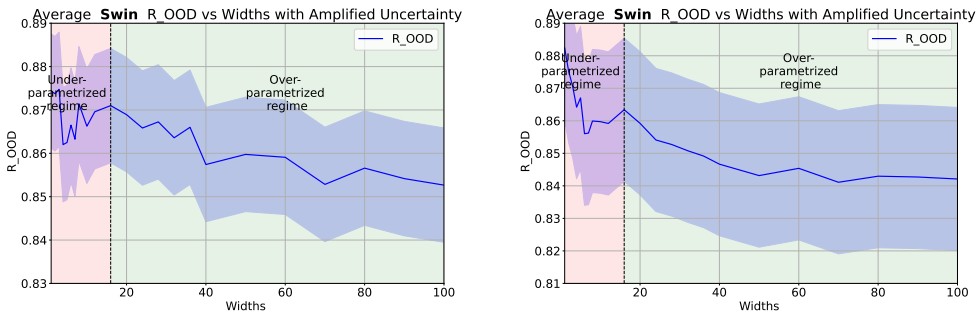

Figure 17: OOD risk metric versus model width. Experiments performed on Swin with CIFAR10 as ID dataset and SUN (left) and ImageNet-O (right) as OOD datasets.

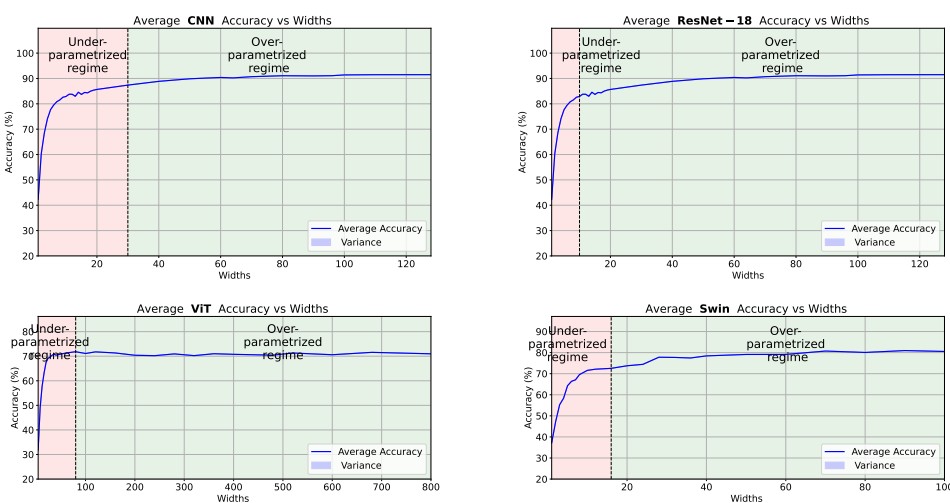

Figure 18: Accuracy versus model width. Experiments performed on CNN (top left), ResNet-18 (top right), ViT (bottom left) and Swin (bottom right) with CIFAR10 as ID dataset in the noiseless setting.

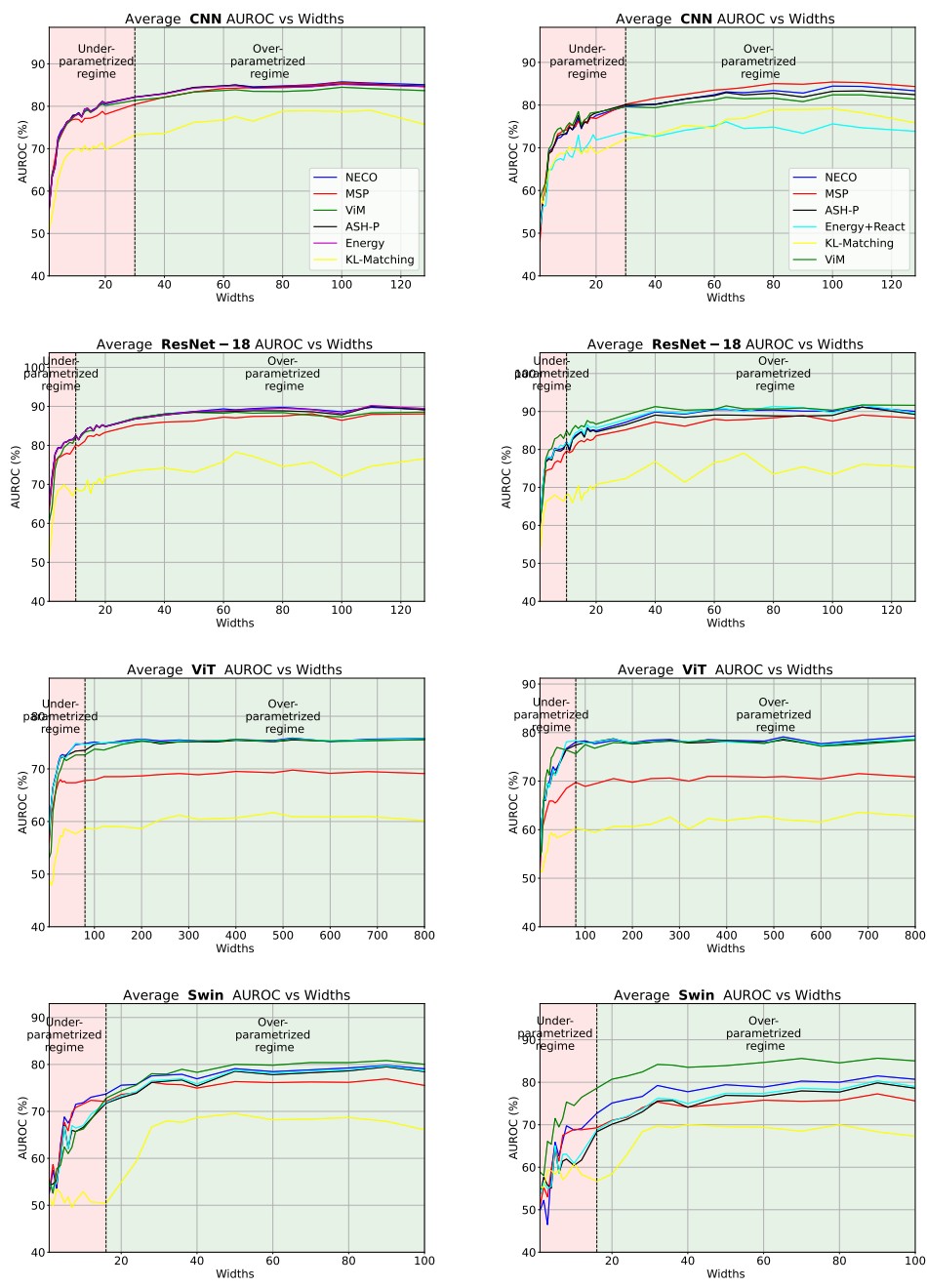

Figure 19: OOD detection (`AUC`) metric versus model width. Experiments performed on CNN, ResNet-18, ViT and Swin with CIFAR10 as ID dataset and CIFAR-100 (left) and ImageNet-O as (right) OOD datasets in the noiseless setting.

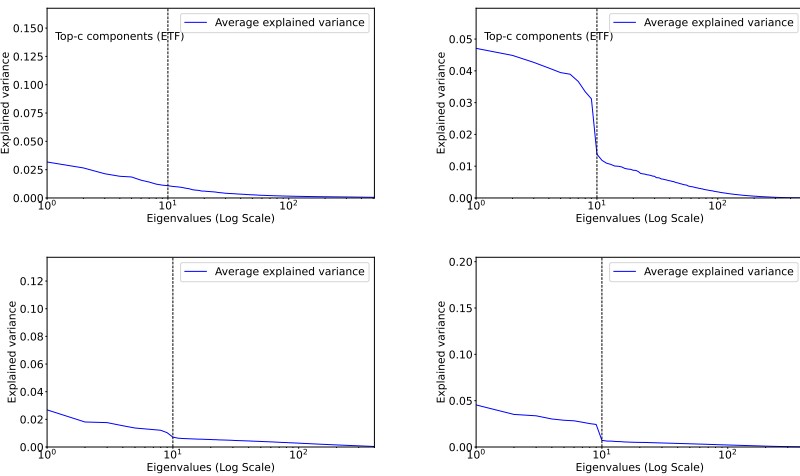

Figure 20: Explained variance versus eigenvalues in the overparametrized region for CNN (width 64), ResNet-18 (width 64), ViT (width 400) and Swin (width 50) with CIFAR-10 as ID dataset. Black line depicts the $10^{th}$ eigenvalue.

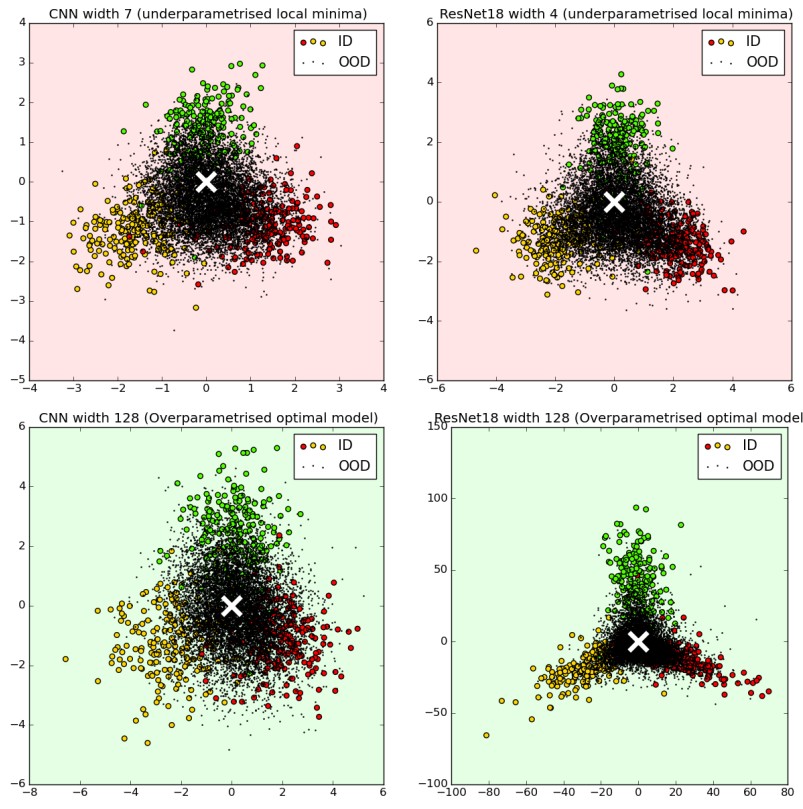

Figure 21: Visualization of the last-layer activations on the test set for ResNet and CNN in the underparametrized local minima and the overparametrized width 128 model, with cifar10 as ID and cifar100 as OOD datasets. ID points are shown in colors and OOD in black. ResNet underparamterized (top right), ResNet overparametrized (bottom right), CNN underparametrized (top left) CNN overparametrized (bottom left).

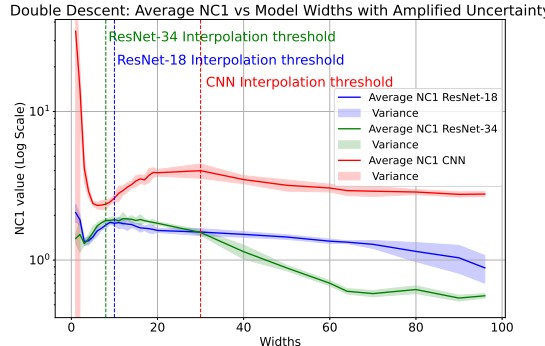

Figure 22: NC1 metrics evolution (Log scale) versus model width. ResNet-18 is shown in blue, ResNet-34 in green CNN in red. Dashed lines represent the interpolation thresholds for each model with matching color.

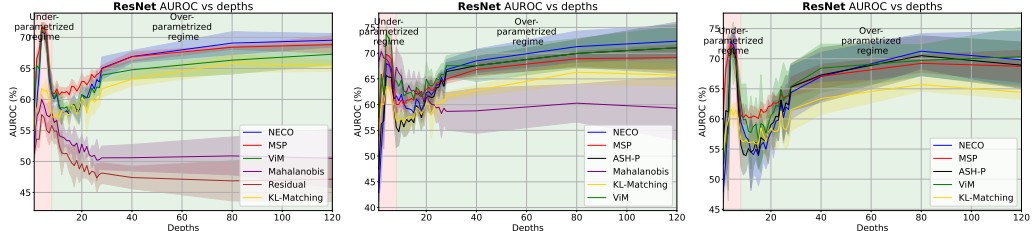

Figure 23: OOD detection (`AUC`) metric versus **model depth** (total number of residual layers across four blocks), with width fixed to 10. Experiments performed ResNet type models, with CIFAR10 as ID dataset and CIFAR100 (left), ImageNet-o (middle), and SUN (right) as OOD dataset.

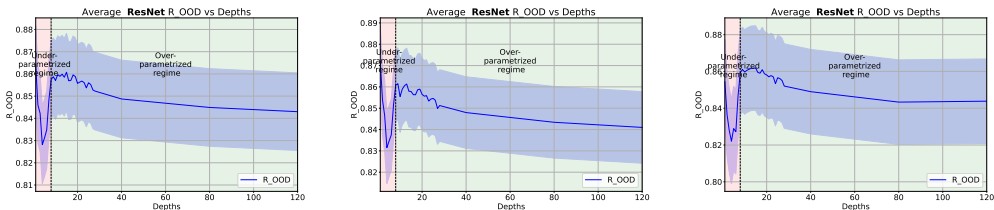

Figure 24: OOD risk metric versus model **depth**. Experiments performed on a ResNet with CIFAR10 as ID dataset and CIFAR100 (left), ImageNet-o (middle), and SUN (right) as OOD datasets.

