# OpenReview forum: "Double Descent Meets Out-of-Distribution Detection: Theoretical Insights and Empirical Analysis on the Role of Model Complexity"
_NeurIPS.cc/2025/Conference — NeurIPS 2025 poster_

### Official Review · Reviewer_rex2 · 2025-06-17

**Clarity:** 3
**Significance:** 3
**Originality:** 4
**Rating:** 5
**Confidence:** 3

**Summary:**

The authors analyze the **Double Descent** phenomenon in post-hoc out-of-distribution (OOD) detection.

They define a proxy risk that measures:
1) the performance of the predictor on in-distribution (ID) data, and
2) how close to ambiguous prediction the model is on OOD data.

They theoretically analyze this risk in a simplified learning setup and derive bounds that suggest the existence of the double descent phenomenon. The authors then observe the phenomenon on a toy example.

When noise is added to labels the authors also observe the phenomenon on real world data.

**Questions:**

1) line 148: You refer to the vector $\mathbb{y}$ of targets as "vector of target probabilities". As this vector is generated by a random process with Gaussian noise, the values of $y_i$ are not bounded and may not lie in [0,1]. How do you deal with this?


2) Eq. 4: The provided coefficients $\hat w_\mathcal{T}$ would minimize the risk for the case of $\phi(x) = x$, but not otherwise. Even though the authors later assume that $\phi$ is monotonic, and thus has an inversion $\phi^{-1}$, this does not allow us to minimize the risk in such a simple manner (a natural idea would be to compute $\phi^{-1}(y)$, but this generally leads to a different optimization result for $w$). Could you please clarify the formulation? In the current form is seems incorrect to claim that Eq. 4 minimizes the empirical risk.


3) lines 321-333: You propose that the neural collapse might indicate quality of OOD detection. Could the same results not be observed if the OOD detection capability was just strongly correlated with ID accuracy, without the need for neural collapse? Could you, please, comment on this?

**Ethical Concerns:**

["NO or VERY MINOR ethics concerns only"]

**Final Justification:**

The authors have successfully addressed my main concerns in the rebuttal, particularly the critical issue regarding Section 4.1. The explanation of the typo in Equation 4 resolves my concern and makes the theoretical analysis logically sound.

All the strong points that were brought up in my initial review still stand.

**Limitations:**

Yes.

**Paper Formatting Concerns:**

No concerns.

**Quality:**

4

**Strengths And Weaknesses:**

# Notes
- Exceptionally well written and easy to understand introduction, related work and preliminaries.
- lines 140-155:
        - line 148: You refer to the vector $\mathbb{y}$ of targets as "vector of target probabilities". As this vector is generated by a random process with Gaussian noise, the values of $y_i$ are not bounded and may not lie in [0,1]
        - line 152: You write "... To analytically solve equation 3, we assume ...". Eq. 3 does not define a problem, so technically, there is nothing to solve there; however, I understand the intent to minimize the risk w.r.t. the weights.
        - Eq. 4: The provided coefficients $\hat w_\mathcal{T}$ would minimize the risk for the case of $\phi(x) = x$, but not otherwise. Even though the authors later assume that $\phi$ is monotonic, and thus has an inversion $\phi^{-1}$, this does not allow us to minimize the risk in such a simple manner (a natural idea would be to compute $\phi^{-1}(y)$, but this generally leads to a different optimization result for $w$).
                - I see this as the biggest detriment to the theoretical analysis. The theoretical bounds derivations seem correct and are a non-trivial modification of Belkin et al. (2020). However, they stand upon the assumption that the weights are set as in Eq. 4, which is in my opinion incorrect.
- lines 159-167: The definition of a proxy response $z(x)$ seems unintuitive to me. The proposed OOD risk can be (up to a constant multiplier) be instead written with the term $(\hat f(x) - f^{OOD}(x) - \frac{\epsilon}{2})^2$ inside the expectations. What is the motivation for defining $z(x)$?
- Figure 2: Just a minor point, but the X-axis is not labeled. It is however described in the caption
- Table 1: "... underparametrized local minima ..."
        - There are often two local minima, at width = 0 and width ~= the interpolation threshold. Is this a typo that should say maxima? If not, which minimum is meant here?
- lines 321-333: You propose that the neural collapse might indicate quality of OOD detection. Could the same results not be observed if the OOD detection capability was just strongly correlated with ID accuracy, without the need for neural collapse? Could you, please, comment on this?

# Strengths:
- Novelty
- Very well written
- Convincing empirical results
- Theoretical analysis of a simplified setup; though based on previous work, the modification is non-trivial

# Weaknesses:
- (Resolved in the rebuttal - typo in original manuscript) The definition of the binary classifier in Section 4.1 is not clear, specifically, the weights $w$ do not seem to minimize the defined empirical risk
- The definition of the noisy response z(x) seems unnecessary
- (Resolved in the rebuttal - typo in original manuscript)The derivations of the OOD risk bounds require the specific setting of the weights $w$ by the pseudo inverse

# Comment on the rating:
I am torn on this paper. It is very well written, the idea to analyze double descent in context of OOD detection is novel, the topic is interesting, and the empirical results are nice. However, the derivation of the bounds is based on Eq. 4, with which I have issues (All of which were resolved in the rebuttal). It is not clear to me why the weights would be set in such a manner. My issues with the paper are limited to Section 4.1. If the authors can address my concerns with Eq. 4 I am willing to increase my rating. In its current form, I believe that the paper would be stronger without the theoretical analysis and simply as an empirical study.

---

> ### Author Rebuttal · Authors · 2025-07-30
>
> Dear reviewer rex2,
>
> We thank you for the thorough review and your encouraging comments.
>
> As highlighted in the review, our paper provides a **comprehensive empirical evaluation** across a **wide range of architectures** (CNNs, ResNets, ViTs, Swin) and ***post-hoc* OOD detection methods**. We want to highlight that the **major contribution** of our study lies in the **empirical observations of a double descent phenomenon in *post-hoc* OOD detection**. To the best of our knowledge, the double descent phenomenon has not been observed before in the OOD context. Our experiments depict how model complexity affects *post-hoc* OOD detection methods and exhibit, for the first time, a double descent phenomenon in OOD detection. Furthermore, we observe that *post-hoc* OOD detection methods may perform less effectively on overparameterized models compared to underparameterized ones. While the paper also includes a **theoretical part**, this part provides **insights** rather than an exhaustive analysis. In particular, this part depicts a **first attempt to mathematically study** the **double descent phenomenon in *post-hoc* OOD detection** methods based on confidence through a simplified model.
>
> ### A.1 & A.2 Least-Squares Binary Classifier Definition (Q1 & Q2):
>
> Our theoretical insights show that the expected OOD risk for least-squares binary classifiers on Gaussian covariate models under some assumptions exhibits an **infinite peak** when the **number of parameters is close to the number of data**, which we associate with the double descent phenomenon. To establish this result, we derive a lower bound $c(n,p,\sigma)$ for the expected OOD risk, and we show that $c(n,p,\sigma)=\infty$ when $n-1 \leq p \leq n+1$. $c(n, p, \sigma)$ describes $E_X[|| \hat w - w^\text{OOD}||^2]$.
>
> The confusion surrounding questions Q1 and Q2 arises from a typo in the definition of the least-squares binary classifier in equation 4. As correctly noted in the review, the least-squares binary classifier minimizes the logit-based expected risk rather than the probability-based expected risk. Therefore, in the definition of $\hat w_\mathcal{T}$ in equation 4, $y$ should be replaced with $u=X w^\text{OOD}$, which is the vector of logits of the optimal classifier. Note that we considered the logit-based definition of $\hat w_\mathcal{T}$ in the proofs, e.g., equation 12 (l.139). While the least-binary classifier minimizes the logits-based expected risk, **Theorem 1 focuses on the probability-based expected OOD risk**. The objective of this theorem is to show that probability-based *post-hoc* OOD detection methods may exhibit a double descent behavior. Without loss of generality, note also that the Gaussian noise could be added to the logit vector $u$. Accordingly, we will update definitions 4 and 5. We sincerely thank you for bringing this element to our attention. Corrections will be made promptly to eliminate further misinterpretation.
>
> We emphasize that this work focuses primarily on **empirical investigation** and practical analysis rather than theoretical derivation. Its core contribution lies in empirically analyzing the relationship between model complexity and *post-hoc* OOD detection methods. The major contribution stems from the empirical observation that *post-hoc* OOD detection methods also exhibit a double descent phenomenon. While the paper includes theoretical elements, these serve as **insights** with an illustrative example rather than exhaustive analysis. They depict a **first attempt** to mathematically study the double descent phenomenon in *post-hoc* OOD detection method through a **simplified model**. As mentioned in the review, although the theoretical framework is constrained to a particular model, the derivation of Theorem 1 is a non-trivial extension of Theorem 1 in [1]. We hope that future works could extend this analysis to more complex scenarios, such as linear models with random features, gradient descent-based algorithms, or sub-gaussian distributions. We believe these theoretical extensions fall outside our current scope, and we reserve these explorations for future research.
>
> ### A.3 Correlations between ID Accuracy and OOD Detection (Q3):
>
> We would like to clarify that we did not claim a direct link between NC and OOD quality. Rather, we explained lines 318-321 that we use NC as a tool to describe the structure of the final-layer features. This representation helped us gain a better understanding of the quality of OOD detection, as we showed empirically, lines 327-333. Also, while OOD detection performance and ID accuracy may be positively correlated, prior works show that this correlation is not universally consistent [1, 2]. For instance, Kim et al. [1] showed that ResNet-50 v1 outperforms its v2 counterpart on OOD benchmarks despite having lower in-distribution accuracy (76% vs. 80%). [2] and [3] observed a similar trend. These results align with our empirical findings. For example, ViT demonstrates lower ID accuracy in overparameterized regimes but achieves superior OOD detection performance using NECO and ViM methods in Figure 2. Furthermore, variations between ID accuracy and OOD detection performance may not show a consistent correlation. For instance, feature-based OOD methods do not exhibit a double descent phenomenon consistently, while the ID accuracy describes a double descent curve. To better illustrate this, the Table below compares OOD detection performance (for logit- and a hybrid-based methods) vs accuracy/NC ratios like in Table 1. The accuracy ratio is defined as the local maximum accuracy in the over-parameterized regime divided by the local maximum in the under-parameterized regime. It follows that value greater than 1 indicate improved accuracy with over-parametrization. We highlight in bold the ratio if its value correlates with the OOD detection performance improvement or degradation. As illustrated, the accuracy ratio fails to provide information on OOD detection performance in the under- and over-parameterized regimes.
>
> |Model|Method|Acc ratio/NC ratio|SUN(U/O)|Places365(U/O)|CIFAR100(U/O)|
> |-|-|-|-|-|-|
> |CNN|MaxLogit|0.99/**0.88**|72.98/60.13|73.33/61.25|73.38/70.37|
> |CNN|ASH-P||68.60/60.36|69.35/61.48|71.11/70.45|
> |ResNet-18|MaxLogit|**1.08**/**1.96**|70.64/72.51|70.69/72.64|69.76/73.65|
> |ResNet-18|ASH-P||69.11/71.73|69.19/71.85|67.58/72.89|
> |Swin|MaxLogit|**1.07**/**1.70**|59.91/70.75|59.84/70.46|61.79/66.95|
> |Swin|ASH-P||59.49/70.74|58.15/70.42|55.10/66.89|
> |ViT|MaxLogit|0.89/**2.32**|66.09/70.30|66.13/69.79|64.60/66.69|
> |ViT|ASH-P||64.79/70.27|64.86/69.79|63.0866.61|
> |ResNet-Depth|MaxLogit|**0.98**/**0.97**|72.57/71.00|72.56/70.76|70.66/70.82|
> |ResNet-Depth|ASH-P||71.79/70.50|71.79/70.19|69.40/70.34|
>
> ### A.4 Typos
>
> Regarding other feedback:
> - We will add the x-axis label in Figure 2.
> - We will change the minima to maxima in Table 1.
> - $z(x)$ depicts predictions rescaled within the range [-1, 1]. We believe this formulation helps in better understanding the expected OOD risk definition, particularly concerning Remark 4.1 and Remark 4.2.
>
> We would like to thank you again for your thoughtful review.
>
> ### References
>
> [1] Kim, C., Singh, S., & Kolter, J. Z. Investigation of Out-of-Distribution Detection Across Various Models and Training Methodologies. Neural Networks, Volume 172, 2024, Pages 1–12.
>
> [2] Mueller, J., Zhu, Y., & Liang, P. Mahalanobis++: Improving OOD Detection via Feature Normalization. Proceedings of the 42nd International Conference on Machine Learning (ICML), 2025.
>
> [3] Yang, Jingkang, et al. "Openood: Benchmarking generalized out-of-distribution detection." Advances in Neural Information Processing Systems 35 (2022): 32598-32611.
>
> [4] Ammar, M. B., Belkhir, N., Popescu, S., Manzanera, A., and Franchi, G. NECO: NEural Collapse Based Out-of-distribution detection. In The Twelfth International Conference on Learning
> Representations, 2024.
>
> [5] Ming, Y., Sun, Y., Dia, O., and Li, Y. How to Exploit Hyperspherical Embeddings for Out-of-Distribution Detection. International Conference on Learning Representations,2023.
>
> [6] Haas, J., Yolland, W., and Rabus, B. T. Linking Neural Collapse and L2 Normalization with Improved Out-of-Distribution Detection in Deep Neural Networks. Transactions on Machine Learning Research, 2023. ISSN 2835-8856.

---

> > ### Comment · Reviewer_rex2 · 2025-08-04
> >
> > I thank the authors for the detailed rebuttal. They have successfully addressed my main concerns, particularly the critical issue regarding Section 4.1. The explanation of the typo in Equation 4 resolves my concern and makes the theoretical analysis logically sound.
> >
> > I recommend acceptance.
> >
> > A minor note on clarity: The proxy response $z(x)$ made the OOD risk definition less intuitive to me.
> > For the final version, I recommend the authors re-consider whether direct formulation may not enhance readability. Perhaps have colleagues read the paper and see if they understand it at a glance.

---

### Official Review · Reviewer_7SLJ · 2025-06-30

**Clarity:** 2
**Significance:** 2
**Originality:** 2
**Rating:** 4
**Confidence:** 3

**Summary:**

This paper mainly investigates the relationship between model capacity and the post-hoc out-of-distribution (OOD) detection, with a particular focus on whether the "double descent" phenomenon exists. The paper empirically demonstrates the double descent phenomenon in OOD detection in CNN and Transformer architectures, and provides an analysis based on random matrix theory.

**Questions:**

Same as the weakness mentioned above.

If the authors could provide a clearer explanation or point out if I have any misunderstandings about the OOD risk, I would be willing to increase my score.

There are additional questions:
1. In Table 1, could the authors report the NC1 metric for both the underparameterized and overparameterized regimes for each architecture, rather than just their ratio? This would make it easier to compare neural collapse across different architectures.
2. One issue regarding overparameterization theory is whether there is any theoretical understanding of the impact of the number of parameters across different architectures. It is generally believed that transformers have more parameters than ResNets, but they do not necessarily achieve better OOD detection performance, even in the overparameterized regime. Is there any theoretical explanation for this?

**Ethical Concerns:**

["NO or VERY MINOR ethics concerns only"]

**Final Justification:**

The rebuttal addressed most of my concerns, particularly clarifying the NC1 analysis. The only remaining point is that the current definition of OOD risk may be somewhat simplified, though it still reasonably captures key intuitions. I will keep my score and recommend acceptance of the paper.

**Limitations:**

Yes

**Quality:**

2

**Strengths And Weaknesses:**

**Strengths:**
1. The authors conducted extensive experiments, making comprehensive comparisons of several post-hoc OOD detection algorithms across four different network architectures: CNN, ResNet, ViT, and Swin Transformer.
2. The authors’ analysis of the experiments is well-organized and easy to follow. In particular, they discovered different behaviors between logit-based and feature-based OOD detection algorithms in the overparameterized regime, which offers some insights for future research.
3. The overparameterized ResNet is effective in detecting OOD samples, providing a strong backbone for further OOD research.
4. The authors effectively explain why CNNs tend to perform worse in the overparameterized regime, while other architectures can achieve better results, by leveraging Neural Collapse theory.

**Weaknesses:**

The main issue lies in the authors’ definition of OOD risk. The OOD risk, as defined by the authors, is very similar to the risk associated with samples that lie between two classes in a binary classification problem. However, OOD samples are not limited to this type. For example, in a binary Gaussian mixture model, samples that are far away from the decision boundary can also be considered OOD. Therefore, I believe the authors’ definition of OOD risk is quite limited.

---

> ### Author Rebuttal · Authors · 2025-07-30
>
> Dear reviewer 7SLJ,
>
> We would like to thank you for your interest and for appreciating both the novelty and experimental findings of our study.
>
> ### A.1 Expected OOD Risk (Q1):
>
> We acknowledge that out-of-distribution (OOD) samples arise in many forms, including those that are far from the in-distribution (ID) support or decision boundary. Our definition of the expected OOD risk draws inspiration from the OpenOOD benchmark [1], which categorizes semantic shifts by assigning a binary score (1 for ID data, 0 for OOD data) through an unsupervised approach. Drawing inspiration from both this framework and traditional expected risk, we propose the **expected OOD risk to evaluate whether confidence predictions from a binary classifier $f(\cdot)$ can be used to define an OOD scoring function**. In particular, the expected OOD risk measures how far the predictions are from an idealized confidence function $f^\text{OOD}(\cdot)$, which satisfies:
> $f^{\text{OOD}} (x) \approx 0$ or $1$ and is aligned with the probabilities of a perfect classifier when $x$ is ID;
> $f^{\text{OOD}}(x) \approx 0.5$ when $x$ is OOD and depicts maximal uncertainty.
>
> The greater the deviation from this ideal classifier, the higher the expected OOD risk. **We emphasize that the expected OOD risk is not a direct OOD detection metric but evaluates whether a classifier's confidence predictions can be effectively used in *post-hoc* OOD detection methods**, such as Maximum Softmax Probability (MSP) or MaxLogit approaches [2].
>
> ### A.2 NC1 metric in Under- Over-parameterized Regimes (Q2):
>
> We agree that reporting the NC1 metrics alongside their corresponding ratios would enhance clarity. While we plan to incorporate these values into the final version, we can already share their values in the table below. Note that we have also added values for the depth-wise experiments as recommended by Reviewer ekWE.
>
> | Architecture | NC1 (Under-parameterized) | NC1 (Over-parameterized) |
> |--------------|---------------------------|---------------------------|
> | ResNet-18    | 1.35                      | 0.687                     |
> | ResNet-34    |          1.29           |         0.64           |
> | CNN          | 2.37                      | 2.71                      |
> | Swin         | 35.08                     | 20.65                     |
> | ViT          | 29.52                     | 12.71                     |
> | ResNet (Depth)         |         1.53             |    1.57                  |
>
> ### A.3 Overparameterization across Architectures (Q3):
>
> Transformer architectures may indeed have more parameters than conventional neural networks due to their attention mechanisms. However, increasing attention parameters doesn't automatically lead to superior performance, and the effects of overparameterization in transformers remain poorly understood. For instance, recent studies have shown that these architectures may suffer from attention rank collapse, i.e., a phenomenon where all tokens converge to a single representation with wider or deeper architectures [3-6]. Other works observed that the norm of the residual stream grows exponentially along the layers over the forward pass of multiple transformer blocks [7-9]. [10] shows that such behavior may hurt *post-hoc* OOD detection.
>
> In our study, we have decided to increase the model complexity of transformers by increasing the width of the MLP in the transformer block, following a similar strategy to that described in [11]. Note also that in our experiments, the number of parameters is not consistent and varies significantly across architectures. Additionally, the training procedure differs for each architecture, which may also impact OOD detection [12].
>
> ### References
>
> [1] Yang, Jingkang, et al. "Openood: Benchmarking generalized out-of-distribution detection." Advances in Neural Information Processing Systems 35 (2022): 32598-32611.
>
> [2] Hendrycks, Dan, and Kevin Gimpel. "A Baseline for Detecting Misclassified and Out-of-Distribution Examples in Neural Networks." International Conference on Learning Representations. 2017.
>
> [3] Dong, Yihe, Jean-Baptiste Cordonnier, and Andreas Loukas. "Attention is not all you need: Pure attention loses rank doubly exponentially with depth." International conference on machine learning. PMLR, 2021.
>
> [4] Noci, Lorenzo, et al. "Signal propagation in transformers: Theoretical perspectives and the role of rank collapse." Advances in Neural Information Processing Systems 35 (2022): 27198-27211.
>
> [5] Saada, Thiziri Nait, Alireza Naderi, and Jared Tanner. "Mind the Gap: a Spectral Analysis of Rank Collapse and Signal Propagation in Attention Layers." arXiv preprint arXiv:2410.07799 (2024).
>
> [6] Geshkovski, Borjan, et al. "A mathematical perspective on transformers." Bulletin of the American Mathematical Society 62.3 (2025): 427-479.
>
> [7] Wang, Haoqi, Tong Zhang, and Mathieu Salzmann. "Demystifying Singular Defects in Large Language Models." Forty-second International Conference on Machine Learning.
>
> [8] Heimersheim, Stefan, and Alex Turner. "Residual stream norms grow exponentially over the forward pass." AI Alignment Forum. 2023.
>
> [9] Merrill, William, et al. "Effects of Parameter Norm Growth During Transformer Training: Inductive Bias from Gradient Descent." Proceedings of the 2021 Conference on Empirical Methods in Natural Language Processing. 2021.
>
> [10] Mueller, J., Zhu, Y., & Liang, P. Mahalanobis++: Improving OOD Detection via Feature Normalization. Proceedings of the 42nd International Conference on Machine Learning (ICML), 2025.
>
> [11] Nakkiran, Preetum, et al. "Deep double descent: Where bigger models and more data hurt." Journal of Statistical Mechanics: Theory and Experiment 2021.12 (2021): 124003.
>
> [12] Kim, C., Singh, S., & Kolter, J. Z. Investigation of Out-of-Distribution Detection Across Various Models and Training Methodologies. Neural Networks, Volume 172, 2024, Pages 1–12.

---

> > ### Comment · Reviewer_7SLJ · 2025-08-06
> >
> > I appreciate the authors’ rebuttal, which has addressed many of my concerns. In particular, the analysis of the NC1 metric for under-parameterized and over-parameterized networks across different architectures clarifies my concern that ResNet serves as a stronger backbone for OOD detection. However, I remain skeptical about whether the proposed OOD risk adequately captures our intuition behind OOD samples, given that OOD samples constitute not a single class but a set of diverse classes. Based on this, I will keep my score.

---

> ### Author Response · Authors · 2025-08-06
>
> Thank you for the reply and for appreciating the rebuttal.
>
> We believe there may have been a misunderstanding, and we would like to offer some clarifications. There are two key aspects to address: the **OOD criterion** and the **OOD evaluation metric**.
>
> First, regarding the **OOD criterion**, we follow standard *post-hoc* approaches commonly used in the literature, such as the Maximum Softmax Probability (MSP) [1]. In this setting, we would like to emphasize that our method does not deviate from established practices. In binary classification setups, for a given sample $x$, the MSP score is computed as $\text{MSP}(x)=\max(\hat f(x), 1-\hat f(x))$, where $\hat f(x)$ denotes the predicted probability of a trained classifier $\hat f(\cdot)$. The MSP method effectively performs under two conditions:
> $\hat f(\cdot)$ exhibits a high confidence for IN samples (yielding $\text{MSP}(x) \approx 1$, i.e., $\hat{f}(x) \approx 0$ or $\approx 1$), and
> $\hat f(\cdot)$ depicts low confidence on OOD samples (resulting in $\text{MSP}(x) \approx 0.5$, i.e., $\hat{f}(x) \approx 0.5$).
>
> Second, concerning the **OOD evaluation metrics**, conventional experimental papers use measures such as **AUROC**, **AUPR**, and **FPR@95**. For our insights, the **OOD evaluation metrics** is the **expected OOD risk** as a **first attempt** to mathematically study the double descent phenomenon in post-hoc OOD detection method through a **simplified model**. Additionally, we would like to stress that the concept of expected OOD risk is not presented as a novel contribution in our work. Similar formulations have already been introduced in prior research (e.g., [2]) to support PAC-style generalization bounds for OOD detection.
>
> Thus, both our **OOD risk definition** and our **OOD criterion** are aligned with existing literature and not introduced as new elements. While we acknowledge that multiple classes of OOD examples exist, we believe these variations should be addressed by the choice of OOD criterion itself and not the **OOD risk definition**.
>
> We hope those clarifications have addressed the concerns on the definition of the expected OOD risk raised in the review.
>
> [1] Hendrycks, Dan, and Kevin Gimpel. "A Baseline for Detecting Misclassified and Out-of-Distribution Examples in Neural Networks." International Conference on Learning Representations. 2017.
>
> [2] Fang, Zhen, et al. "On the Learnability of Out-of-distribution Detection." Journal of Machine Learning Research 25.84 (2024): 1-83.

---

> > ### Comment · Reviewer_7SLJ · 2025-08-07
> >
> > Thank you for your further clarifications. I believe we are on the same page regarding the OOD criterion and evaluation, with no major misunderstandings. My only remaining concern is that the current definition of OOD risk—evaluating whether a binary classifier can be used to define an OOD scoring function—may be somewhat simplified. Nevertheless, we acknowledge that this definition still reasonably captures some of our intuitive understanding of out-of-distribution detection.

---

### Official Review · Reviewer_ekWE · 2025-07-02

**Clarity:** 3
**Significance:** 2
**Originality:** 2
**Rating:** 4
**Confidence:** 3

**Summary:**

This paper explores the relationship between model complexity and post-hoc out-of-distribution (OOD) detection, with a focus on the double descent phenomenon. The authors provide theoretical analysis using a simplified Gaussian covariate model and support their findings with broad empirical results across multiple architectures. They demonstrate that OOD detection performance exhibits a double descent pattern with respect to model width, and highlight scenarios where smaller models outperform larger ones. While the theoretical setup is somewhat limited in scope, the paper presents novel observations and thorough experimentation, offering useful insights into model selection for OOD detection.

**Questions:**

* Could you elaborate on the assumptions made about the OOD distribution, particularly in the theoretical analysis?

* Have you investigated whether the double descent phenomenon appears when varying model depth instead of width?

* To what extent do your theoretical insights generalize beyond the Gaussian covariate model and binary classification with least-squares loss?

**Ethical Concerns:**

["NO or VERY MINOR ethics concerns only"]

**Final Justification:**

Thank you to the authors for the rebuttal and for addressing my questions and comments. I am keeping my score based on the response.

Best,
Reviewer

**Limitations:**

Yes

**Paper Formatting Concerns:**

No concerns

**Quality:**

3

**Strengths And Weaknesses:**

Strengths:
The paper offers a novel perspective by connecting the double descent effect with post-hoc out-of-distribution (OOD) detection, an area that has been largely unexplored.

The paper provides a comprehensive empirical evaluation across a wide range of architectures (CNNs, ResNets, ViTs, Swin) and OOD detection methods, enhancing the robustness and generality of its findings.

Weaknesses:
The theoretical framework relies on simplified assumptions—specifically Gaussian input distributions and binary classification using least-squares loss—which may limit its applicability to more realistic or complex settings.

Theoretical novelty appears limited, as Theorem 1 closely mirrors results from Belkin et al. (2020).

[1] Belkin et al., Two models of double descent for weak features, 2020.

---

> ### Author Rebuttal · Authors · 2025-07-30
>
> Dear reviewer ekWE,
>
> We would like to thank you for your interest and for appreciating the novelty of our study.
>
> As highlighted in the review, our paper provides a **comprehensive empirical evaluation** across a **wide range of architectures** (CNNs, ResNets, ViTs, Swin) and ***post-hoc* OOD detection methods**. We want to highlight that the **major contribution** of our study lies in the **empirical observations of a double descent phenomenon in *post-hoc* OOD detection** that had never been observed. Our experiments depict how model complexity affects *post-hoc* OOD detection methods and exhibit, **for the first time**, a **double descent phenomenon in OOD detection**. We would like to emphasize the challenges involved in conducting these experiments with real-world deep networks. Furthermore, we observe that *post-hoc* OOD detection methods may perform less effectively on overparameterized models compared to underparameterized ones. While the paper also includes a **theoretical part**, this part  provides **insights** rather than an exhaustive analysis. In particular, this part depicts a **first attempt to mathematically study** the **double descent phenomenon in *post-hoc* OOD detection** methods based on confidence through a simplified model.
>
> ### A.1 Assumptions on OOD distribution (Q1):
>
> Our theoretical insights show that the expected OOD risk for least-squares binary classifiers on Gaussian covariate models under some assumptions exhibits an **infinite peak** when the **number of parameters is close to the number of data**, which we associate with the double descent phenomenon.
>
> In Theorem 1, the **influence of the OOD distribution** is encapsulated within the **matrix $\Sigma^\text{OOD}$**. Assumptions made in the paper ensure that matrices $\Sigma^\text{OOD}$ and $\Sigma$ are positive-definite, for which we have $\lambda_{\min}(\Sigma)>0$ and $\lambda_{\min}(\Sigma^\text{OOD})>0$. The non-singularities of these matrices ensure that the expected OOD risk is lower bounded by $c(n, p, \sigma)$. Since the expected OOD risk is also upper bounded by $c(n, p, \sigma)$ and that $c(n, p, \sigma)=\infty$ when $n-1 \leq p \leq n+1$, we can deduce from the squeeze theorem that $E_X[R_\text{OOD}(\hat f)]=\infty$ when $n-1 \leq p \leq n+1$. Note that **$c(n, p, \sigma)$** depicts the behavior of $E_X[|| \hat{w} - w^\text{OOD}||^2]$ and **does not depend on the OOD distribution**.
>
> For $p < n-1$ or $p>n+1$, the degree of distributional shift between in- and out-of-distribution scenarios affects how tightly the bounds approximate the expected OOD risk. However, our primary objective was not to provide accurate proxies for the expected OOD risk, but rather to **highlight the existence of an infinite peak when $n$ is close to $p$** that we associate with the double descent phenomenon.
>
> In the experimental part, we focused on **semantic shifts** and considered both **near and far OOD distributions**, i.e., shifts that are close to or distant from the in-distribution (ID). Our results show that both types of semantic shift are similarly affected by the double descent phenomenon, following the same curve (see Figure 2 and Appendix Figures 4-9). Additionally, to the best of our knowledge, we think that quantifying the amount of shift between two shifted distributions is more relevant to covariate shift scenarios linked to domain adaptation as in [5]. In our setting, divergence-based assumptions are less applicable.
>
> ### A.2 Depth versus Width (Q2):
>
> Most works on the double descent phenomenon focus on **varying the network width rather than depth** [1-4]. A primary reason stems from challenges in modifying network depth across empirical and theoretical frameworks. From an empirical perspective, increasing depth often introduces training instabilities and necessitates hyperparameter adjustments (e.g., learning rates) that complicate comparative analyses. From a theoretical perspective, studies typically rely on frameworks like Random Matrix Theory (RMT) or statistical physics, which often model networks as linear systems with random features. Increasing the depth also increases nonlinear interactions and hierarchical dependencies, which may limit the applicability of these mathematical tools.
>
> Nonetheless, **we have recently investigated the influence of the model depth on the double descent phenomenon**. In these experiments, we used ResNet-like models with fixed width and increased the model complexity by varying their depth. We used CIFAR10 as the ID dataset and reused the same OOD datasets from the width experiments. Note that we kept the same training hyperparameters for the study of the depth. Our results depict comparable double descent patterns in both ID generalization and *post-hoc* OOD detection performance (measured via AUC).  We are open to including these figures in the paper. To the best of our knowledge, they represent the **first empirical illustration of depth-wise double descent in realistic models and datasets**. Since we can not add additional figures during the rebuttal, we include a table similar to Table 1 to describe our results. In particular, we report the accuracy results for the underparametrized optimum (depth = 4), the interpolation threshold (depth =  8),  and overparametrized optimum (depth = 120). These results are averaged across five seeds.
>
> **Accuracy (Under / Interpolation / Over):** 77.31% / 66.98% /  76.21%
> **NC1 Ratio (Under / Over):**  $\frac{1.53}{1.57} = 0.97$
>
> | Method      | CIFAR100 (U/O) | SUN (U/O) | iNaturalist (U/O) | ImageNet-O (U/O) | Textures (U/O) | Places365 (U/O) |
> |-------------|----------------|-----------|--------------------|-------------------|----------------|------------------|
> | MSP           | 71.62 / 68.84          |  72.45   / 69.23    |  64.80   /   66.53        |    69.78 /       69.13    |    64.03 /  67.40      |   72.47  / 69.00         |
> | MaxLogit      |  70.66   /    70.82    |   72.57  /   71.00  | 65.18    /  68.73         |    67.37 / 71.42          |   52.79  / 68.56       |    72.56 /    70.76      |
> | Energy        |  69.43   /    70.81    |   71.82  /  70.98   |   64.68  /   68.73        |   65.69  /      71.41     | 53.37    / 68.92       |  71.83   /  70.74        |
> | ReAct         |   70.13 /    70.71    |   71.83  /  71.56   |  64.31   /   66.93        |    66.89 /       71.32    |  55.73   / 69.61       |  71.89   / 71.30        |
> | NECO          | 70.85 / 69.55          |   72.83  /  71.24   |  65.62   /      68.47     |    67.98 / 72.32          |   61.15  / 72.72       |    72.74 /   70.77       |
> | ViM           |   71.14  /   67.29     |    71.86 /    71.32|   65.10  /     64.35      |   73.67  / 71.06          |   79.24  /    75.22    |   71.07  /   69.46       |
> | ASH-P         |   69.40  /  70.34      |   71.79  /   70.50  |  64.99   /      69.35     |   65.71  / 71.05          |    53.37 / 69.19       |   71.79  /   70.19       |
>
>
>
>
> ### A.3 Generalization of the Theoretical Results (Q3):
>
> We emphasize that this work focuses primarily on **empirical investigation** and practical analysis rather than theoretical derivation. Its **core contribution** lies in empirically analyzing the **relationship between model complexity and *post-hoc* OOD detection methods**. The major contribution stems from the empirical observation that ***post-hoc* OOD detection methods** also exhibit a **double descent phenomenon**. While the paper includes theoretical elements, they serve as illustrative tools rather than exhaustive analysis. They depict a first attempt to mathematically study the double descent phenomenon in ***post-hoc* OOD detection** through a simplified model. Although the theoretical framework is constrained to a specific model architecture, the derivation of Theorem 1 is a non-trivial extension of Theorem 1 in [1]. Future works could extend this analysis to more complex scenarios, such as linear models with random features, gradient descent-based algorithms, or sub-Gaussian distributions. We believe these theoretical extensions fall outside our current scope, and we reserve these explorations for future research.
>
> ### References
>
> [1] Belkin, Mikhail, Daniel Hsu, and Ji Xu. "Two models of double descent for weak features." SIAM Journal on Mathematics of Data Science 2.4 (2020): 1167-1180.
>
> [2] Couillet, Romain, and Zhenyu Liao. Random matrix methods for machine learning. Cambridge University Press, 2022.
>
> [3] Bach, F. (2024). High-dimensional analysis of double descent for linear regression with random projections. SIAM Journal on Mathematics of Data Science, 6(1), 26-50.
>
> [4] Nakkiran, Preetum, et al. "Deep double descent: Where bigger models and more data hurt." Journal of Statistical Mechanics: Theory and Experiment 2021.12 (2021): 124003.
>
> [5] Ben-David, S., Blitzer, J., Crammer, K., & Pereira, F. (2007). "Analysis of Representations for Domain Adaptation." NeurIPS 2006.

---

### Official Review · Reviewer_UoHZ · 2025-07-02

**Clarity:** 3
**Significance:** 2
**Originality:** 2
**Rating:** 4
**Confidence:** 3

**Summary:**

This work shows that the double descent phenomenon reported by Belkin et al. (2020) for regression also arises in out-of-distribution detection. They confirmed this behavior experimentally for multiple model architectures, supervised OOD detection techniques, and datasets.

**Questions:**

See Weaknesses.

**Ethical Concerns:**

["NO or VERY MINOR ethics concerns only"]

**Final Justification:**

While the Gaussian data-distribution assumption remains a concern, I have no objection to the claim ”comprehensive empirical evaluation across a wide range of architectures”. And A.2 addressed my concern.

**Limitations:**

Yes, but the Gaussian constraint of the data $X$ should be mentioned.

**Paper Formatting Concerns:**

Nothing.

**Quality:**

3

**Strengths And Weaknesses:**

**Strengths**

- The authors validate their theory not only on toy data but also across multiple standard model architectures, benchmark datasets, and detection techniques.
- Their formulation of Out-of-Distribution Risk is nice.

**Weaknesses**

1. *Gaussian constraint*

The analysis assumes that $X$ is Gaussian. Is this assumption needed for the expectation identities in Lines 961 - 963 in Appendix?
For image data, it implies uncorrelated pixel noise, which differs greatly from realistic images. How might this assumption be relaxed?

2. *Concerns about the distribution of OOD*

This work extends Belkin’s theory for regression to out-of-distribution detection. Unlike regression, OOD detection cannot ignore the manifold hypothesis [1][2]: in-distribution samples $X \in R^D$ lie on a manifold of intrinsic dimension $d << D$, and OOD samples are viewed as lying off that manifold.
However, the paper assumes  $X$ is Gaussian in $R^D$,  implying that the in-distribution supports the full ambient space with no manifold structure. In this setting, how is the OOD distribution defined? Can the proposed theory be reconciled with manifold-based arguments?
Moreover, if $X$ truly lies on a lower-dimensional manifold, then $\Sigma$ and $\Sigma^{OOD}$ would be rank-deficient; does that affect the interpretation of Theorem 1?

- [1] Ross, Brendan Leigh, et al. "A Geometric Framework for Understanding Memorization in Generative Models." ICLR 2025.
- [2] Kamkari, Hamidreza, et al. "A geometric explanation of the likelihood OOD detection paradox." ICML 2024.

3. *The degree of the distribution divergence*

In OOD detection, the divergence between in-distribution and OOD data distribution is as critical to task difficulty as the training sample size $n$ and model parameter count $p$, yet the proposed model neither defines the OOD distribution nor provides a parameter to control this divergence.

4. *CIFAR-10/100 experimental setup is not clearly described*

 (my apologies if they are elsewhere) The paper calls the method post-hoc; does this imply the detector’s classifier is trained solely on in-distribution data? Were OOD samples ever used in training? If so, which ones and in what quantity relative to in-distribution samples? Since the in-distribution to OOD sample ratio likely impacts detection results, is this captured in the proposed theoretical framework?

----
The above Points 2 and 3 arise specifically because the theory is applied to OOD detection.
The reviewer remains unclear why this work selected OOD detection rather than classification as Belkin’s extension; clarifying this rationale would strengthen the paper.
The reviewer rated this submission as a borderline reject, but there is potential to improve its score.

---

> ### Author Rebuttal · Authors · 2025-07-30
>
> Dear reviewer UoHZ,
>
> We would like to thank you for your interest and for appreciating both the novelty and the experimental findings of our study.
>
> As highlighted in the review, our paper provides a **comprehensive empirical evaluation** across a **wide range of architectures** (CNNs, ResNets, ViTs, Swin) and ***post-hoc* OOD detection methods**. We want to highlight that the **major contribution** of our study lies in the **empirical observations of a double descent phenomenon in *post-hoc* OOD detection**. To the best of our knowledge, the double descent phenomenon has not been observed before in the OOD context. Our experiments depict how model complexity affects *post-hoc* OOD detection methods and exhibit, for the first time, a double descent phenomenon in OOD detection. Furthermore, we observe that *post-hoc* OOD detection methods may perform less effectively on overparameterized models compared to underparameterized ones. The **theoretical section** in our paper should be seen as **insights** with an illustrative example of a toy distribution, and not as an exhaustive analysis. However, this part depicts a **first attempt to mathematically study** the **double descent phenomenon in *post-hoc* OOD detection** methods based on confidence through a simplified model.
>
> ### A.1 Gaussian Constraints & Assumptions on OOD Distribution (Q1+Q2+Q3):
>
> Our theoretical insights show that the expected OOD risk for least-squares binary classifiers on Gaussian covariate models under some assumptions exhibits an **infinite peak** when the **number of parameters is close to the number of data**, which we associate with the double descent phenomenon.
>
> In this work, we adopt the conventional OOD framework, where we consider **two distinct distributions**: an IN distribution $P_{\mathcal{X}, \mathcal{Y}}$ from which training samples are drawn and a separate OOD distribution $P_{\mathcal{X}, \mathcal{Y}}^\text{OOD}$. In Theorem 1, the **influence of the OOD distribution** is encapsulated within the **matrix $\Sigma^\text{OOD}$**. Assumptions made in the paper ensure that matrices $\Sigma^\text{OOD}$ and $\Sigma$ are positive-definite, for which we have $\lambda_{\min}(\Sigma)>0$ and $\lambda_{\min}(\Sigma^\text{OOD})>0$. While the choice of the IN and OOD distributions may induce a manifold structure, **we do not need to leverage the manifold structure** to show the non-singularities of $\Sigma^\text{OOD}$ or $\Sigma$. Indeed, the matrix $\Sigma^\text{OOD}$ is positive-definite if we have $u^T\Sigma^\text{OOD}u>0$ for all non-zeros $u \in \mathbb{R}^d$. Let $u \in \mathbb{R}^d$. From the mean-value theorem and since $\phi(\cdot)$ is differentiable and positive, we have
>
> $$u^T \Sigma^\text{OOD} u=u^T E_{(x, \cdot) \sim P_{\mathcal{X}, \mathcal{Y}}^\text{OOD}} \biggl[\Bigl(\tfrac{\phi(x^T \hat w)-\phi(x^T w^\text{OOD})}{x^T \hat w-x^T w^\text{OOD}}\Bigr)^2 x x^T \biggr] u = \int_{\mathbb{R}^d}\Bigl(\tfrac{\phi(x^T \hat w)-\phi(x^T w^\text{OOD})}{x^T \hat w-x^T w^\text{OOD}}\Bigr)^2 (x^T u)^2 \mu^\text{OOD}(x)  dx >0.$$
>
> The non-singularities of these matrices ensure that the expected OOD risk is lower bounded by $c(n, p, \sigma)$. Since the expected OOD risk is also upper bounded by $c(n, p, \sigma)$ and that $c(n, p, \sigma)=\infty$ when $n-1 \leq p \leq n+1$, we can deduce from the squeeze theorem that $E_X[R_\text{OOD}(\hat f)]=\infty$ when $n-1 \leq p \leq n+1$. Note that **$c(n, p, \sigma)$** depicts the behavior of $E_X[|| \hat w - w^\text{OOD}||^2]$ and **does not depend on the OOD distribution**. As noted in the review, we derive $c(n, p, \sigma)=\infty$ by using the **Gaussian assumption on $X$** and Wishart distribution properties. Note that the Gaussian constraint on $X$ is often mentioned in the paper and is explicitly defined in l.141-147.
>
> For $p < n-1$ or $p>n+1$, the degree of distributional shift between in- and out-of-distribution scenarios affects how tightly the bounds approximate the expected OOD risk. However, our primary objective was not to provide accurate proxies for the expected OOD risk, but rather to **highlight the existence of an infinite peak when $n$ is close to $p$** that we associate with the double descent phenomenon. **This explains why we do not consider assumptions on divergences between IN/OOD distributions**. We hope those clarifications have addressed the concerns raised in the review.
>
> ### A.2 Ambiguity in CIFAR-10/100 Experiments (Q4):
>
> As stated in the paper (l.40-43), our study considers ***post-hoc*  OOD detection methods**.  *Post-hoc* OOD detection methods operate on top of a pretrained classifier after the model’s training process is finished and leverage internal model properties (probabilities, logits, or features) to define a scoring function measuring the prediction confidence. All classifiers are **trained exclusively on in-distribution** (ID) data, i.e., **without using OOD samples during training**. The OOD samples are only used to measure the performance of *post-hoc* OOD detection methods with the AUC and FPR@95.

---

### Official Review · Reviewer_RJrF · 2025-07-20

**Clarity:** 2
**Significance:** 3
**Originality:** 3
**Rating:** 5
**Confidence:** 2

**Summary:**

This paper presents both a theoretical study of and empirical results for the double descent phenomenon in OOD detection.  Using simplifying assumptions such as a linear classifier, monotonic activations and Gaussian embedding, the authors show that as network caapacity increases, a double descent in loss occurs.  They demonstrate the phenomenon on mutiple architectures with many different OOD detection methods and show that the second descent is sometimes beneficial and sometimes overfits leading to reduced performance.

**Questions:**

How is one to know what to do in a particular OOD application given this result?  How is the theorem of value or use to practicioners?

**Ethical Concerns:**

["NO or VERY MINOR ethics concerns only"]

**Final Justification:**

A strong theoretical paper, that allow not very accessible, makes interesting points about double descent for OOD.  Worthwhile.

**Limitations:**

It is not clear why the double descent phenomenon is important for OOD detection.

**Paper Formatting Concerns:**

None.

**Quality:**

3

**Strengths And Weaknesses:**

A strong theoretical underpinning for a useful observation about OOD training.  Very detailed results, and proofs seem to be well developed.

Difficult to follow the proof steps and to understand the value of the main result as it's use is not really discussed clearly. Given the mixed bag of results, some networks benefit from increased capacity and training to the second descent, others don't.

---

> ### Author Rebuttal · Authors · 2025-07-30
>
> Dear reviewer RJrF,
>
> We would like to thank you for your interest and feedback regarding our paper.
>
> We would like to remind that **overparametrization** is known to benefit generalization (through the double descent phenomenon), but its **effects on OOD detection still remain unclear**.
> In this work, we investigate how the model complexity affects the effectiveness of *post-hoc* OOD detection methods. *Post-hoc* OOD detection methods operate on top of a pretrained classifier after the model’s training process is finished and leverage internal model properties (probabilities, logits, or features) to define a scoring function measuring the prediction confidence. The objective in this paper is to investigate and better understand the **relationship** between **model complexity** and ***post-hoc* OOD detection methods**.
>
> We first focus experimentally to highlight that ***post-hoc* OOD detection methods** are **sensitive** to the **model complexity**. In particular, we observed that **logit-based** and **hybrid methods describe a double descent phenomenon**, while **feature-based methods may not** as depicted in Figure 2 and Table 1. Those results suggest that both logit-based and hybrid OOD methods performance improve as the model complexity increases in the overparametrized regime, while we may observe an opposite behavior for feature-based methods. Furthermore, we observed that some *post-hoc* OOD methods may demonstrate stronger performance in the under-parameterized regime than in the overparameterized regime for certain architectures, e.g. as shown by the CNN in Figure 2. Interestingly, we also found that Neural Collapse can be used to provide insights into whether an overparameterized or underparameterized model is more suitable for OOD detection.
>
> In the **theoretical insights**, we show that the expected OOD risk for least-squares binary classifiers on Gaussian covariate models under some assumptions exhibits an infinite peak when the number of parameters is close to the number of data, which we associate with the double descent phenomenon. This result may suggest that confidence-based *post-hoc* OOD detection methods may exhibit a double descent phenomenon.
>
> **Overall, our study provides practical guidelines for selecting *post-hoc* OOD methods and choosing the appropriate model complexity for effective OOD detection.**

---

> > ### Comment · Reviewer_RJrF · 2025-08-05
> >
> > Thanks for the response, this helps clarify the utility of the investigation to practitioners.

---

### Author Response · Authors · 2025-08-05

Dear Reviewers,

We sincerely thank you for your thoughtful, constructive, and encouraging feedback. Your time and effort in carefully reviewing our work are  appreciated.

We would also like to warmly acknowledge reviewers who initiated discussions with us.

We hope the clarifications provided during the rebuttal have addressed all concerns you may have raised in your reviews. If any ambiguities remain or further points require elaboration, we would be delighted to address them. Please feel free to contact us directly with any questions.

Once again, we thank you for your valuable insights and reviews.

Sincerely,

The authors

---

### Decision · Program_Chairs · 2025-09-17

**Decision:**

Accept (poster)

**Comment:**

The paper extends the double descent phenomenon (previously demonstrated in regression by Belkin et al. 2020) to out-of-distribution (OOD) detection. The authors show both theoretically and empirically that OOD detection performance exhibits a characteristic double descent pattern as model capacity increases.

Strengths:

- extensive experimental validation across multiple standard model architectures, benchmark datasets, and detection techniques and wide range of architectures.

- novelty of applying double descent analysis to OOD detection, which has been largely unexplored

Weaknesses:

- reliance on simplified assumptions, e.g. Gaussian data-distribution assumption

- OOD distribution and OOD risk definition potentially limited

- limited practical guidance for practitioners